# EquiReg: Equivariance Regularized Diffusion for Inverse Problems

## Abstract

Diffusion models represent the state-of-the-art for solving inverse problems such as image restoration tasks. Diffusion-based inverse solvers incorporate a likelihood term to guide prior sampling, generating data consistent with the posterior distribution. However, due to the intractability of the likelihood, most methods rely on isotropic Gaussian approximations, which can push estimates off the data manifold and produce inconsistent, poor reconstructions. We propose *Equivariance Regularized* (EquiReg) diffusion, a general plug-and-play framework that improves posterior sampling by penalizing those that deviate from the data manifold. EquiReg formalizes manifold-preferential equivariant functions that exhibit low equivariance error for on-manifold samples and high error for off-manifold ones, thereby guiding sampling toward symmetry-preserving regions of the solution space. We highlight that such functions naturally emerge when training non-equivariant models with augmentation or on data with symmetries. EquiReg is particularly effective under reduced sampling and measurement consistency steps, where many methods suffer severe quality degradation. By regularizing trajectories toward the manifold, EquiReg implicitly accelerates convergence and enables high-quality reconstructions. EquiReg consistently improves performance in linear and nonlinear image restoration tasks and solving partial differential equations.

## 1 Introduction

Inverse problems aim to recover an unknown signal $\boldsymbol{x}^* \in \mathbb{R}^d$ from undersampled noisy measurements:

$$\boldsymbol{y} = \mathcal{A}(\boldsymbol{x}^*) + \boldsymbol{\nu} \in \mathbb{R}^m, \tag{1}$$

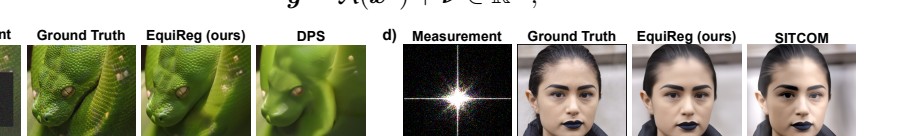

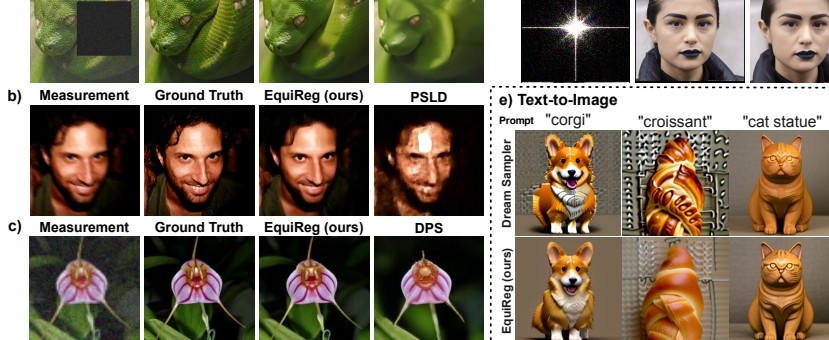

Figure 1: **EquiReg's broad applicability.** a-d) image restoration inverse problems and e) text-guided image generation, resulting in artifact reduction and more realistic generation. Here, EquiReg refers to our regularization being applied to the diffusion sampling method on the same row.

where $\mathcal{A}$ is a known measurement operator, and $\boldsymbol{\nu}$ is an unknown noise (Groetsch, 1993). Inverse problems are widely studied in science and engineering, including imaging and astrophotography.

Inverse problems are ill-posed, i.e., the inversion process can have many solutions; hence, they require prior information about the desired solution (Kabanikhin, 2008). In the Bayesian formulation, the solution maximizes the posterior distribution $p(\boldsymbol{x}|\boldsymbol{y}) \propto p(\boldsymbol{y}|\boldsymbol{x})p(\boldsymbol{x})$, where $p(\boldsymbol{y}|\boldsymbol{x})$ is the likelihood

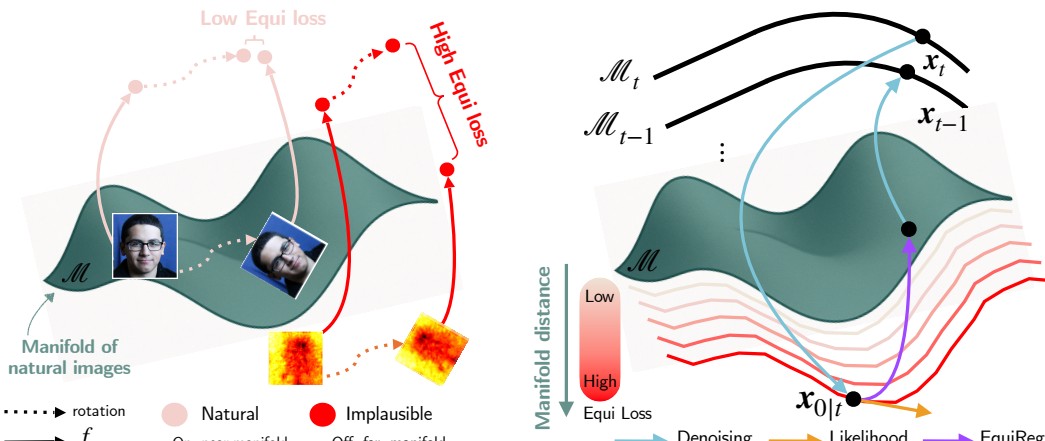

Figure 2: **Equivariance Regularized (EquiReg) diffusion for inverse problems.** (left) Manifold preferential equivariance (MPE) functions whose equivariance error is lower for on-manifold and higher for off-manifold data. (right) EquiReg regularizes the posterior sampling trajectory for improved performance. It penalizes off-manifold trajectories via MPE-based regularization.

of the measurements and $p(\boldsymbol{x})$ is a prior describing the signal structure (Stuart, 2010). Examples of handcrafted priors include sparsity (Donoho, 2006) and low-rankness (Candès et al., 2011).

This paper focuses on methods that leverage unconditionally pre-trained score-based generative diffusion models as learned priors (Ho et al., 2020; Song and Ermon, 2019) with applications in image restoration (Chung et al., 2023), medical imaging (Chung et al., 2022a), and solving partial differential equations (PDEs) (Huang et al., 2024; Yao et al., 2025). These methods define a sequential noising process $\boldsymbol{x}_0 \sim p_{\text{data}} \rightarrow \boldsymbol{x}_t \rightarrow \boldsymbol{x}_T \sim p_T(\boldsymbol{x}) \approx \mathcal{N}(\boldsymbol{0}, \boldsymbol{I})$ and a reverse denoising process parameterized by a neural network score $\nabla_{\boldsymbol{x}_t} \log p_t(\boldsymbol{x}_t)$ (Vincent, 2011). During sampling, these approaches incorporate gradient signals carrying likelihood information to solve inverse problems.

Solving inverse problems with diffusion (Zhang et al., 2025a; Alkhouri et al., 2025) requires computing the conditional score $\nabla_{\boldsymbol{x}_t} \log p_t(\boldsymbol{x}_t|\boldsymbol{y})$, decomposed into $\nabla_{\boldsymbol{x}_t} \log p_t(\boldsymbol{x}_t) + \nabla_{\boldsymbol{x}_t} \log p_t(\boldsymbol{y}|\boldsymbol{x}_t)$. This introduces challenges, as the likelihood score $\nabla_{\boldsymbol{x}_t} \log p_t(\boldsymbol{y}|\boldsymbol{x}_t) = \nabla_{\boldsymbol{x}_t} \log \int p(\boldsymbol{y}|\boldsymbol{x}_0) p_t(\boldsymbol{x}_0|\boldsymbol{x}_t) \mathrm{d}\boldsymbol{x}_0$ is only computationally tractable when $t = 0$. To handle the likelihood for $t > 0$, many methods approximate the posterior $p_t(\boldsymbol{x}_0|\boldsymbol{x}_t)$ with the isotropic Gaussian distribution (Zhang et al., 2025a), where the distribution expectation is computed using the optimal denoising score (Robbins, 1956). The Gaussian approximation can be inaccurate for complex distributions (Figure 3), leading to errors in likelihood computation, especially with point estimations (Chung et al., 2023). Since the posterior expectation is a conditional expectation, a linear combination of all possible $\boldsymbol{x}_0$, it may lie off the data manifold even when individual samples remain on it. These issues are further amplified in latent diffusion models (LDMs), introducing artifacts (Rout et al., 2023).

Prior work has attempted to address this challenge via projection-based (He et al., 2024; Zirvi et al., 2025) or decoupled optimization strategies (Zhang et al., 2025a), aimed at reducing the propagation of measurement consistency errors during sampling. However, they still rely on the isotropic Gaussian assumption, which can lead to failures on difficult tasks or when the number of sampling steps is reduced. While higher-order statistics can reduce errors (Boys et al., 2024), most approaches still rely on the approximation for its efficiency, scalability, and simplicity (Alkhouri et al., 2025), often coupled with large-scale LDMs (Peebles and Xie, 2023). This raises a key question: how can we ensure the reliability and practicality of conditional diffusion models under this approximation?

Equivariance provides a natural mechanism to keep sampling trajectories close to the data manifold. We therefore address this challenge with a regularization scheme that leverages equivariance to improve posterior sampling by guiding diffusion trajectories toward symmetry-preserving solution spaces. Prior studies have enforced equivariance directly on the generation or denoising process (Chen et al., 2023a; Terris et al., 2024), with extensions to probabilistic symmetries (Bloem-Reddy et al., 2020) enabling more sample-efficient diffusion models (Wang et al., 2024).

Our approach differs as follows: rather than strictly enforcing equivariance within denoising architectures, which can hinder tasks requiring symmetry breaking (Lawrence et al., 2025), we employ equivariance as a plug-and-play regularizer to guide diffusion trajectories toward the data manifold.

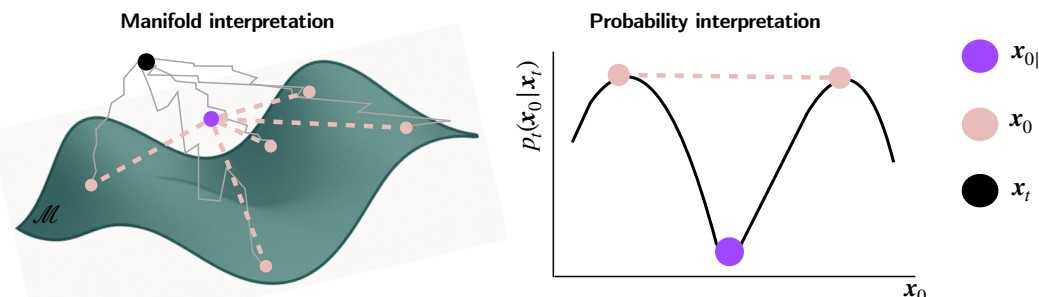

Figure 3: **Off-manifold posterior expectation.** This impacts the likelihood score $p_t(\boldsymbol{y}|\boldsymbol{x}_t) = \int p(\boldsymbol{y}|\boldsymbol{x}_0)p_t(\boldsymbol{x}_0|\boldsymbol{x}_t)\mathrm{d}\boldsymbol{x}_0$ computation achieved via isotropic Gaussian modelling of $p_t(\boldsymbol{x}_0|\boldsymbol{x}_t)$.

**Our contributions.** We propose *Equivariance Regularized* (EquiReg) diffusion, an equivariance-based regularization framework for solving inverse problems with diffusion models (Figure 2). EquiReg leverages equivariance to *regularize* likelihood-induced errors during posterior sampling, guiding diffusion trajectories toward more consistent, on-manifold solutions. Crucially, it employs *Manifold-Preferential Equivariant* (MPE) functions, which discriminate on-manifold from off-manifold data by exhibiting low equivariance error in-distribution and higher error out-of-distribution. We formalize that an effective regularizer should capture such a global property, and MPE functions provide a principled way to direct sampling toward plausible solutions. This design makes EquiReg architecture-agnostic: the regularizer operates independently of the diffusion model itself. With a suitable MPE function, EquiReg improves performance across models, including those with equivariant scores, where likelihood guidance may otherwise push trajectories off the manifold.

We observe that many practical functions behave as MPEs: their equivariance error is small on the training or data manifold but grows off-manifold. This behavior arises in learned models trained with data augmentation, as well as in data with inherent symmetries such as those from physical systems. Rather than treating the degradation off-manifold as a limitation, we exploit it as a signal: equivariance error serves as a natural discriminator of equivariance for identifying undesirable states during diffusion sampling. Building on this idea, we construct pre-trained MPEs as the foundation of our EquiReg loss. The choice of this function is independent of the denoiser in diffusion models and can be derived separately. For instance, if the diffusion model architecture is itself equivariant, it cannot be leveraged for regularization as it cannot discriminate between on- and off-manifold samples. Instead, a separate non-equivariant architecture can be used to train to derive EquiReg.

We validate the effectiveness of EquiReg through extensive experiments across diverse diffusion models, inverse problems, and datasets. We demonstrate that EquiReg improves perceptual image quality and remains effective in cases where baselines fail. We show that EquiReg improves the performance of SITCOM (Alkhouri et al., 2025) and DPS (Chung et al., 2023) when the number of measurement consistency and sampling steps are reduced, thus moving toward more efficient diffusion-based solvers. Our method is particularly useful when applied to LDMs. EquiReg reduces failure cases, and consistently improves PSLD (Rout et al., 2023), ReSample (Song et al., 2023a), and DPS (Chung et al., 2023) on linear and nonlinear image restoration tasks. For example, EquiReg significantly improves the LPIPS (Song et al., 2023a) of ReSample by $51\%$ for motion deblur and the FID of DPS (Chung et al., 2023) by $59\%$ on super-resolution. We also include diversity analyses, demonstrating that EquiReg maintains diversity without collapse of single mode reconstruction.

We extend EquiReg's applicability to function-space diffusion models and demonstrate its added benefit for solving PDEs. EquiReg achieves a $7.3\%$ relative reduction in the $\ell_2$ error of FunDPS (Mammadov et al., 2024a; Yao et al., 2025) on the Helmholtz equation and a $7.5\%$ relative reduction on the Navier-Stokes equation. Lastly, we include preliminary experiments on EquiReg improving the realism and plausibility of text-guided image generation, emphasizing that the benefits of EquiReg extend beyond image restorations. Overall, the flexibility of EquiReg as a plug-and-play regularization framework suggests that its utility will extend well beyond the specific methods studied in this paper.

## 2 PRELIMINARIES AND RELATED WORKS

**Diffusion models.** Diffusion generative models (Ho et al., 2020; Song and Ermon, 2019; Sohl-Dickstein et al., 2015; Kadkhodaie and Simoncelli, 2021) are state-of-the-art in computer vision for image (Esser et al., 2024) and video generation (Brooks et al., 2024; Zhang et al., 2025b), with score-based methods (Song et al., 2021) being among the most widely used. Diffusion models generate data

via a reverse noising process. The forward noising process transforms the data sample $\boldsymbol{x}_0 \sim p_{\text{data}}$ via a series of additive noise into an approximately Gaussian distribution ($p_{\text{data}} \to p_t \to \mathcal{N}(0, I)$ as $t \to \infty$), described by the stochastic differential equation (SDE) $\mathrm{d}\boldsymbol{x} = -\frac{\beta_t}{2}\boldsymbol{x}\mathrm{d}t + \sqrt{\beta_t}\mathrm{d}\boldsymbol{w}$, where $\boldsymbol{w}$ is a standard Wiener process, and the drift and diffusion coefficients are parameterized by a monotonically increasing noise scheduler $\beta_t \in (0, 1)$ in time $t$ (Ho et al., 2020). Reversing the forward diffusion process is described by (Anderson, 1982)

$$\mathrm{d}\boldsymbol{x} = [-\tfrac{\beta_t}{2}\boldsymbol{x} - \beta_t \nabla_{\boldsymbol{x}_t} \log p_t(\boldsymbol{x}_t)]\,\mathrm{d}t + \sqrt{\beta_t}\mathrm{d}\bar{\boldsymbol{w}} \tag{2}$$

with $\mathrm{d}t$ moving backward in time or in discrete steps from $T$ to $0$. This reverse SDE is used to sample data from the distribution $p_{\text{data}}$, where the unknown gradient $\nabla_{\boldsymbol{x}_t} \log p_t(\boldsymbol{x}_t)$ is approximated by a scoring function $s_\theta(\boldsymbol{x}_t, t)$, parameterized by a neural network and learned via denoising score matching methods (Hyvärinen and Dayan, 2005; Vincent, 2011). Solving inverse problems is described as a conditional generation where the data is sampled from the posterior $p(\boldsymbol{x}|\boldsymbol{y})$:

$$\mathrm{d}\boldsymbol{x} = [-\tfrac{\beta_t}{2}\boldsymbol{x}\mathrm{d}t - \beta_t(\nabla_{\boldsymbol{x}_t} \log p_t(\boldsymbol{x}_t) + \nabla_{\boldsymbol{x}_t} \log p_t(\boldsymbol{y}|\boldsymbol{x}_t))]\mathrm{d}t + \sqrt{\beta_t}\mathrm{d}\bar{\boldsymbol{w}} \tag{3}$$

For solving general inverse problems where the diffusion is *pre-trained* unconditionally, the prior score $\nabla_{\boldsymbol{x}_t} \log p_t(\boldsymbol{x}_t)$ can be estimated using $s_\theta(\boldsymbol{x}_t, t)$. However, the likelihood score $\nabla_{\boldsymbol{x}_t} \log p_t(\boldsymbol{y}|\boldsymbol{x}_t)$ is only known at $t = 0$, otherwise it is computationally intractable.

**Diffusion models for inverse problems.** Solving inverse problems with pre-trained diffusion models requires approximating the intractable likelihood score $\nabla_{\boldsymbol{x}_t} \log p_t(\boldsymbol{y}|\boldsymbol{x}_t)$. Training-free solvers differ in how they approximate $p_t(\boldsymbol{y}|\boldsymbol{x}_t)$ and combine it with the sampling prior $p_t(\boldsymbol{x}_t)$ (Peng et al., 2024). Since $p_t(\boldsymbol{y}|\boldsymbol{x}_t) = \int p(\boldsymbol{y}|\boldsymbol{x}_0)p_t(\boldsymbol{x}_0|\boldsymbol{x}_t)\mathrm{d}\boldsymbol{x}_0$, the common choice is to approximate $p_t(\boldsymbol{x}_0|\boldsymbol{x}_t)$ by an isotropic Gaussian $\mathcal{N}(\boldsymbol{x}_{0|t}, r_t^2 \boldsymbol{I})$ (Chung et al., 2023; Song et al., 2023b; Zhu et al., 2023; Zhang et al., 2025a). With an optimal denoising score $s_\theta(\boldsymbol{x}_t, t)$, the posterior mean $\boldsymbol{x}_{0|t} := \mathbb{E}[\boldsymbol{x}_0|\boldsymbol{x}_t]$ follows from Tweedie's formula (Robbins, 1956; Miyasawa et al., 1961; Efron, 2011). Although this yields an MMSE estimate, for complex or multimodal distributions, $p_t(\boldsymbol{x}_0|\boldsymbol{x}_t)$ may not be concentrated around its mean, leading to off-manifold solutions (see Figure 3).

**Equivariance.** Equivariance is a property describing how functions transform predictably under group actions. It serves as a powerful strategy for incorporating symmetries into deep learning (Bronstein et al., 2021). Prior work has applied equivariance to graph networks (Satorras et al., 2021), convolutional networks (Cohen and Welling, 2016; Romero and Lohit, 2022), Lie groups for modelling dynamical systems (Finzi et al., 2020), and diffusion models (Wang et al., 2024) with applications in molecular generation (Hoogeboom et al., 2022; Cornet et al., 2024), autonomous driving (Chen et al., 2023b), robotics (Brehmer et al., 2023), crystal structure prediction (Jiao et al., 2023), and audio inverse problems (Moliner et al., 2023). Equivariance guidance has also been used to improve temporal consistency in video generation (Daras et al., 2024). The benefits of equivariance as a prior to solve inverse problems (Scanvic et al., 2025) are theoretically supported in compressed sensing (Tachella et al., 2023). An equivariant function respects symmetries under group transformations, i.e.,

**Definition 2.1** (Equivariance). *Let $G$ act on $\mathcal{Z}$ via $T_g : \mathcal{Z} \to \mathcal{Z}$ and on $\mathcal{X}$ via $S_g : \mathcal{X} \to \mathcal{X}$. A function $f : \mathcal{Z} \to \mathcal{X}$ is equivariant if for all $g \in G$ and $\boldsymbol{z} \in \mathcal{Z}$, $f(T_g(\boldsymbol{z})) = S_g(f(\boldsymbol{z}))$.*

While prior work leverages exact equivariance as in Definition 2.1 to directly incorporate symmetries into deep neural networks, recent studies explore approximate equivariant networks to relax strict mathematical symmetries that may not fully hold in real-world data, aiming to improve performance (Wang et al., 2022). They propose a definition of approximate equivariance (Definition 2.2), along with an equivariance error of functions to quantify the deviation from perfect symmetry.

**Definition 2.2** (Approximate Equivariant Functions). *Let $G$ act on $\mathcal{Z}$ via $T_g : \mathcal{Z} \to \mathcal{Z}$ and on $\mathcal{X}$ via $S_g : \mathcal{X} \to \mathcal{X}$. A function $f : \mathcal{Z} \to \mathcal{X}$ is $\epsilon$-approximate equivariant if for all $g \in G$ and $\boldsymbol{z} \in \mathcal{Z}$, $\|S_g(f(\boldsymbol{z})) - f(T_g(\boldsymbol{z}))\| \leq \epsilon$. The equivariance error of the function $f : \mathcal{Z} \to \mathcal{X}$ is defined as $\sup_{\boldsymbol{z}, g} \|S_g(f(\boldsymbol{z})) - f(T_g(\boldsymbol{z}))\|$. Hence, $f$ is $\epsilon$-approximate equivariant iff its error $< \epsilon$.*

Finally, this paper uses the term manifold which refers to the data manifold hypothesis (see Assumption H.1 ) (Cayton et al., 2005) that assumes data is sampled from a low-dimensional manifold embedded in a high-dimensional space. This hypothesis is popular in machine learning (Bordt et al., 2023) and diffusion-based solvers (He et al., 2024; Chung et al., 2022b; 2023), supported by empirical evidence for imaging (Weinberger and Saul, 2006).

## 3 EQUIREG: EQUIVARIANCE REGULARIZED DIFFUSION

We begin by presenting a generalized regularization framework for improving diffusion-based inverse solvers. We then focus on the property of *equivariance* and introduce a new class of functions whose equivariance errors are distribution-dependent (low for on- or near-manifold samples and high for off-manifold samples). Finally, we leverage these functions to regularize diffusion models, guiding sampling trajectories toward better inverse solutions.

This paper addresses the propagation error introduced by the approximation of posterior $p_t(\boldsymbol{x}_0|\boldsymbol{x}_t)$ by incorporating an explicit regularization term. The proposed framework is general and can be applied as plug-in on a wide range of pixel and latent-space diffusion models. Given $p_t(\boldsymbol{y}|\boldsymbol{x}_t) = \int p(\boldsymbol{y}|\boldsymbol{x}_0)p_t(\boldsymbol{x}_0|\boldsymbol{x}_t)\mathrm{d}\boldsymbol{x}_0$, let $\tilde{p}_t(\boldsymbol{x}_0|\boldsymbol{x}_t)$ denote an approximation of the posterior to make the likelihood tractable. We formulate the regularized reverse diffusion dynamics as

$$\mathrm{d}\boldsymbol{x} = [-\tfrac{\beta_t}{2}\boldsymbol{x}\mathrm{d}t - \beta_t\nabla_{\boldsymbol{x}_t}(\log p_t(\boldsymbol{x}_t) + \log \int p(\boldsymbol{y}|\boldsymbol{x}_0)\tilde{p}_t(\boldsymbol{x}_0|\boldsymbol{x}_t)\mathrm{d}\boldsymbol{x}_0 - \mathcal{R}(\boldsymbol{x}_t))]\mathrm{d}t + \sqrt{\beta_t}\mathrm{d}\bar{\boldsymbol{w}}, \quad (4)$$

where $\mathcal{R}(\boldsymbol{x}_t)$ is the regularizer. Applying this to DPS (Chung et al., 2023) takes the form in Algorithm 1). This formulation raises two questions: i) how to design the regularizer, and ii) how to interpret the role of $\mathcal{R}$ in regularizing conditional diffusion models and its impact on the sampling trajectory. We gain insight into the desirable properties of an optimal regularizer by reinterpreting the reverse conditional diffusion process as a time-inhomogeneous Wasserstein gradient flow (Ferreira and Valencia-Guevara, 2018) (see Propositions G.1 and G.2 in Appendix). The analysis clarifies that an ideal regularizer should yield low values for on-manifold and high values for off-manifold samples, enabling accurate posterior sampling even when the likelihood score is approximated.

We further interpret this property in terms of sampling dynamics, i.e., when applied at each reverse-diffusion step, the regularizer effectively penalizes trajectories leaving the data manifold and reinforces those aligned with high-probability regions. This motivates designing a regularizer that corrects the entire functional being minimized globally, in contrast to prior works that focus only on locally reducing likelihood error. The ideal property of a useful regularizer would be to produce high error on undesirable samples and low error on desirable samples. We instantiate this

**Algorithm 1** Equi-DPS for Inverse Problems.

**Require:** $T, \boldsymbol{y}, \{\zeta_t\}_{t=1}^T, \{\tilde{\sigma}_t\}_{t=1}^T, \boldsymbol{s}_\theta, \mathcal{R}(\cdot), \{\lambda_t\}_{t=1}^T$
1: $\boldsymbol{x}_T \sim \mathcal{N}(\boldsymbol{0}, \boldsymbol{I})$
2: **for** $t = T - 1$ **to** $0$ **do**
3: $\quad \hat{\boldsymbol{s}} \leftarrow \boldsymbol{s}_\theta(\boldsymbol{x}_t, t)$
4: $\quad \boldsymbol{x}_{0|t} \leftarrow \frac{1}{\sqrt{\bar{\alpha}_t}}(\boldsymbol{x}_t + (1 - \bar{\alpha}_t)\hat{\boldsymbol{s}})$
5: $\quad \boldsymbol{\epsilon} \sim \mathcal{N}(\boldsymbol{0}, \boldsymbol{I})$
6: $\quad \boldsymbol{x}'_{t-1} \leftarrow \frac{\sqrt{\alpha_t}(1-\bar{\alpha}_{t-1})}{1-\bar{\alpha}_t}\boldsymbol{x}_t + \frac{\sqrt{\bar{\alpha}_{t-1}}\beta_t}{1-\bar{\alpha}_t}\boldsymbol{x}_{0|t} + \tilde{\sigma}_t\boldsymbol{\epsilon}$
7: $\quad \boldsymbol{x}_{t-1} \leftarrow \boldsymbol{x}'_{t-1} - \zeta_t\nabla_{\boldsymbol{x}_t}\|\boldsymbol{y} - \mathcal{A}(\boldsymbol{x}_{0|t})\|_2^2$
8: $\quad \boldsymbol{x}_{t-1} \leftarrow \boldsymbol{x}_{t-1} - \lambda_t\nabla_{\boldsymbol{x}_t}\mathcal{R}(\boldsymbol{x}_t)$
9: **end for**
10: **return** $\boldsymbol{x}_0$

ideal regularizer using equivariance, a global property that enforces geometric symmetries and guides the diffusion process toward the data manifold. To realize this idea, we seek functions that exhibit approximate equivariance and discriminate on- from off-manifold samples.

Thus, we propose to quantify equivariance of a function relative to a data distribution. Specifically, while the literature has primarily studied the equivariance properties of functions for general inputs, we propose a new definition for functions in which their equivariance error is distribution-dependent and defined under the support of an input data distribution (Definition 3.1).

**Definition 3.1** (Distribution-Dependent Equivariant Functions). *Let $G$ act on $\mathcal{Z}$ via $T_g : \mathcal{Z} \to \mathcal{Z}$ and on $\mathcal{X}$ via $S_g : \mathcal{X} \to \mathcal{X}$. The equivariance error of the function $f : \mathcal{Z} \to \mathcal{X}$ under the distribution $p$ is defined as $\sup_g \mathbb{E}_{\boldsymbol{z}\sim p}[\|S_g(f(\boldsymbol{z})) - f(T_g(\boldsymbol{z}))\|]$.*

The above definition enables us to define functions whose equivariance error can differentiate on-manifold samples from off-manifold ones. Particularly, we aim to find functions whose equivariance error is low for on-manifold data and high elsewhere. We also introduce a constrained version of equivariance error, where the input is implicitly regularized to lie on the manifold $\mathcal{M}$ in addition to minimizing the equivariance error (Definition 3.2). Both equivariance errors are non-local, defined at the distribution level. When used to regularize the reverse conditional diffusion process, they are computed via local evaluations over the sampled data.

**Definition 3.2** (Manifold-Constrained Distribution-Dependent Equivariant Functions). *Let $G$ act on $\mathcal{Z}$ via $T_g : \mathcal{Z} \to \mathcal{Z}$ and on $\mathcal{X}$ via $S_g : \mathcal{X} \to \mathcal{X}$. The manifold-constrained equivariance error of the function $f : \mathcal{Z} \to \mathcal{X}$ under the data distribution $p$ is $\sup_g \mathbb{E}_{\boldsymbol{z}\sim p}[\|\boldsymbol{z} - h(S_g^{-1}(f(T_g(\boldsymbol{z}))))\|]$ where $h : \mathcal{X} \to \mathcal{Z}$, and the pair $(f, h)$ forms a vanishing-error autoencoder (see Appendix I).*

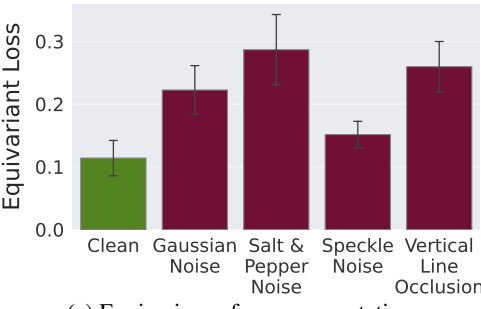 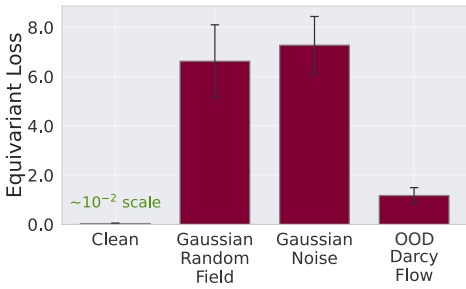

(a) Equivariance from augmentation.    (b) Equivariance from data symmetries.

Figure 4: **MPE function examples.**

To define our method, we term a class of *manifold-preferential equivariant (MPE)* functions, whose equivariance error is lower for samples on the data manifold than for off-manifold samples. EquiReg is a regularization framework, not a manifold projection method. EquiReg penalizes states that deviate from symmetry-preserving regions; when an MPE function is used, these regions align with the data manifold. In practice, MPE functions can emerge in different ways, which we illustrate with examples from augmented training and from data symmetries. MPE can emerge when functions are trained with symmetry-preserving mechanisms such as data augmentation. Prior work has studied equivariant properties of learned representations in deep networks (Lenc and Vedaldi, 2015), showing that data augmentations (Krizhevsky et al., 2012) and representation compression via reduced model capacity (Bruintjes et al., 2023) promote equivariant features even when equivariance is not explicitly built into the architecture. Importantly, the trained network is only approximately equivariant, and prior studies have noted that symmetry-preserving properties degrade for inputs deviating from in-distribution data (Azulay and Weiss, 2019). A few studies have leveraged this emergent MPE in trained networks for out-of-distribution detection (Zhou, 2022; Kaur et al., 2022; 2023).

To demonstrate the widespread MPE property of learned mappings, we have considered additional pre-trained models and quantified their equivariance loss for several set of data distributions, i.e., natural images and corrupted ones (see Section I of Appendix.) Figure 4a illustrates the MPE property, emergent via training with augmentations, of $\mathcal{E}$-$\mathcal{D}$ of a pre-trained autoencoder, currently used in LDMs. Specifically, it shows that the equivariance error is lower for natural images and increases when images deviate from the clean data distribution. Based on Definitions 3.1 and 3.2, we propose *Equi* and *EquiCon* losses using a pre-trained encoder-decoder for diffusion-based inverse solvers:

$$\text{Equi}_{\text{pixel}} \ \mathcal{R}(\boldsymbol{x}_t) = \|S_g(\mathcal{E}(\boldsymbol{x}_{0|t})) - \mathcal{E}(T_g(\boldsymbol{x}_{0|t}))\|_2^2$$
$$\text{Equi}_{\text{latent}} \ \mathcal{R}(\boldsymbol{z}_t) = \|S_g(\mathcal{D}(\boldsymbol{z}_{0|t})) - \mathcal{D}(T_g(\boldsymbol{z}_{0|t}))\|_2^2 \tag{5}$$
$$\text{EquiCon}_{\text{latent}} \ \mathcal{R}(\boldsymbol{z}_t) = \|\boldsymbol{z}_{0|t} - \mathcal{E}(S_g^{-1}(\mathcal{D}(T_g(\boldsymbol{z}_{0|t}))))\|_2^2,$$

where $\boldsymbol{x}_{0|t}$ and $\boldsymbol{z}_{0|t}$ are function of $\boldsymbol{x}_t$ and $\boldsymbol{z}_t$, respectively. MPE can also emerge due to symmetries present in the data itself during training. This often occurs in physics systems where coefficient functions, boundary values, and solution functions of PDEs remain valid under invertible coordinate transformations. Formally, let $\mathcal{G}(a) \mapsto u$ be a PDE operator mapping initial condition $a$ to solution $u$, and let $T_g$ and $S_g$ be invertible transformations that preserve PDE structure and boundary conditions. Then, $S_g(\mathcal{G}(a)) = \mathcal{G}(T_g(a))$. Neural operators (Kovachki et al., 2021), popular architectures for modelling physics, trained on PDEs with such inherent symmetries can learn equivariance properties. Figure 4b shows that we can construct an MPE function with Fourier Neural Operators (FNOs (Li et al., 2021)) trained on non-augmented physics data for Navier-Stokes, yielding lower error $\|S_g(\text{FNO}(\boldsymbol{x}_{0|t})) - \text{FNO}(T_g(\boldsymbol{x}_{0|t}))\|_2^2$ on in-distribution as opposed to out-of-distribution data, with reflection as the group action.

The key message from our MPE examples is that MPE properties naturally emerge when a function (e.g., a neural network) is trained with appropriate augmentations or when the data itself exhibits inherent symmetries. Our paper leverages this property to distinguish on-manifold samples from off-manifold ones and to regularize the posterior sampling trajectory toward high-probability regions. Finally, we note that the choice of symmetry group may often be a challenge depending on application domain, a shared challenge in the broader equivariance literature. We provide guidelines on how to choose symmetry groups in Section H with literature reference on automatic symmetry discovery from data (Zhou et al.; Quessard et al., 2020; Dehmamy et al., 2021; Mohapatra et al., 2025).

Table 1: **Robustness and computational efficiency of applying EquiReg under various periods during sampling.** EquiReg maintains performance when applied every $\{1, 2, 5, 10\}$ DDIM steps while incurring minimal computational overhead.

| | | Super Resolution | | | Gaussian Blur | | | |
|---|---|---|---|---|---|---|---|---|
| Method | Period | Runtime (s) | PSNR↑ | LPIPS↓ | FID↓ | Runtime (s) | PSNR↑ | LPIPS↓ | FID↓ |
| DPS | N/A | 46.20 | 22.99 (1.93) | 0.20 (0.05) | 135.7 | 46.50 | 24.59 (2.25) | 0.15 (0.03) | 88.70 |
| Equi-DPS | 1 | 51.10 | 26.73 (1.99) | 0.12 (0.03) | 87.97 | 52.20 | 26.08 (2.25) | 0.12 (0.03) | 87.11 |
| Equi-DPS | 2 | 48.90 | 26.73 (1.99) | 0.12 (0.03) | 87.98 | 49.10 | 26.06 (2.24) | 0.12 (0.03) | 87.19 |
| Equi-DPS | 5 | 47.10 | 26.73 (1.99) | 0.12 (0.03) | 87.98 | 47.30 | 26.06 (2.24) | 0.12 (0.03) | 87.32 |
| Equi-DPS | 10 | 46.90 | 26.73 (1.99) | 0.12 (0.03) | 87.99 | 47.00 | 26.05 (2.24) | 0.12 (0.03) | 87.04 |

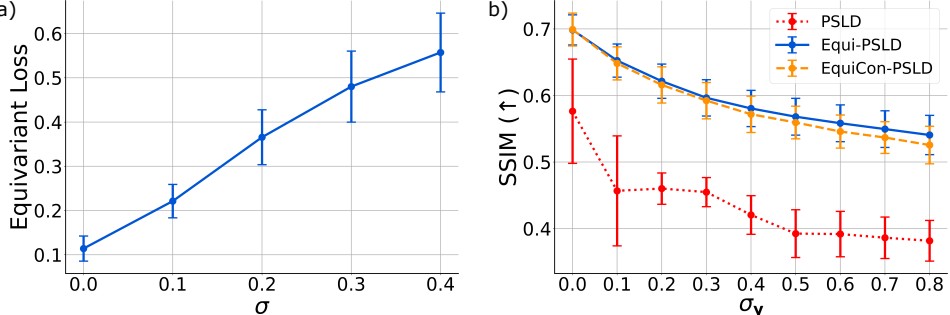

Figure 5: **EquiReg is effective across a range of measurement noise levels.** a) Equivariance error computed over a pre-trained decoder on increasingly noisy inputs. b) EquiReg performance computed over a range of measurement noise levels on the FFHQ dataset.

## 4 RESULTS

This section provides experimental results on the performance of EquiReg for inverse problems, including linear and nonlinear image restoration tasks and solving PDEs. To fairly assess EquiReg's impact, we deliberately use a duo-setting comparison (e.g., PSLD vs. Equi-PSLD) across experiments, where all other factors (architecture, training, sampling) remain fixed. This ensures that any observed improvement can be attributed to EquiReg, not the underlying model or inference procedure. We also evaluate the impact of EquiReg under reduced measurement consistency and sampling steps, providing a path toward faster diffusion-based inverse solvers. Results emphasize the usefulness of EquiReg when the baseline performance deteriorates. Lastly, we provide preliminary analysis on EquiReg improving the realism of text-guided image generation.

**Image restoration tasks.** We evaluate the performance of EquiReg when applied to: SITCOM (Alkhouri et al., 2025), PSLD (Rout et al., 2023), ReSample (Song et al., 2023a), and DPS (Chung et al., 2023). We compare against several manifold-preserving or geometry-constraint approaches including MCG (Chung et al., 2022b), MPGD-AE (He et al., 2024), and DiffState-Grad (Zirvi et al., 2025). We measure performance via perceptual similarity (LPIPS), distribution alignment (FID), pixel-wise fidelity (PSNR), and structural consistency (SSIM). We test EquiReg on two datasets: a) the FFHQ $256 \times 256$ validation dataset (Karras et al., 2021), and b) the ImageNet $256 \times 256$ validation dataset (Deng et al., 2009). For pixel-based experiments, we use i) the pre-trained model from (Chung et al., 2023) on FFHQ, and ii) the pre-trained model from (Dhariwal and Nichol, 2021) on ImageNet. For latent diffusion experiments, we use i) the unconditional LDM-VQ-4 model (Rombach et al., 2022) on FFHQ, and ii) the Stable Diffusion v1.5 (Rombach et al., 2022) model on ImageNet.

Table 2: **EquiReg improves SITCOM under reduced measurement consistency steps ($K_{\text{meas}}$).** We reduce $K_{\text{meas}}$ and add an equal amount of EquiReg steps ($K_{\text{EquiReg}}$). Evaluated on motion deblur for FFHQ sampled with 50 DDIM steps.

| $K_{\text{meas.}}$ | $K_{\text{EquiReg}}$ | PSNR↑ | SSIM↑ | Runtime (s) |
|---|---|---|---|---|
| 10 | N/A | 28.06 | 0.81 | 21.57 |
| 5 | 5 | **29.26** | **0.83** | **11.09** |
| 20 | N/A | 27.04 | 0.79 | 38.85 |
| 10 | 10 | **28.93** | **0.82** | **20.92** |
| 30 | N/A | 27.79 | 0.80 | 58.84 |
| 15 | 15 | **29.63** | **0.84** | **30.19** |
| 40 | N/A | **30.40** | **0.85** | 78.08 |
| 20 | 20 | 29.50 | 0.83 | **41.02** |
| 60 | N/A | 28.35 | 0.81 | 108.57 |
| 30 | 30 | **31.36** | **0.87** | **59.38** |

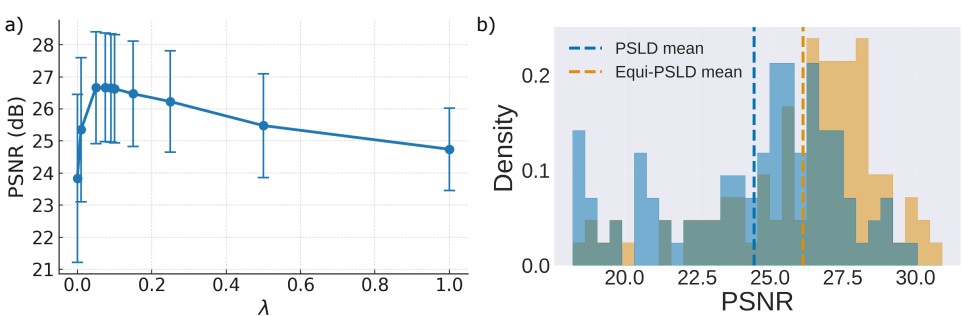

Figure 6: **Robustness of EquiReg, demonstrated on PSLD.** a) EquiReg is robust to the choice of $\lambda_t$. b) EquiReg reduces failure cases and enhances reconstruction fidelity for super-resolution on FFHQ.

We evaluate EquiReg on a variety of linear and nonlinear restoration tasks for natural images (see Section E for task details). We adopt the pre-trained encoder-decoder $\mathcal{E}$-$\mathcal{D}$ as our MPE function. For FFHQ, we use vertical reflection as the symmetry group, which preserves upright facial orientation. For ImageNet, we define a rotation group $G = \{0, \pi/2, \pi, 3\pi/2\}$, and uniformly at random select the group action for each sample. Finally, the loss functions given in Equation (5) are used to regularize. We note that while our main experiment explore the reflection and rotation groups with small cardinality, EquiReg does not rely on full group coverage. Sampling even a sparse or randomly chosen subset of group actions is sufficient, as long as the function used for regularization exhibits the MPE property across the group (see additional experiment in Appendix).

First, we show that adding EquiReg optimization steps consistently enables SITCOM to achieve superior performance with significantly faster runtime using fewer measurement consistency steps (Table 2). Next, we show that EquiReg maintains strong performance even as the number of DDIM steps is reduced, whereas DPS suffers a significant drop; Equi-DPS consistently outperforms DPS, with the performance gap widening at lower step counts (Figure 13). We also show that EquiReg is able to preserve performance when applied with lower frequency (Table 1).

Table 3, Table 4a, and Table 5 highlights the benefits of EquiReg for latent diffusion models by consistently improving the performance of ReSample and PSLD across several tasks on both FFHQ and ImageNet. We attribute this improvement in part to the reduction of failure cases (Figure 6b). EquiReg also significantly improves the performance of pixel-based methods (see Equi-DPS vs. DPS, Table 4b).

We observe that EquiReg achieves its largest improvements on perceptual metrics (FID and LPIPS), suggesting it generates more realistic images that lie closer to the data manifold (see Appendix E for

Table 3: **EquiReg for ReSample on linear and nonlinear tasks.** FFHQ $256 \times 256$ with $\sigma_y = 0.01$.

| Task | Method | LPIPS↓ | FID↓ | PSNR↑ | SSIM↑ |
|------|--------|--------|------|-------|-------|
| *Linear* | | | | | |
| Gaussian deblur | ReSample | 0.253 | 55.65 | 27.78 | 0.757 |
| | Equi-ReSample | 0.197 | 64.86 | **29.08** | **0.825** |
| | EquiCon-ReSample | **0.156** | **54.72** | 28.18 | 0.777 |
| Motion deblur | ReSample | 0.160 | 40.14 | 30.55 | 0.854 |
| | Equi-ReSample | 0.120 | 46.28 | **30.92** | **0.870** |
| | EquiCon-ReSample | **0.078** | **37.61** | 30.73 | 0.860 |
| Super-res. ($\times 4$) | ReSample | 0.204 | 40.46 | 28.02 | 0.790 |
| | Equi-ReSample | **0.098** | 43.56 | **29.74** | **0.849** |
| | EquiCon-ReSample | 0.112 | **40.38** | 28.27 | 0.801 |
| Box inpainting | ReSample | 0.198 | 108.30 | 19.91 | 0.807 |
| | Equi-ReSample | **0.150** | **59.69** | **22.56** | **0.832** |
| | EquiCon-ReSample | 0.171 | 110.70 | 21.04 | 0.815 |
| Random inpainting | ReSample | 0.115 | 36.12 | 31.27 | 0.892 |
| | Equi-ReSample | **0.047** | 29.88 | **31.47** | **0.908** |
| | EquiCon-ReSample | **0.047** | **28.81** | 31.21 | 0.904 |
| *Nonlinear* | | | | | |
| HDR | ReSample | 0.190 | **49.06** | **24.88** | **0.819** |
| | Equi-ReSample | **0.133** | 49.52 | 24.71 | 0.815 |
| | EquiCon-ReSample | 0.135 | 49.98 | 24.67 | 0.817 |
| Phase retrieval | ReSample | 0.237 | 97.86 | 27.61 | 0.750 |
| | Equi-ReSample | **0.155** | **85.22** | **28.16** | 0.770 |
| | EquiCon-ReSample | 0.159 | 88.75 | 28.11 | **0.774** |
| Nonlinear deblur | ReSample | 0.188 | 56.06 | 29.54 | 0.842 |
| | Equi-ReSample | 0.128 | 55.09 | 29.45 | 0.840 |
| | EquiCon-ReSample | **0.125** | **54.62** | **29.55** | **0.843** |

supporting qualitative results). EquiReg improves performance under high measurement noise (Figure 5b). This result aligns with Figure 5a, which shows the equivariance error is lower on clean images than noisy ones, indicating that EquiReg enforces an effective denoising. Lastly, we note that EquiReg is robust to regularizing hyperparameter $\lambda_t$ (Figure 6a, see Section C for details).

Table 4: **EquiReg for diffusion models on FFHQ**. $256 \times 256$ with $\sigma_y = 0.05$.

| Method | Gaussian deblur | | | Motion deblur | | | Super-resolution ($\times 4$) | | | Box inpainting | | | Random inpainting | | |
|---|---|---|---|---|---|---|---|---|---|---|---|---|---|---|---|
| | LPIPS↓ | FID↓ | PSNR↑ | LPIPS↓ | FID↓ | PSNR↑ | LPIPS↓ | FID↓ | PSNR↑ | LPIPS↓ | FID↓ | PSNR↑ | LPIPS↓ | FID↓ | PSNR↑ |
| PSLD | 0.357 | 106.2 | 22.87 | **0.322** | **84.62** | 24.25 | 0.313 | 89.72 | 24.51 | 0.158 | 43.02 | 24.22 | 0.246 | 49.77 | 29.05 |
| Equi-PSLD | 0.344 | 94.09 | **24.42** | 0.338 | 99.14 | **24.83** | 0.289 | 90.88 | **26.32** | 0.098 | **31.54** | 24.19 | **0.188** | 41.61 | **30.43** |
| EquiCon-PSLD | **0.320** | **83.18** | 24.38 | **0.322** | 89.87 | 25.14 | **0.277** | **79.39** | 26.14 | **0.092** | 35.07 | **24.26** | 0.204 | 40.75 | 29.99 |

(a) Latent diffusion.

| Method | Gaussian deblur | | | Motion deblur | | | Super-resolution ($\times 4$) | | | Box inpainting | | | Random inpainting | | |
|---|---|---|---|---|---|---|---|---|---|---|---|---|---|---|---|
| | LPIPS↓ | FID↓ | PSNR↑ | LPIPS↓ | FID↓ | PSNR↑ | LPIPS↓ | FID↓ | PSNR↑ | LPIPS↓ | FID↓ | PSNR↑ | LPIPS↓ | FID↓ | PSNR↑ |
| DPS | 0.145 | 104.8 | 25.48 | 0.132 | 99.75 | 26.75 | 0.191 | 125.4 | 24.38 | 0.133 | 56.89 | 23.10 | 0.113 | 51.32 | 29.63 |
| Equi-DPS (ours) | **0.114** | **48.76** | **26.32** | **0.094** | **41.71** | **28.23** | **0.120** | **51.00** | **27.15** | **0.099** | **40.47** | **23.39** | **0.068** | 33.65 | **32.16** |
| DiffStateGrad-DPS | 0.128 | 52.73 | 26.29 | 0.118 | 50.14 | 27.61 | 0.186 | 73.02 | 24.65 | 0.114 | 47.53 | **24.10** | 0.107 | 49.42 | 30.15 |
| MCG | 0.340 | 101.2 | 6.72 | 0.702 | 310.5 | 6.72 | 0.520 | 87.64 | 20.05 | 0.309 | **40.11** | 19.97 | 0.286 | **29.26** | 21.57 |
| MPGD-AE | 0.150 | 114.9 | 24.42 | 0.120 | 104.5 | 25.72 | 0.168 | 137.7 | 24.01 | 0.138 | 248.7 | 21.59 | 0.172 | 339.0 | 25.22 |

(b) Pixel-based diffusion.

**Diversity analysis.** To study posterior sampling diversity of EquiReg, we generated $K = 10$ posterior samples for 20 test images across three inverse problems of box inpainting, Gaussian deblurring, $4\times$ super-resolution, and measured diversity using two complementary metrics: Intra-LPIPS for perceptual diversity and Pixel-Std for spatial diversity. Table 11 demonstrates that Equi-DPS achieves favorable fidelity-diversity trade-offs. We further investigated diversity scaling by varying box inpainting mask size from $128 \times 128$ to $192 \times 192$ pixels (Figure 18). Results show that diversity metrics increase linearly with task difficulty, demonstrating that Equi-DPS naturally expands sampling as problems become more ill-posed rather than artificially constraining solutions. This linear relationship indicates healthy, predictable posterior sampling behavior across the difficulty spectrum. Lastly, Figure 7 provides qualitative confirmation through visual examples showing four posterior samples per image. Observable variations in facial features, expressions, and eye gaze validate our quantitative measurements, confirming EquiReg can generate genuinely diverse reconstructions rather than collapsing to a single solution.

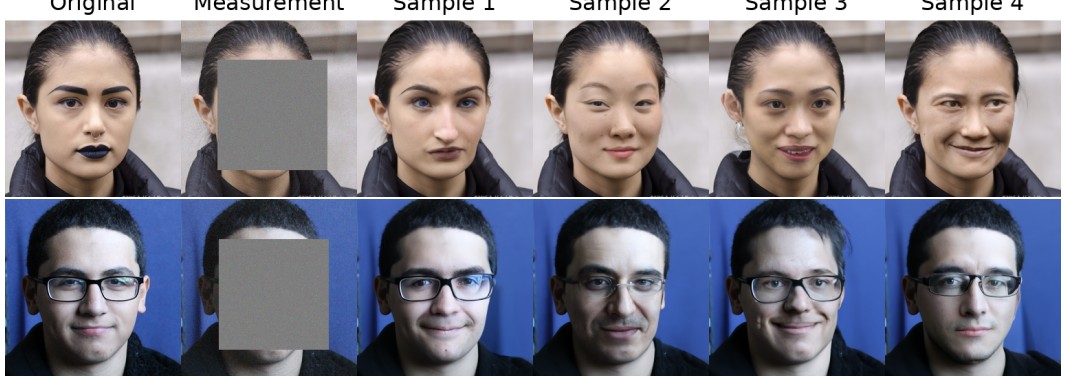

Figure 7: **Qualitative diversity examples for box inpainting.** We show $K = 4$ posterior samples for two test images with $160 \times 160$ masks. Each sample exhibits perceptually distinct facial features (expressions, eye gaze, facial structure) while maintaining high fidelity to the ground truth, demonstrating EquiReg generates diverse plausible reconstructions rather than collapsing to a single mode.

**Solving PDEs from sparse observations.** EquiReg is evaluated on two important PDE problems: the Helmholtz and Navier-Stokes equations (see Section F). The objective is to solve both forward and inverse problems in sparse sensor settings. The forward problem involves predicting the solution function or the final state using measurements from only $3\%$ of the coefficient field or the initial state. The inverse problem, conversely, aims to predict the input conditions from observations of $3\%$ of the system's output. This task is challenging due to the nonlinearity of the equations, the complex structure of Gaussian random fields, and the sparsity of observations.

Table 5: **EquiReg for latent diffusion models on ImageNet.** $256 \times 256$ with $\sigma_{\boldsymbol{y}} = 0.05$.

| Method | Gaussian deblur | | Motion deblur | | Super-resolution (x4) | | Box inpainting | | Random inpainting | |
|---|---|---|---|---|---|---|---|---|---|---|
| | FID↓ | PSNR↑ | FID↓ | PSNR↑ | FID↓ | PSNR↑ | FID↓ | PSNR↑ | FID↓ | PSNR↑ |
| PSLD | 263.9 | 20.70 | 252.1 | 21.26 | 224.3 | 22.29 | 151.4 | 16.28 | 83.22 | 26.56 |
| EquiCon-PSLD | **214.5** | **22.01** | **196.3** | **22.69** | **198.5** | **22.34** | **137.6** | **19.25** | **65.14** | **27.03** |

Recent studies (Huang et al., 2024; Mammadov et al., 2024a; Yao et al., 2025) have demonstrated the superiority of diffusion models over deterministic single-forward methods for solving PDEs. DiffusionPDE (Huang et al., 2024) decomposes the conditional log-likelihood into a learned diffusion prior and a measurement score. FunDPS (Yao et al., 2025) extends the sampling process to a more natural infinite-dimensional spaces, achieving better accuracy and speed via function space models.

We integrate EquiReg into the state-of-the-art FunDPS framework (Mammadov et al., 2024a; Yao et al., 2025), where we compute the Equi loss with respect to equivariance learned by an FNO trained on the corresponding inverse problem. We use reflection symmetry (i.e., flipping along the $y = x$ axis), and observe no

Table 6: **Solving PDEs from sparse observations.**

| Method | Steps ($N$) | Helmholtz | | Navier-Stokes | |
|---|---|---|---|---|---|
| | | Forward | Inverse | Forward | Inverse |
| DiffusionPDE | 2000 | 12.64% | 19.07% | 3.78% | 9.63% |
| FunDPS | 500 | 2.13% | 17.16% | 3.32% | 8.48% |
| Equi-FunDPS (ours) | 500 | **2.12%** | **15.91%** | **3.06%** | **7.84%** |

significant performance difference when using other transformations such as rotations or alternating flips. Equi-FunDPS improves performance (Table 6), measured by relative $\ell_2$ loss, across various tasks, especially in inverse problems where a strong data prior is critical.

**Text-to-image guidance.** Given the "source" image, DreamSampler (Kim et al., 2024) transforms the source image using the prompt. Applying EquiReg to DreamSampler, we observe perceptual improvement of generated images as well as artifact reduction. Figure 1 shows the "source" cat, being transformed into the prompt (e.g., "corgi"). Equi-DreamSampler generates more realistic images than DreamSampler. Notably, EquiReg resolves the three-front-legged corgi into a two-front-legged one (for an implicit acceleration of image generation when EquiReg is imposed, see Section A).

## 5 CONCLUSION

We introduce *Equivariance Regularized* (EquiReg) diffusion for inverse problems. EquiReg regularizes sampling trajectories to stay closer to the data manifold, leveraging manifold-preferential equivariance (MPE): functions with low equivariance error on-manifold and high error off-manifold. Such functions arise naturally in trained networks and can serve as plug-and-play regularizers without modifying the diffusion denoiser. EquiReg is agnostic across pixel- and latent-space diffusion models and remains robust under reduced sampling, effectively accelerating convergence. Across diverse inverse problems, it consistently improves perceptual and reconstruction metrics while reducing failure cases, highlighting its generality and efficiency.

**Limitations and future work.** EquiReg's effectiveness depends on the quality of the pre-trained backbone diffusion. EquiReg is a plug-and-play regularization framework that can be applied to a variety of guidance-based diffusion models; thus, it does not directly address the approximations of the underlying diffusion models, but instead regularizes for improved performance. Also, since EquiReg is a regularization mechanism, it improves performance precisely in regimes where baseline methods degrade or fail. Hence, one cannot expect EquiReg to improve the performance of a diffusion model beyond the capability of a regularization framework. Finally, applying EquiReg requires task-specific design choices: selecting an appropriate symmetry group and identifying suitable MPE functions for the problem at hand. While we presented two systematic approaches to construct MPE functions for imaging and PDEs, the process of identifying MPE functions varies across applications and represents an important area for methodological development, which we have provide guidelines for in this paper. This task-specific design also makes EquiReg broadly adaptable across diverse domains beyond the considered applications. Finally, while our paper formalizes distribution-dependent equivariant functions and MPE functions, a full theoretical characterization of the conditions under which MPE properties emerge in trained networks or its joint training with diffusion remains an important and valuable direction for future work.

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

# APPENDICES FOR "EQUIREG: EQUIVARIANCE REGULARIZED DIFFUSION FOR INVERSE PROBLEMS"

We provide our source code when EquiReg. We will provide a publicly available source code upon acceptance. This supplementary materials contain the following:

- Section A includes additional experiments on text-to-image guidance. We regularize DreamSampler (Kim et al., 2024) with EquiReg for an improved performance (see Figures 8 to 12).

- Section B includes additional experiments on robustness including robustness to $\lambda_t$, reduced number of DDIM steps, and reduced number of measurent consistency steps.

- Section C includes qualitative analysis on the performance of methods with and without EquiReg. Results show a reduction of artifacts and an improved perceptual quality of the solution. This section also includes the equivariance error of a pre-trained encoder used in EquiReg (Figure 15a) and a histogram of Equi's improvement on DPS (Figure 14).

- Section D includes diversity experiments. Results show that EquiReg achieves favorable fidelity-diversity tradeoffs (Table 11, Figure 18, and Figure 19).

- Section E demonstrates EquiReg experimental setup and implementation for PSLD, ReSample, and DPS (Algorithms 2 to 6). It also contains information about the EquiReg hyperparameters for image restoration tasks.

- Section F contains information on the PDE reconstruction experiment. It discusses the equations along with implementation details and hyperparameters.

- Section G provides theoretical proofs of Propositions G.1 and G.2.

- Section H contains additional background information on solving inverse problems, vanishing-error autoencoders, and equivariance.

- Section I provides additional experiment on MPE functions.

- Section J discloses computing resources used to conduct the experiments.

- Section K credits code assets used for our experiments.

- Section L discusses the broader impacts of this paper, the developed method, the conducted experiments, and their overall implications.

- Section M concludes the appendix with a "responsible release" statement.

The authors acknowledge the usage of LLMs on proofreading of the manuscript. The authors have not used LLMs for content generation.

## A EQUIREG FOR TEXT-TO-IMAGE GUIDANCE

Given the "source" image, DreamSampler (Kim et al., 2024) is asked to transform the source image using the prompt. Applying EquiReg to DreamSampler, we observe perceptual improvement of generated images as well as artifact reduction.

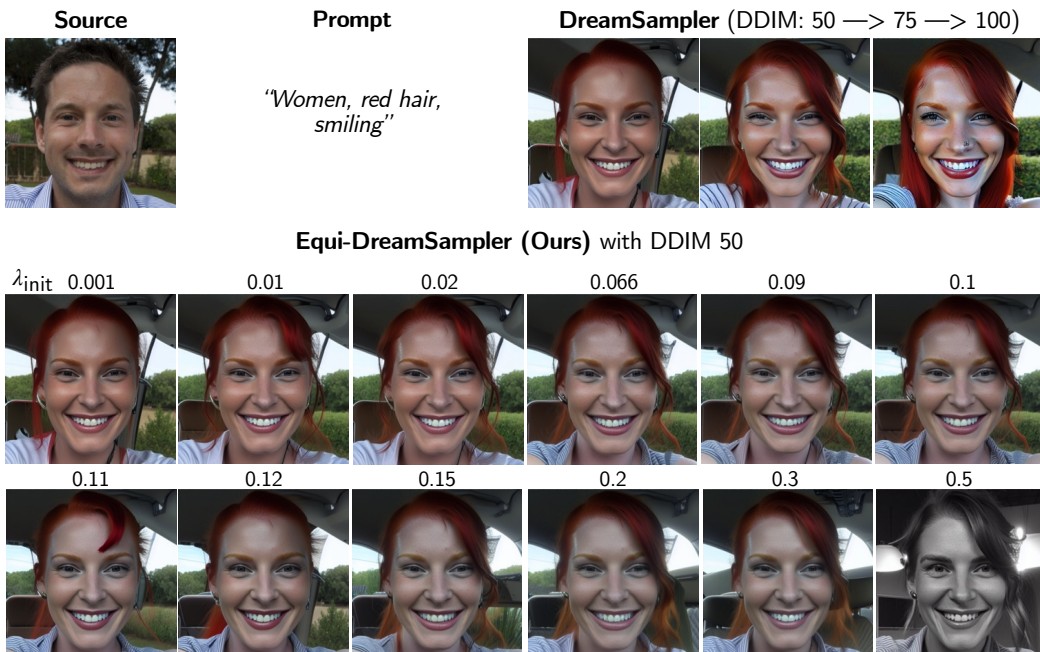

Figure 8: **Impact of EquiReg parameter $\lambda_t$, implicit acceleration, and introduction of more image details.** Women, red hair, smiling.

We have observed an implicit acceleration of image generation when EquiReg is imposed (Figure 8). Equi-DreamSampler with 50 DDIM steps can generate images that are only possible with DreamSampler when the DDIM steps are increased. We attribute this to EquiReg's ability to generate images that are closer to the data manifold. For example, the increase of DDIM steps in DreamSampler (from 50 to 75 to 100) has a relatively similar effect to increasing the EquiReg regularizer $\lambda_t$ at a fixed 50 DDIM steps. Figure 8 shows that increasing the regularization $\lambda_t$ results in addition of a car in the background. For DreamSampler, an early notion of the car seat in the background start to arise only when DDIM is increased to 100 (see also Figures 9 to 12).

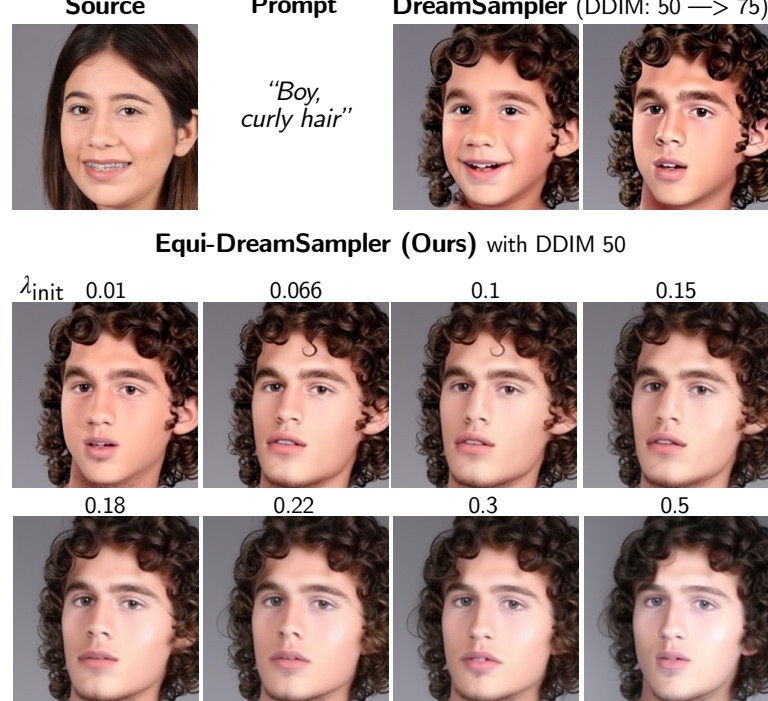

Figure 9: **Adding EquiReg into the text-to-image guidance method DreamSampler for improved performance.** Boy, curly hair.

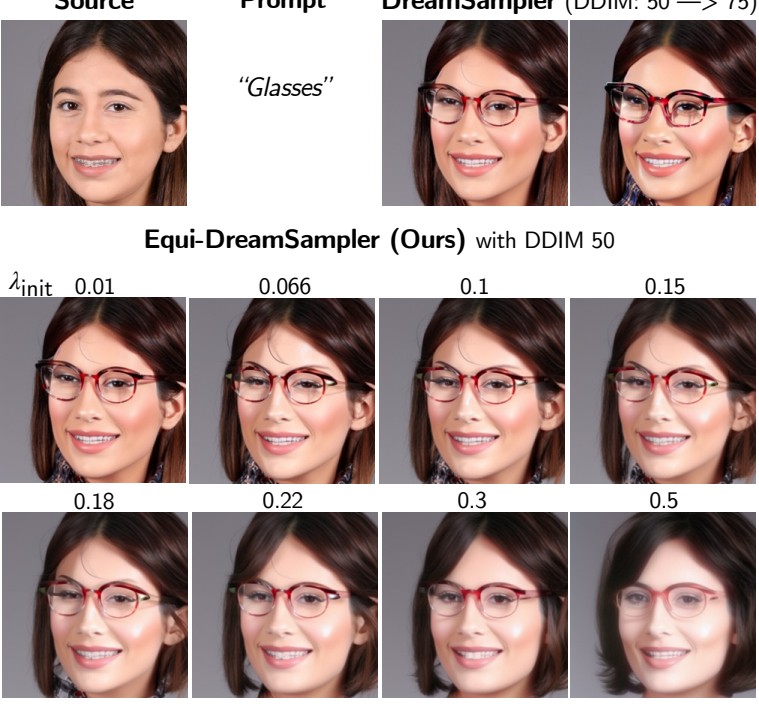

Figure 10: **Adding EquiReg into the text-to-image guidance method DreamSampler for improved performance.** Glasses.

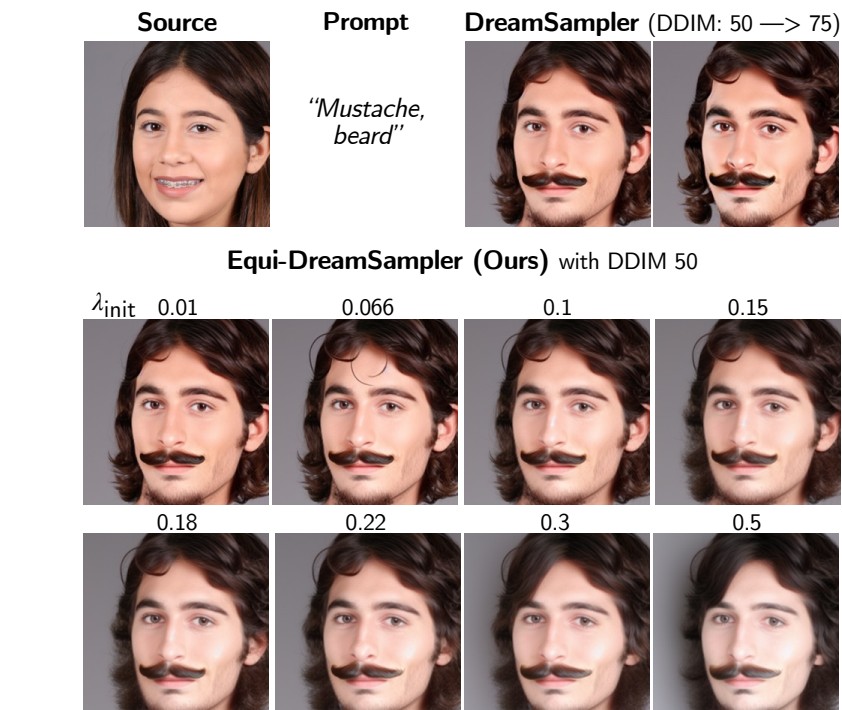

Figure 11: **Adding EquiReg into the text-to-image guidance method DreamSampler for improved performance.** Mustache, beard.

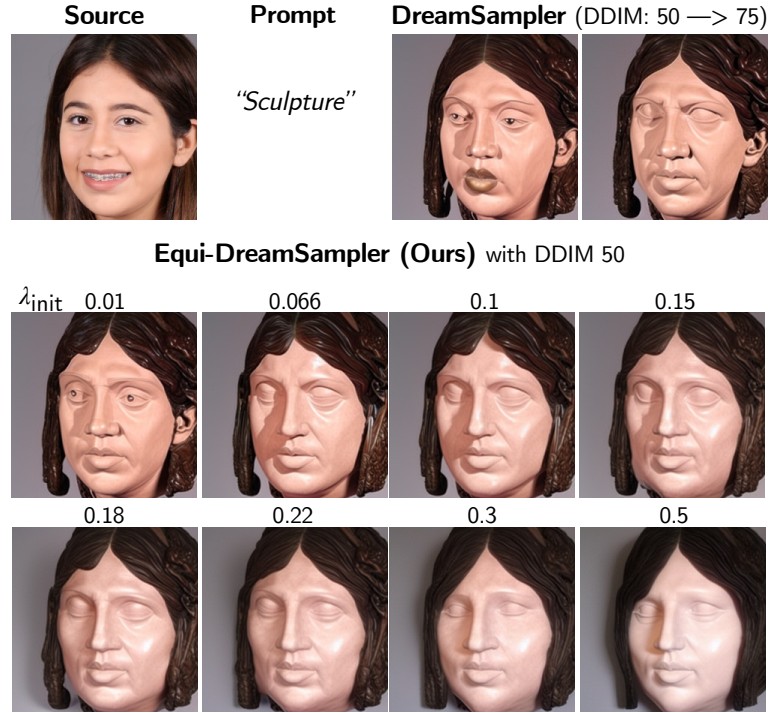

Figure 12: **Adding EquiReg into the text-to-image guidance method DreamSampler for improved performance.** Sculpture.

## B   ADDITIONAL EXPERIMENTS ON ROBUSTNESS

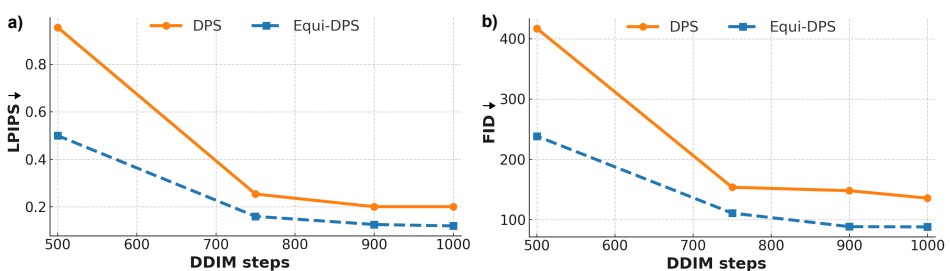

Figure 13: **Advantages of EquiReg under reduced DDIM steps.** Super-resolution on FFHQ.

Table 7: **Robustness to $\lambda_t$.** Sensitivity analysis for DPS and PSLD across different values of $\lambda_t$.

|  | **DPS** | | | **PSLD** | | |
|---|---|---|---|---|---|---|
| $\lambda_t$ | PSNR↑ | SSIM↑ | LPIPS↓ | PSNR↑ | SSIM↑ | LPIPS↓ |
| 0.0 | 24.34 (1.03) | 0.664 (0.061) | 0.156 (0.051) | 23.83 (2.61) | 0.63 (0.12) | 0.315 (0.07) |
| 0.001 | 25.44 (1.22) | 0.708 (0.057) | 0.118 (0.038) | – | – | – |
| 0.01 | 25.44 (1.22) | 0.708 (0.057) | 0.118 (0.038) | 25.35 (2.24) | 0.70 (0.09) | 0.280 (0.07) |
| 0.1 | 25.44 (1.22) | 0.708 (0.057) | 0.118 (0.038) | 26.63 (1.68) | 0.74 (0.08) | 0.337 (0.06) |
| 0.25 | – | – | – | 26.22 (1.57) | 0.72 (0.08) | 0.366 (0.05) |
| 1.0 | 25.44 (1.22) | 0.709 (0.057) | 0.118 (0.038) | 24.74 (1.28) | 0.66 (0.07) | 0.438 (0.05) |

Table 8: **EquiReg improves performance under reduced DDIM steps.** Pixel-based super-resolution on FFHQ $256 \times 256$.

|  | **DPS** | | | | **Equi-DPS** (ours) | | | |
|---|---|---|---|---|---|---|---|---|
| Steps | PSNR↑ | SSIM↑ | LPIPS↓ | FID↓ | PSNR↑ | SSIM↑ | LPIPS↓ | FID↓ |
| 500 | 13.89 | 0.0937 | 0.955 | 417.07 | 20.61 | 0.366 | 0.500 | 238.51 |
| 750 | 21.77 | 0.540 | 0.254 | 153.74 | 25.60 | 0.704 | 0.160 | 110.89 |
| 900 | 22.97 | 0.628 | 0.201 | 148.03 | 26.52 | 0.755 | 0.126 | 88.46 |
| 1000 | 22.99 | 0.649 | 0.201 | 135.71 | 26.73 | 0.767 | 0.120 | 88.00 |

Table 9: **EquiReg improves SITCOM under reduced measurement consistency steps ($K_{\text{meas}}$).** Motion deblur on FFHQ sampled with 50 DDIM steps.

| $K_{\text{meas.}}$ | $K_{\text{EquiReg}}$ | PSNR↑ | SSIM↑ | Runtime (s) |
|---|---|---|---|---|
| 10 | N/A | 28.06 | 0.81 | 21.57 |
| 10 | 1 | 28.71 | 0.82 | 21.07 |
| 5 | 5 | **29.26** | **0.83** | **11.09** |
| 20 | N/A | 27.04 | 0.79 | 38.85 |
| 20 | 1 | 28.54 | **0.82** | 37.74 |
| 10 | 10 | **28.93** | **0.82** | **20.92** |
| 30 | N/A | 27.79 | 0.80 | 58.84 |
| 30 | 1 | 28.35 | 0.81 | 55.51 |
| 15 | 15 | **29.63** | **0.84** | **30.19** |
| 40 | N/A | 30.40 | **0.85** | 78.08 |
| 40 | 1 | **30.58** | **0.85** | 69.83 |
| 20 | 20 | 29.50 | 0.83 | **41.02** |
| 60 | N/A | 28.35 | 0.81 | 108.57 |
| 60 | 1 | 27.02 | 0.78 | 95.62 |
| 30 | 30 | **31.36** | **0.87** | **59.38** |

Table 10: **EquiReg Effectiveness with Subset of Group Actions.**

| **PSLD** | | **Equi-PSLD** (90, 270 deg) | |
|---|---|---|---|
| PSNR↑ | SSIM↑ | PSNR↑ | SSIM↑ |
| 15.86 (1.19) | 0.77 (0.03) | 17.60 (1.60) | 0.79 (0.03) |

# C  VISUALIZATIONS FOR IMAGE RESTORATION EXPERIMENTS

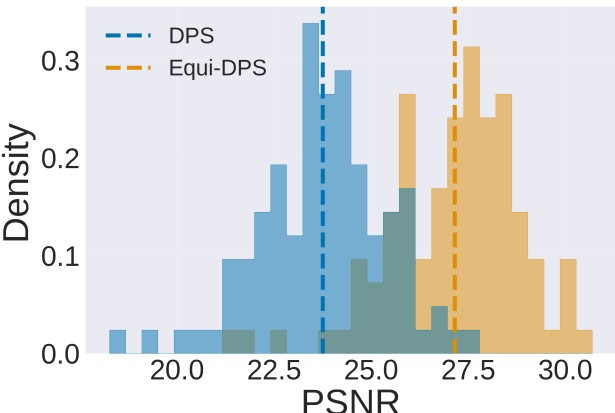

Figure 14: **Histogram of EquiReg improvement for DPS.** Super-resolution using FFHQ $256 \times 256$.

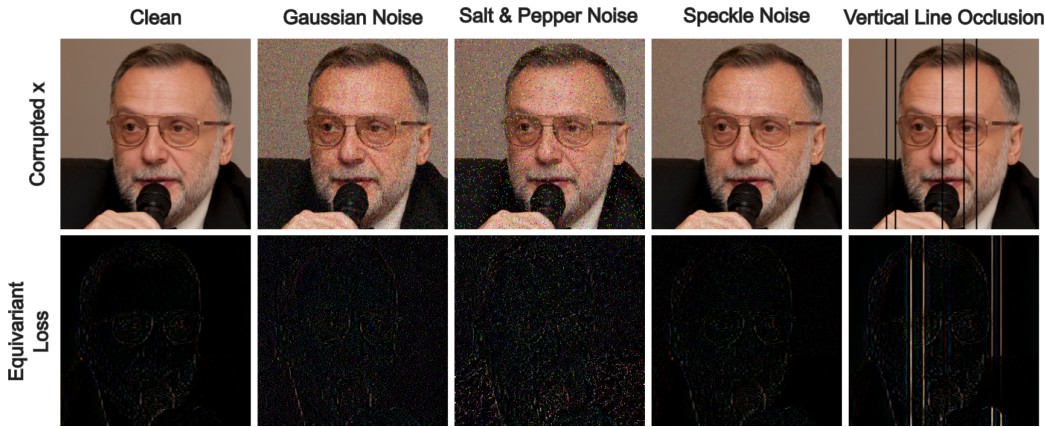

(a) The equivariance error of the encoder is lower on clean, natural images than corrupted ones.

(b) Example visualizations of used images and corresponding equivariance error computed using the decoder (see Figure 4a).

Figure 15: **Training induced equivariance for a pre-trained function.**

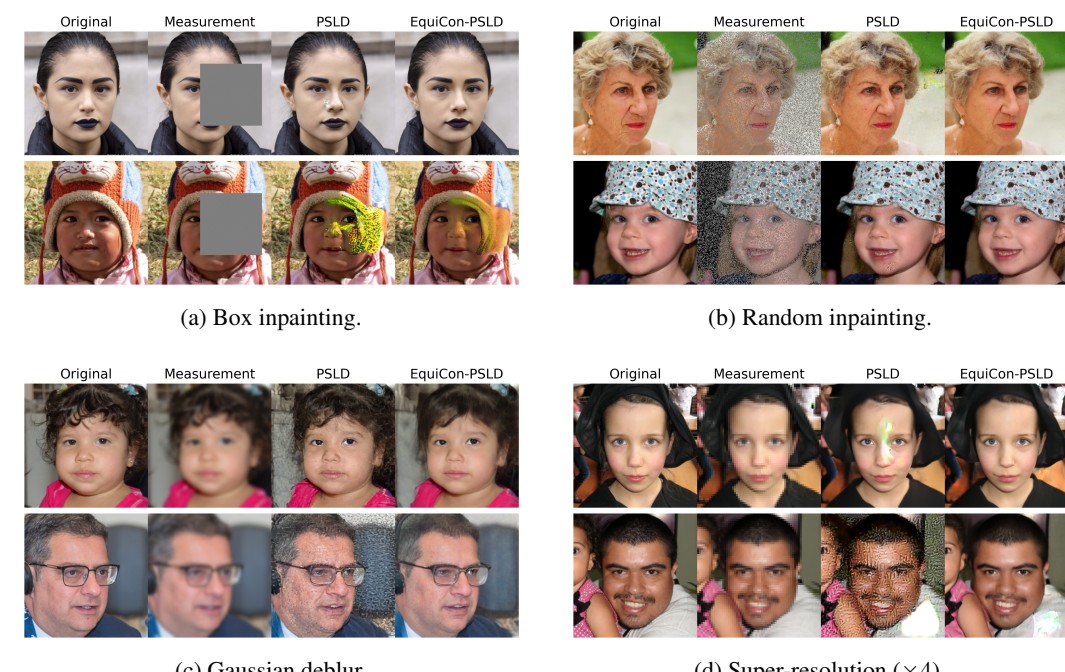

(a) Box inpainting.

(b) Random inpainting.

(c) Gaussian deblur.

(d) Super-resolution (×4).

Figure 16: **Qualitative comparison of EquiCon-PSLD and PSLD on FFHQ 256 × 256.**

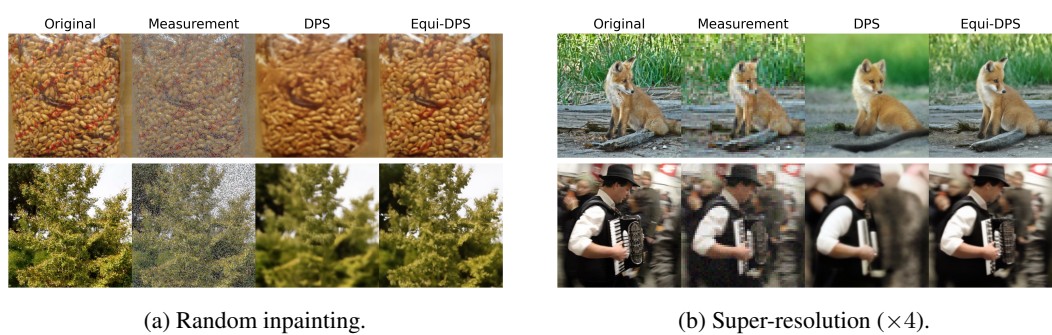

(a) Random inpainting.

(b) Super-resolution (×4).

Figure 17: **Qualitative comparison of Equi-DPS and DPS on ImageNet 256 × 256.**

## D    DIVERSITY ANALYSIS

In the Bayesian setting, the objective of solving inverse problems with diffusion models is to sample from high-probability regions of the posterior distribution. While the goal is not to maximize "diversity", the true diversity emerges when the posterior admits meaningful variability. In practice, diversity-related concerns in inverse problems arise when a method suffers from mode collapse, i.e., the sampler becomes biased and fails to explore multiple plausible modes of the posterior. Thus, the relevant question is whether a method properly explores the posterior rather than whether it maximizes diversity in an unconstrained sense.

Because closed-form posteriors are unavailable for real image restoration tasks, the standard practice in the diffusion inverse-problem literature is to evaluate diversity through variation among plausible reconstructions consistent with the measurement, without collapsing to a single solution. This is the notion of "diversity" our work adopts.

Given the goal of posterior sampling, EquiReg is not designed to maximize diversity for its own sake. Its objective is to incorporate data-inherent geometric structure (equivariance) to guide sampling toward high-probability regions of the posterior. Hence, diversity arises naturally from the ill-posedness of the inverse problem; it is a consequence of posterior uncertainty, not the goal of the regularizer.

To quantify this effect, in addition to reconstruction quality, we analyzed the diversity of posterior samples produced by EquiReg. We evaluate diversity metrics across multiple tasks and difficulty levels to characterize the sampling behavior of our method.

## D.1 EXPERIMENTAL SETUP

To evaluate diversity, we generate multiple posterior samples and measure variation across these samples. For each of 20 test images, we generate K=10 reconstructions using different random seeds. We evaluate diversity using two complementary metrics: Intra-LPIPS, which measures perceptual diversity by computing the average LPIPS distance between all pairs of samples, and Pixel-Std, which measures spatial diversity through pixel-wise standard deviation across samples. Higher values for both metrics indicate greater diversity. For Intra-LPIPS, we compute distances for all $\binom{K}{2} = 45$ pairs per image and average across all test images. For Pixel-Std, we compute the standard deviation at each pixel location across the K samples, then average across all pixels and test images. We evaluate diversity across three inverse problems (box inpainting, Gaussian deblurring, and $4\times$ super-resolution) comparing EquiReg against DPS (Chung et al., 2023) without equivariance regularization. To investigate how diversity scales with task difficulty, we additionally vary the inpainting mask size from $128 \times 128$ (standard) to $160 \times 160$ to $192 \times 192$ pixels.

## D.2 RESULTS AND DISCUSSION

Table 11 shows that Equi-DPS achieves favorable fidelity-diversity trade-offs across three inverse problems. For box inpainting and super-resolution, equivariance regularization improves both fidelity and diversity simultaneously. For Gaussian deblurring, Equi-DPS achieves 15-20% better fidelity while retaining 80-85% of baseline diversity, representing a modest but justified trade-off. These results demonstrate that equivariance constraints do not inherently suppress diversity; rather, they can guide sampling toward regions of higher data fidelity while maintaining posterior exploration.

Table 11: **Fidelity and diversity comparison across inverse problems.** Evaluated on 20 test images with $K = 10$ samples per image. Equi-DPS improves fidelity while largely preserving or enhancing sampling diversity.

| Task | Method | Fidelity Metrics | | Diversity Metrics | |
|------|--------|------------------|------|-------------------|-----------|
| | | LPIPS↓ | FID↓ | Intra-LPIPS↑ | Pixel-Std↑ |
| Box inpainting | DPS | 0.140 | 70.89 | 0.112 | 9.286 |
| | Equi-DPS (ours) | **0.112** | **59.70** | **0.118** | **10.59** |
| Gaussian deblur | DPS | 0.150 | 76.71 | **0.114** | **6.565** |
| | Equi-DPS (ours) | **0.120** | **63.02** | 0.092 | 5.669 |
| Super-resolution ($\times 4$) | DPS | 0.683 | 99.11 | 0.134 | 7.956 |
| | Equi-DPS (ours) | **0.703** | **87.52** | **0.187** | **23.52** |

Figure 18 reveals linear diversity scaling with task difficulty. Diversity metrics grow proportionally with task difficulty, indicating Equi-DPS naturally expands sampling as problems become more ill-posed. This linear relationship demonstrates stable, predictable behavior across difficulty levels without artificial diversity suppression.

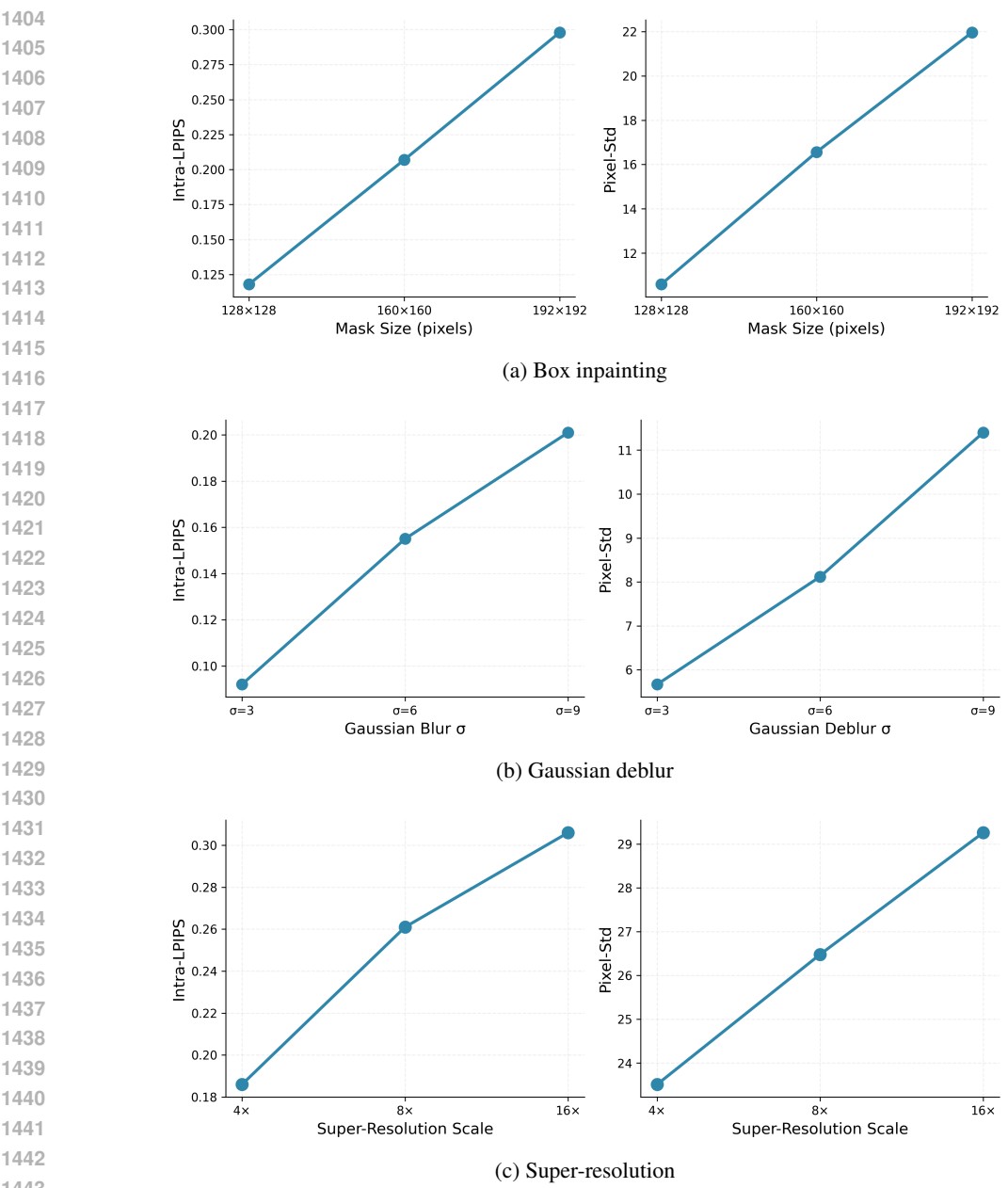

(a) Box inpainting

(b) Gaussian deblur

(c) Super-resolution

Figure 18: **Diversity vs task difficulty across three inverse problems.** As task difficulty increases (larger inpainting mask, stronger blur, higher SR scale), both diversity metrics increase proportionally, demonstrating that Equi-DPS maintains healthy posterior sampling behavior across a wide difficulty spectrum.

Figures 7 and 19 provide qualitative results.

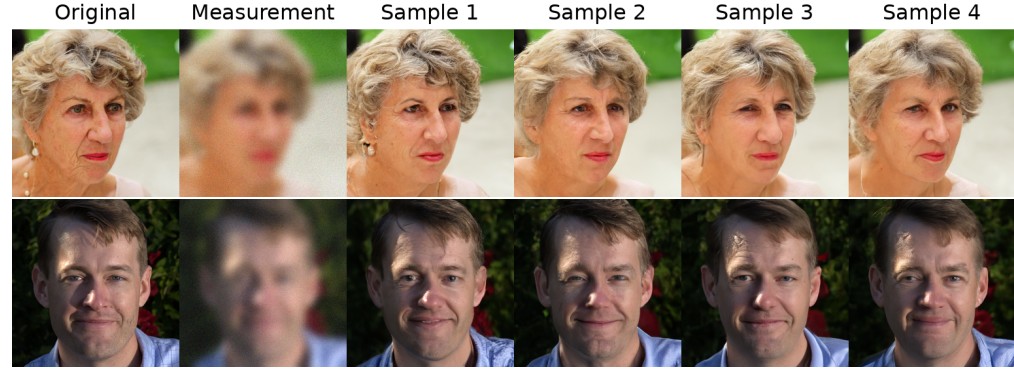

(a) Gaussian deblur

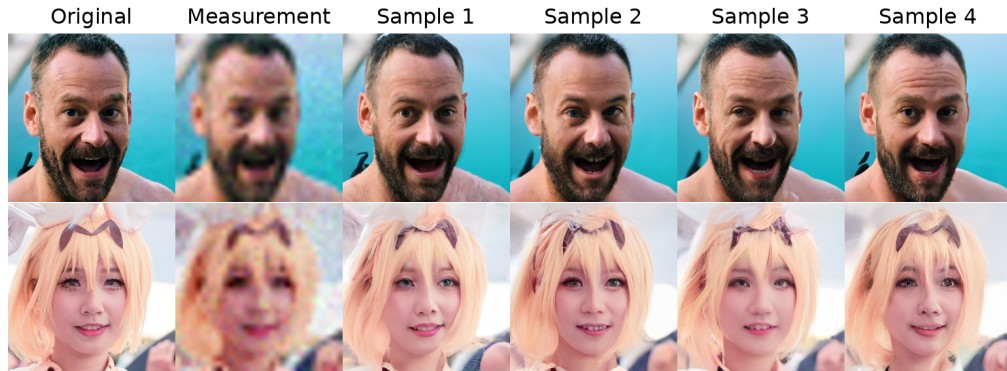

(b) Super-resolution

Figure 19: **Qualitative diversity examples across Gaussian deblur and super-resolution.** Each subfigure shows $K = 4$ posterior samples for two different test images. (a) Gaussian deblur: samples differ in facial expressions and accessories (i.e., earrings in first test image). (b) Super-resolution: samples differ in facial features (i.e., teeth in the first test image, eye color and eyelashes in the second test image). Across both tasks, EquiReg produces diverse plausible reconstructions rather than collapsing to a single mode.

## D.3 CONCLUSION

Finally, we highlight that EquiReg improves both fidelity and diversity on 2 of the 3 considered tasks, an encouraging outcome that is uncommon given the general behavior of classical regularizers. Hand-crafted regularizers such as TV and $\ell_1$ may suppress diversity by shrinking solutions toward simple structures. By contrast, EquiReg leverages data-dependent regularization that captures the richness and structural complexity of the underlying data manifold, enabling it to preserve manifold-consistent variability while suppressing implausible samples.

High diversity without fidelity is not meaningful for posterior sampling. A method that samples the entire solution space, including low-probability and artifacted regions, may score well on diversity but fail to provide useful reconstructions. Equi-DPS avoids this failure mode: it maintains meaningful diversity while reducing artifacts and improving perceptual quality. In the experiments conducted during the rebuttal, our goal was to demonstrate clearly that EquiReg preserves meaningful diversity, reflecting the posterior uncertainty, rather than unstructured or unconstrained variability.

## E IMPLEMENTATION DETAILS FOR IMAGE RESTORATION TASKS

**Experimental Setup.** We evaluate EquiReg on a variety of linear and nonlinear restoration tasks for natural images. We fix sets of 100 images from FFHQ and ImageNet as our validation sets. All images are normalized from $[0, 1]$. For the majority of experiments, we use noise level $\sigma_{\boldsymbol{y}} = 0.05$

(we indicate $\sigma_{\boldsymbol{y}}$ in our tables). For linear inverse problems, we consider (1) box inpainting, (2) random inpainting, (3) Gaussian deblur, (4) motion deblur, and (5) super-resolution. We apply a random $128 \times 128$ pixel box for box inpainting, and a $70\%$ random mask for random inpainting. For Gaussian and motion deblur, we use kernels of size $61 \times 61$, with standard deviations of 3.0 and 0.5, respectively. For super-resolution, we downscale images by a factor of 4 using a bicubic resizer. For nonlinear inverse problems, we consider (1) phase retrieval, (2) nonlinear deblur, and (3) high dynamic range (HDR). We use an oversampling rate of 2.0 for phase retrieval, and due to instability of the task, we generate four independent reconstructions and take the best result (as also done in DPS (Chung et al., 2023), DAPS (Zhang et al., 2025a), and DiffStateGrad (Zirvi et al., 2025)). We use the default setting from (Tran et al., 2021) for nonlinear deblur, and a scale factor of 2 for HDR.

**Hyperparameters.** Our method introduces a single hyperparameter $\lambda_t$ that controls the amount of regularization applied. Below we include a table detailing the use of this hyperparameter in the main experiments (Table 12). For majority of experiments, we keep $\lambda_t$ constant throughout iterations. For all unscaled experiments, we employ early stopping, setting $\lambda_t = 0$ for the last 10% of sampling.

Table 12: Equivariance regularization weight $\lambda_t$ used in main experiments.

| Method | Box Inpainting | Random Inpainting | Gaussian Deblur | Motion Deblur | Super-resolution ($\times 4$) |
|---|---|---|---|---|---|
| *FFHQ* $256 \times 256$ | | | | | |
| Equi-PSLD | 0.05 | 0.05 | 0.03 | 0.03 | 0.02 |
| EquiCon-PSLD | 0.01 | 0.01 | 0.01 | 0.01 | 0.01 |
| Equi-ReSample | 0.03 | 0.05 | 0.02 | 0.02 | 0.05 |
| EquiCon-ReSample | 0.001 | 0.001 | 0.001 | 0.001 | 0.001 |
| Equi-DPS | 0.0001 | 0.001 | 0.001 | 0.001 | 0.1 |
| *ImageNet* $256 \times 256$ | | | | | |
| EquiCon-PSLD | 0.0015 | 0.05 | 0.06 | 0.07 | 0.001 |

**PSLD.** We integrate EquiReg into PSLD by simply adding an additional gradient update step using our regularization term (Algorithms 2 and 3).

In our experiments, we use the official PSLD implementation from Rout et al. (2023), running with its default settings to reproduce the baseline results. We note that in our code, we do not square the norm when computing the gradient, aligning with PSLD's implementation.

---

**Algorithm 2** Equi-PSLD for Image Restoration Tasks

---

**Require:** $T, \boldsymbol{y}, \{\eta_t\}_{t=1}^T, \{\gamma_t\}_{t=1}^T, \{\tilde{\sigma}_t\}_{t=1}^T$
**Require:** $\mathcal{E}, \mathcal{D}, \mathcal{A}\boldsymbol{x}_0^*, \mathcal{A}, \boldsymbol{s}_\theta, T_g \text{ and } S_g, \{\lambda_t\}_{t=1}^T$
1: $\boldsymbol{z}_T \sim \mathcal{N}(\mathbf{0}, \boldsymbol{I})$
2: **for** $t = T - 1$ **to** $0$ **do**
3: $\quad \hat{\boldsymbol{s}} \leftarrow \boldsymbol{s}_\theta(\boldsymbol{z}_t, t)$
4: $\quad \boldsymbol{z}_{0|t} \leftarrow \frac{1}{\sqrt{\bar{\alpha}_t}}(\boldsymbol{z}_t + (1 - \bar{\alpha}_t)\hat{\boldsymbol{s}})$
5: $\quad \boldsymbol{\epsilon} \sim \mathcal{N}(\mathbf{0}, \boldsymbol{I})$
6: $\quad \boldsymbol{z}_{t-1}' \leftarrow \frac{\sqrt{\alpha_t}(1-\bar{\alpha}_{t-1})}{1-\bar{\alpha}_t}\boldsymbol{z}_t + \frac{\sqrt{\bar{\alpha}_{t-1}}\beta_t}{1-\bar{\alpha}_t}\boldsymbol{z}_{0|t} + \tilde{\sigma}_t\boldsymbol{\epsilon}$
7: $\quad \boldsymbol{z}_{t-1}'' \leftarrow \boldsymbol{z}_{t-1}' - \eta_t\nabla_{\boldsymbol{z}_t}\|\boldsymbol{y} - \mathcal{A}(\mathcal{D}(\boldsymbol{z}_{0|t}))\|_2^2$
8: $\quad \boldsymbol{z}_{t-1} \leftarrow \boldsymbol{z}_{t-1}'' - \gamma_t\nabla_{\boldsymbol{z}_t}\|\boldsymbol{z}_{0|t} - \mathcal{E}(\mathcal{A}^T\mathcal{A}\boldsymbol{x}_0^* + (\boldsymbol{I} - \mathcal{A}^T\mathcal{A})\mathcal{D}(\boldsymbol{z}_{0|t}))\|_2^2$
9: $\quad \boldsymbol{z}_{t-1} \leftarrow \boldsymbol{z}_{t-1} - \lambda_t\nabla_{\boldsymbol{z}_t}\|S_g(\mathcal{D}(\boldsymbol{z}_{0|t})) - \mathcal{D}(T_g(\boldsymbol{z}_{0|t}))\|_2^2$
10: **end for**
11: **return** $\mathcal{D}(\boldsymbol{z}_{0|t})$

---

**Algorithm 3** EquiCon-PSLD for Image Restoration Tasks

**Require:** $T, \boldsymbol{y}, \{\eta_t\}_{t=1}^T, \{\gamma_t\}_{t=1}^T, \{\tilde{\sigma}_t\}_{t=1}^T$
**Require:** $\mathcal{E}, \mathcal{D}, \mathcal{A}\boldsymbol{x}_0^*, \mathcal{A}, \boldsymbol{s}_\theta, T_g \text{ and } S_g, \{\lambda_t\}_{t=1}^T$
1: $\boldsymbol{z}_T \sim \mathcal{N}(\mathbf{0}, \boldsymbol{I})$
2: **for** $t = T - 1$ **to** $0$ **do**
3:     $\hat{\boldsymbol{s}} \leftarrow \boldsymbol{s}_\theta(\boldsymbol{z}_t, t)$
4:     $\boldsymbol{z}_{0|t} \leftarrow \frac{1}{\sqrt{\bar{\alpha}_t}}(\boldsymbol{z}_t + (1 - \bar{\alpha}_t)\hat{\boldsymbol{s}})$
5:     $\boldsymbol{\epsilon} \sim \mathcal{N}(\mathbf{0}, \boldsymbol{I})$
6:     $\boldsymbol{z}_{t-1}' \leftarrow \frac{\sqrt{\bar{\alpha}_t}(1 - \bar{\alpha}_{t-1})}{1 - \bar{\alpha}_t}\boldsymbol{z}_t + \frac{\sqrt{\bar{\alpha}_{t-1}}\beta_t}{1 - \bar{\alpha}_t}\boldsymbol{z}_{0|t} + \tilde{\sigma}_t\boldsymbol{\epsilon}$
7:     $\boldsymbol{z}_{t-1}'' \leftarrow \boldsymbol{z}_{t-1}' - \eta_t\nabla_{\boldsymbol{z}_t}\|\boldsymbol{y} - \mathcal{A}(\mathcal{D}(\boldsymbol{z}_{0|t}))\|_2^2$
8:     $\boldsymbol{z}_{t-1} \leftarrow \boldsymbol{z}_{t-1}'' - \gamma_t\nabla_{\boldsymbol{z}_t}\|\boldsymbol{z}_{0|t} - \mathcal{E}(\mathcal{A}^T\mathcal{A}\boldsymbol{x}_0^* + (\boldsymbol{I} - \mathcal{A}^T\mathcal{A})\mathcal{D}(\boldsymbol{z}_{0|t}))\|_2^2$
9:     $\boldsymbol{z}_{t-1} \leftarrow \boldsymbol{z}_{t-1} - \lambda_t\nabla_{\boldsymbol{z}_t}\|\boldsymbol{z}_{0|t} - \mathcal{E}(S_g^{-1}(\mathcal{D}(T_g(\boldsymbol{z}_{0|t}))))\|_2^2$
10: **end for**
11: **return** $\mathcal{D}(\boldsymbol{z}_{0|t})$

**ReSample.** We integrate EquiReg into ReSample by adding our regularization term into the hard data consistency step (Algorithms 4 and 5). We note that the ReSample algorithm employs a two-stage approach; initially, it performs pixel-space optimization, and later it performs latent-space optimization. We apply EquiReg in the latent-space optimization stage.

In our experiments, we use the official ReSample implementation from Song et al. (2023a), running with its default settings to reproduce the baseline results.

**Algorithm 4** Equi-ReSample for Image Restoration Tasks

**Require:** Measurements $\boldsymbol{y}$, $\mathcal{A}(\cdot)$, Encoder $\mathcal{E}(\cdot)$, Decoder $\mathcal{D}(\cdot)$, Score function $\boldsymbol{s}_\theta(\cdot, t)$, Pretrained LDM Parameters $\beta_t, \bar{\alpha}_t, \eta, \delta$, Hyperparameter $\gamma$ to control $\sigma_t^2$, Time steps to perform resample $C$, $T_g \text{ and } S_g$, $\{\lambda_t\}_{t=1}^T$
1: $\boldsymbol{z}_T \sim \mathcal{N}(\mathbf{0}, \boldsymbol{I})$                                              ▷ Initial noise vector
2: **for** $t = T - 1, \ldots, 0$ **do**
3:     $\boldsymbol{\epsilon}_1 \sim \mathcal{N}(\mathbf{0}, \boldsymbol{I})$
4:     $\hat{\boldsymbol{\epsilon}}_{t+1} = \boldsymbol{s}_\theta(\boldsymbol{z}_{t+1}, t+1)$                                ▷ Compute the score
5:     $\hat{\boldsymbol{z}}_0(\boldsymbol{z}_{t+1}) = \frac{1}{\sqrt{\bar{\alpha}_{t+1}}}(\boldsymbol{z}_{t+1} - \sqrt{1 - \bar{\alpha}_{t+1}}\hat{\boldsymbol{\epsilon}}_{t+1})$     ▷ Predict $\hat{\boldsymbol{z}}_0$ using Tweedie's formula
6:     $\boldsymbol{z}_t' = \sqrt{\bar{\alpha}_t}\hat{\boldsymbol{z}}_0(\boldsymbol{z}_{t+1}) + \sqrt{1 - \bar{\alpha}_t - \eta\delta^2}\hat{\boldsymbol{\epsilon}}_{t+1} + \eta\delta\boldsymbol{\epsilon}_1$     ▷ Unconditional DDIM step
7:     **if** $t \in C$ **then**                                              ▷ ReSample time step
8:         Initialize $\hat{\boldsymbol{z}}_0(\boldsymbol{y})$ with $\hat{\boldsymbol{z}}_0(\boldsymbol{z}_{t+1})$
9:         **for** each step in gradient descent **do**
10:            $\boldsymbol{g} \leftarrow \nabla_{\hat{\boldsymbol{z}}_0(\boldsymbol{y})}\frac{1}{2}\|\boldsymbol{y} - \mathcal{A}(\mathcal{D}(\hat{\boldsymbol{z}}_0(\boldsymbol{y})))\|_2^2 + \lambda_t\nabla_{\hat{\boldsymbol{z}}_0(\boldsymbol{y})}\|S_g(\mathcal{D}(\hat{\boldsymbol{z}}_0(\boldsymbol{y}))) - \mathcal{D}(T_g(\hat{\boldsymbol{z}}_0(\boldsymbol{y})))\|_2^2$
11:            Update $\hat{\boldsymbol{z}}_0(\boldsymbol{y})$ using gradient $\boldsymbol{g}$
12:         **end for**
13:         $\boldsymbol{z}_t = \text{StochasticResample}(\hat{\boldsymbol{z}}_0(\boldsymbol{y}), \boldsymbol{z}_t', \gamma)$                   ▷ Map back to $t$
14:     **else**
15:         $\boldsymbol{z}_t = \boldsymbol{z}_t'$                           ▷ Unconditional sampling if not resampling
16:     **end if**
17: **end for**
18: $\boldsymbol{x}_0 = \mathcal{D}(\boldsymbol{z}_0)$                                         ▷ Output reconstructed image
19: **return** $\boldsymbol{x}_0$

---

**Algorithm 5** EquiCon-ReSample for Image Restoration Tasks

---

**Require:** Measurements $\boldsymbol{y}$, $\mathcal{A}(\cdot)$, Encoder $\mathcal{E}(\cdot)$, Decoder $\mathcal{D}(\cdot)$, Score function $\boldsymbol{s}_\theta(\cdot, t)$, Pretrained LDM Parameters $\beta_t, \bar{\alpha}_t, \eta, \delta$, Hyperparameter $\gamma$ to control $\sigma_t^2$, Time steps to perform resample $C$, $T_g$ and $S_g$, $\{\lambda_t\}_{t=1}^T$

1: $\boldsymbol{z}_T \sim \mathcal{N}(\mathbf{0}, \boldsymbol{I})$          ▷ Initial noise vector
2: **for** $t = T - 1, \ldots, 0$ **do**
3:     $\boldsymbol{\epsilon}_1 \sim \mathcal{N}(\mathbf{0}, \boldsymbol{I})$
4:     $\hat{\boldsymbol{\epsilon}}_{t+1} = \boldsymbol{s}_\theta(\boldsymbol{z}_{t+1}, t+1)$          ▷ Compute the score
5:     $\hat{\boldsymbol{z}}_0(\boldsymbol{z}_{t+1}) = \frac{1}{\sqrt{\bar{\alpha}_{t+1}}}(\boldsymbol{z}_{t+1} - \sqrt{1 - \bar{\alpha}_{t+1}}\hat{\boldsymbol{\epsilon}}_{t+1})$    ▷ Predict $\hat{\boldsymbol{z}}_0$ using Tweedie's formula
6:     $\boldsymbol{z}_t' = \sqrt{\bar{\alpha}_t}\hat{\boldsymbol{z}}_0(\boldsymbol{z}_{t+1}) + \sqrt{1 - \bar{\alpha}_t - \eta\delta^2}\hat{\boldsymbol{\epsilon}}_{t+1} + \eta\delta\boldsymbol{\epsilon}_1$    ▷ Unconditional DDIM step
7:     **if** $t \in C$ **then**          ▷ ReSample time step
8:        Initialize $\hat{\boldsymbol{z}}_0(\boldsymbol{y})$ with $\hat{\boldsymbol{z}}_0(\boldsymbol{z}_{t+1})$
9:        **for** each step in gradient descent **do**
10:          $\boldsymbol{g} \leftarrow \nabla_{\hat{\boldsymbol{z}}_0(\boldsymbol{y})} \frac{1}{2}\|\boldsymbol{y} - \mathcal{A}(\mathcal{D}(\hat{\boldsymbol{z}}_0(\boldsymbol{y})))\|_2^2 + \lambda_t \nabla_{\hat{\boldsymbol{z}}_0(\boldsymbol{y})}\|\hat{\boldsymbol{z}}_0(\boldsymbol{y}) - \mathcal{E}(S_g^{-1}(\mathcal{D}(T_g(\hat{\boldsymbol{z}}_0(\boldsymbol{y})))))\|_2^2$
11:          Update $\hat{\boldsymbol{z}}_0(\boldsymbol{y})$ using gradient $\boldsymbol{g}$
12:        **end for**
13:        $\boldsymbol{z}_t = \text{StochasticResample}(\hat{\boldsymbol{z}}_0(\boldsymbol{y}), \boldsymbol{z}_t', \gamma)$          ▷ Map back to $t$
14:     **else**
15:        $\boldsymbol{z}_t = \boldsymbol{z}_t'$          ▷ Unconditional sampling if not resampling
16:     **end if**
17: **end for**
18: $\boldsymbol{x}_0 = \mathcal{D}(\boldsymbol{z}_0)$          ▷ Output reconstructed image
19: **return** $\boldsymbol{x}_0$

---

**DPS.** Similar to PSLD, we integrate EquiReg into DPS by simply adding an additional gradient update step using our regularization term (Algorithm 6).

In our experiments, we use the official DPS implementation from Chung et al. (2023), running with its default settings to reproduce the baseline results.

---

**Algorithm 6** Equi-DPS for Image Restoration Tasks

---

**Require:** $T, \boldsymbol{y}, \{\zeta_t\}_{t=1}^T, \{\tilde{\sigma}_t\}_{t=1}^T, \boldsymbol{s}_\theta, \mathcal{E}, T_g$ and $S_g$, $\{\lambda_t\}_{t=1}^T$

1: $\boldsymbol{x}_T \sim \mathcal{N}(\mathbf{0}, \boldsymbol{I})$
2: **for** $t = T - 1$ **to** $0$ **do**
3:     $\hat{\boldsymbol{s}} \leftarrow \boldsymbol{s}_\theta(\boldsymbol{x}_t, t)$
4:     $\boldsymbol{x}_{0|t} \leftarrow \frac{1}{\sqrt{\bar{\alpha}_t}}(\boldsymbol{x}_t + (1 - \bar{\alpha}_t)\hat{\boldsymbol{s}})$
5:     $\boldsymbol{\epsilon} \sim \mathcal{N}(\mathbf{0}, \boldsymbol{I})$
6:     $\boldsymbol{x}_{t-1}' \leftarrow \frac{\sqrt{\alpha_t}(1 - \bar{\alpha}_{t-1})}{1 - \bar{\alpha}_t}\boldsymbol{x}_t + \frac{\sqrt{\bar{\alpha}_{t-1}}\beta_t}{1 - \bar{\alpha}_t}\boldsymbol{x}_{0|t} + \tilde{\sigma}_t\boldsymbol{\epsilon}$
7:     $\boldsymbol{x}_{t-1} \leftarrow \boldsymbol{x}_{t-1}' - \zeta_t \nabla_{\boldsymbol{x}_t}\|\boldsymbol{y} - \mathcal{A}(\boldsymbol{x}_{0|t})\|_2^2$
8:     $\boldsymbol{x}_{t-1} \leftarrow \boldsymbol{x}_{t-1} - \lambda_t \nabla_{\boldsymbol{x}_t}\|S_g(\mathcal{E}(\boldsymbol{x}_{0|t})) - \mathcal{E}(T_g(\boldsymbol{x}_{0|t}))\|_2^2$
9: **end for**
10: **return** $\boldsymbol{x}_0$

---

**SITCOM.** We augment the original SITCOM algorithm by introducing an additional equivariant refinement stage at each reverse diffusion step. After completing the standard measurement and backward-consistency gradient updates, we perform a second optimization over the equivariance loss, enforcing consistency between $\mathcal{E}(T_g(v))$ and $T_g(\mathcal{E}(v))$ (Algorithm 7).

In our experiments, we use the official SITCOM implementation from Alkhouri et al. (2025), running with its default settings to reproduce the baseline results.

---

**Algorithm 7** Equi-SITCOM for Image Restoration Tasks

---

**Require:** Measurements $\mathbf{y}$, forward operator $\mathcal{A}(\cdot)$, pre-trained DM $\epsilon_\theta(\cdot, \cdot)$, diffusion steps $N$, schedule $\bar{\alpha}_i$, measurement gradient steps $K$, equivariant gradient steps $K_{\text{equi}}$, stop $\delta$, lr $\gamma$, reg. $\lambda$.

**Ensure:** Restored image $\hat{\mathbf{x}}$.

1: **Initialize** $\mathbf{x}_N \sim \mathcal{N}(\mathbf{0}, \mathbf{I})$, $\Delta t = \lfloor \frac{T}{N} \rfloor$.

2: **for** $i = N, N-1, \ldots, 1$ **do**          ▷ Reducing diffusion sampling steps

3:     $\mathbf{v}_i^{(0)} \leftarrow \mathbf{x}_i$          ▷ Init for closeness (C3)

4:     **for** $k = 1, \ldots, K$ **do**          ▷ Adam on measurement/backward consistency (C1, C2)

5:        $\mathbf{v}_i^{(k)} \leftarrow \mathbf{v}_i^{(k-1)} - \gamma \nabla_{\mathbf{v}_i} \left[ \left\| \mathcal{A}\left( \frac{1}{\sqrt{\bar{\alpha}_i}} \left( \mathbf{v}_i - \sqrt{1-\bar{\alpha}_i}\, \epsilon_\theta(\mathbf{v}_i, i\Delta t) \right) \right) - \mathbf{y} \right\|_2^2 + \lambda \| \mathbf{x}_i - \mathbf{v}_i \|_2^2 \right] \Big|_{\mathbf{v}_i = \mathbf{v}_i^{(k-1)}}$

6:        **if** $\left\| \mathcal{A}\left( \frac{1}{\sqrt{\bar{\alpha}_i}} \left( \mathbf{v}_i^{(k)} - \sqrt{1-\bar{\alpha}_i}\, \epsilon_\theta(\mathbf{v}_i^{(k)}, i\Delta t) \right) \right) - \mathbf{y} \right\|_2^2 < \delta^2$ **then**

7:           **break**          ▷ Prevent noise overfitting

8:        **end if**

9:     **end for**

10:    $\mathbf{v}_i^{(0)} \leftarrow \mathbf{v}_i^{(k)}$          ▷ Initialize to optimized $\mathbf{v}_i$

11:    **for** $k = 1, \ldots, K_{\text{equi}}$ **do**

12:       $\mathbf{v}_i^{(k)} \leftarrow \mathbf{v}_i^{(k-1)} - \gamma \nabla_{\mathbf{v}_i} \left[ \left\| \mathcal{E}\left( T_g(\mathbf{v}_i^{(k)}) \right) - T_g\left( \mathcal{E}(\mathbf{v}_i^{(k)}) \right) \right\|_2^2 \right] \Big|_{\mathbf{v}_i = \mathbf{v}_i^{(k-1)}}$

13:       **if** $\left\| \mathcal{E}\left( T_g(\mathbf{v}_i^{(k)}) \right) - T_g\left( \mathcal{E}(\mathbf{v}_i^{(k)}) \right) \right\|_2^2 < \delta^2$ **then**

14:          **break**

15:       **end if**

16:    **end for**

17:    $\hat{\mathbf{v}}_i \leftarrow \mathbf{v}_i^{(k)}$          ▷ Backward diffusion consistency (C2)

18:    $\hat{\mathbf{x}}_0' \leftarrow \frac{1}{\sqrt{\bar{\alpha}_i}} \left[ \hat{\mathbf{v}}_i - \sqrt{1-\bar{\alpha}_i}\, \epsilon_\theta(\hat{\mathbf{v}}_i, i\Delta t) \right]$          ▷ Backward consistency (C2)

19:    $\mathbf{x}_{i-1} \leftarrow \sqrt{\bar{\alpha}_{i-1}}\, \hat{\mathbf{x}}_0' + \sqrt{1-\bar{\alpha}_{i-1}}\, \boldsymbol{\eta}_i, \quad \boldsymbol{\eta}_i \sim \mathcal{N}(\mathbf{0}, \mathbf{I})$      ▷ Forward consistency (C3)

20: **end for**

21: **return** $\hat{\mathbf{x}} = \mathbf{x}_0$

---

## F EXPERIMENT SETUP FOR PDE RECONSTRUCTIONS

**Helmholtz equation.** The Helmholtz equation represents wave propagation in heterogeneous media:

$$\nabla^2 u(x) + k^2 u(x) = a(x), \quad x \in (0,1)^2, \tag{6}$$

with $k = 1$ and $u|_{\partial\Omega} = 0$. Coefficient fields $a(x)$ are generated according to $a \sim \mathcal{N}(0, (-\Delta + 9\mathbf{I})^2)$. We note that this system has reflection equivariance along $x_1 = \frac{1}{2}, x_2 = \frac{1}{2}, x_1 = x_2$ and rotation equivariance by $\frac{\pi}{2}, \pi, \frac{3\pi}{2}$.

**Navier-Stokes equations.** Following the methodology of (Li et al., 2020), we model the time evolution of a vorticity field, $u(x, t)$, governed by:

$$\partial_t u(x,t) + \boldsymbol{w}(x,t) \cdot \nabla u(x,t) = \nu \Delta u(x,t) + f(x), \quad x \in (0,1)^2, t \in (0,T], \tag{7}$$

$$\nabla \cdot \boldsymbol{w}(x,t) = 0, \quad x \in (0,1)^2, t \in [0,T], \tag{8}$$

$$u(x,0) = a(x), \quad x \in (0,1)^2, \tag{9}$$

where $\boldsymbol{w}$ is the velocity field; $\nu = \frac{1}{1000}$, viscosity; and $f$, a fixed forcing term. The initial condition $a(x)$ is drawn from $\mathcal{N}(0, 7^{3/2}(-\Delta + 49\mathbf{I})^{-5/2})$ under periodic boundary conditions. The forcing term is $f(x) = 0.1 (\sin(2\pi(x_1 + x_2)) + \cos(2\pi(x_1 + x_2)))$. We borrow the dataset from (Huang et al., 2024). We note that this system has a reflection symmetry along the $x_1 = x_2$ axis.

**Implementation details.** EquiReg, as a regularizer for diffusion posterior sampling, can be adapted to many inverse solvers in a plug-and-play manner. For PDE experiments, we use the same model weights and configurations as FunDPS (Yao et al., 2025). Error rates are calculated using the $L^2$ relative error between the predicted and true solutions, averaged on 100 randomly selected test samples. We provide the information on the EquiReg scaling weights in Table 13.

Table 13: **EquiReg loss used in PDE experiments.**

|  | Helmholtz | | Navier-Stokes | |
| --- | --- | --- | --- | --- |
|  | Forward | Inverse | Forward | Inverse |
| EquiReg Norm Type | MSE | L2 | MSE | L2 |
| EquiReg Weight $\lambda$ | 100 | 100 | 100 | 1000 |

# G  THEORETICAL ANALYSIS

## G.1  SUMMARY OF THE ANALYSIS

The theoretical framework presented in this paper is intended to motivate and guide the design of effective regularizers. This perspective, grounded in optimal transport theory (Ferreira and Valencia-Guevara, 2018), serves as an intuitive interpretation of the dynamics and motivates the design of regularization strategies such as EquiReg. We note that whether diffusion models follow exact Wasserstein dynamics still remains an open problem (Zheng et al., 2025).

**Proposition G.1.** *Let $\rho(\boldsymbol{x}, t)$ be the distribution of $\boldsymbol{x}_{T-t}$ driven by the ideal reverse dynamics (eq. (3)). Then, the evolution of $\rho$ follows the Wasserstein-2 gradient flow associated with minimizing functional $\Phi(\rho, t)$ defined as $\beta_{T-t} \int [\rho\phi(\boldsymbol{x}, t) + \frac{1}{2}\rho \log \rho] d\boldsymbol{x}$, where $\phi(\boldsymbol{x}, t) = -(\log p_{T-t}(\boldsymbol{x}|\boldsymbol{y}) + \frac{1}{4}\|\boldsymbol{x}\|^2)$.*

The dynamics of $\rho$ remain the same if we replace $\phi(\boldsymbol{x}, t)$ with $\phi_C(\boldsymbol{x}, t) := \phi(\boldsymbol{x}, t) - C(t)$ for arbitrary temporal function $C(t)$. Without loss of generality, we assume $\phi_C(\boldsymbol{x}, t) < 0$ for all $\boldsymbol{x}$ and $t$. In practice, the density function $p_{T-t}$ is not available and thus $\phi_C(\boldsymbol{x}, t)$ is approximated as $\hat{\phi}$ with $p_{T-t}(\boldsymbol{x}_{T-t}|\boldsymbol{y}) \approx \tilde{C}p_{T-t}(\boldsymbol{x}_{T-t})p(\boldsymbol{y}|\mathbb{E}[\boldsymbol{x}_0|\boldsymbol{x}_{T-t}])$ where $\tilde{C}$ only depends on $\boldsymbol{y}$.

Because the conditional expectation $\mathbb{E}[\boldsymbol{x}_0|\boldsymbol{x}_{T-t}]$ is a linear combination of all candidate $\boldsymbol{x}_0$, the approximation remains relatively accurate when $T - t$ is small (i.e., $\boldsymbol{x}_{T-t}$ stays close to the data manifold under low noise) but may incur high error for larger $T-t$, as shown in Figure 2b. To mitigate this, we reweight the contributions to the first term of $\Phi$, down-weighting unreliable estimates, and amplifying the reliable ones. The resulting reweighted functional is

$$\tilde{\Phi}(\rho, t) = \beta_{T-t}\left[Z_t^{-1} \int \rho(\boldsymbol{x})\hat{\phi}_c(\boldsymbol{x}, t)e^{\frac{\mathcal{R}(\boldsymbol{x})}{\hat{\phi}_c(\boldsymbol{x}, t)}} \mathrm{d}\boldsymbol{x} + \tfrac{1}{2}\int \rho(\boldsymbol{x}) \log \rho(\boldsymbol{x}) d\boldsymbol{x}\right], \qquad (10)$$

where $Z_t = \int e^{\frac{\mathcal{R}(\boldsymbol{x})}{\hat{\phi}_c(\boldsymbol{x}, t)}} \mathrm{d}\boldsymbol{x}$ is the normalizing factor, and $\mathcal{R}(\boldsymbol{x})$ is a positive regularization that is nearly zero near the data manifold and much larger elsewhere. Intuitively, since $\hat{\phi}_C < 0$, the weight is nearly one for $\boldsymbol{x}$ near the data manifold and much smaller elsewhere.

**Proposition G.2.** *(Informal) The evolution of $\rho$, the probability distribution of $\boldsymbol{x}_{T-t}$ driven by the practical and regularized reverse dynamics (eq. (11)), is an approximation of the Wasserstein-2 gradient flow associated with minimizing $\tilde{\Phi}$.*

$$\mathrm{d}\boldsymbol{x} = [-\tfrac{\beta_t}{2}\boldsymbol{x}\mathrm{d}t - \beta_t\nabla_{\boldsymbol{x}_t}(\log p_t(\boldsymbol{x}_t) + \log \int p(\boldsymbol{y}|\boldsymbol{x}_0)\tilde{p}_t(\boldsymbol{x}_0|\boldsymbol{x}_t)\mathrm{d}\boldsymbol{x}_0 - \mathcal{R}(\boldsymbol{x}_t))]\mathrm{d}t + \sqrt{\beta_t}\mathrm{d}\bar{\boldsymbol{w}} \quad (11)$$

## G.2  PRELIMINARY AND NOTATIONS

We first remind the readers of gradient flow under the Wasserstein-2 metric and introduce the notations related to the diffusion model.

**Wasserstein Gradient Flow**  Let $\mathcal{F} : \mathcal{P}_2(\mathbb{R}^d) \rightarrow \mathbb{R} \cup \{+\infty\}$ be a functional of probability distributions. The Wasserstein gradient flow of $\mathcal{F}$ is characterized by the minimizing movement scheme (also known as JKO scheme) introduced by (Jordan et al., 1998). For a fixed time step $\tau > 0$, the sequence $(\rho_k)_{k\in\mathbb{N}}$ of probability densities is defined recursively by:

$$\rho_{k+1} \in \arg \min_{\rho\in\mathcal{P}_2(\mathbb{R}^d)} \left\{\frac{1}{2\tau}W_2^2(\rho, \rho_k) + \mathcal{F}(\rho)\right\},$$

where $W_2$ denotes the 2-Wasserstein distance, and each $\rho_k$ is a probability density representing the distribution at time $t = k\tau$. In the limit $\tau \rightarrow 0$, this discrete-time scheme recovers the continuous-time gradient flow of $\mathcal{F}$ under the $W_2$ metric.

**Diffusion Model**   A diffusion model defines a forward stochastic process $(\boldsymbol{x}_t)_{t \in [0,T]}$ governed by the Itô SDE:

$$\mathrm{d}\boldsymbol{x}_t = f(\boldsymbol{x}_t, t)\,\mathrm{d}t + \sqrt{\beta_t}\,\mathrm{d}\boldsymbol{w}_t, \tag{12}$$

where $\boldsymbol{w}_t$ is standard Brownian motion, $\beta_t > 0$ is a time-dependent variance schedule, and $f(\boldsymbol{x}, t)$ is a drift term. For instance, $f \equiv 0$ for a variance-exploding SDE and $f(\boldsymbol{x}, t) = -\frac{\beta_t}{2}\boldsymbol{x}$ for a variance-preserving SDE defined in (Song et al., 2021). In this work, we carry out our analysis under a more general setting.

**Assumption G.1.** *The drift term is a gradient field, $f(\boldsymbol{x}, t) = \nabla h(\boldsymbol{x}, t)$ for a scalar function $h$.*

This process progressively transforms an initial data distribution $\boldsymbol{x}_0 \sim p_0$ into a tractable reference distribution (e.g., approximately a Gaussian $\mathcal{N}(0, I)$) at time $T$.

Sampling is performed by simulating the *reverse-time SDE*:

$$\mathrm{d}\boldsymbol{x}_t = [f(\boldsymbol{x}_t, t) - \beta_t \nabla_{\boldsymbol{x}} \log p_t(\boldsymbol{x}_t)]\,\mathrm{d}t + \sqrt{\beta_t}\,\mathrm{d}\bar{\boldsymbol{w}}_t, \tag{13}$$

where $p_t$ is the marginal density of $\boldsymbol{x}_t$, and $\bar{\boldsymbol{w}}_t$ is a standard Brownian motion in reverse time.

In practice, the score function $\nabla_{\boldsymbol{x}} \log p_t(\boldsymbol{x})$ is approximated by a neural network $s_\theta(\boldsymbol{x}, t)$ trained to estimate the score of the forward process. For *conditional sampling*, where we sample $\boldsymbol{x}_0$ given some observed variable $y$, the score is replaced by $\nabla_{\boldsymbol{x}} \log p_t(\boldsymbol{x}|\boldsymbol{y})$ and decomposed as

$$\nabla_{\boldsymbol{x}} \log p_t(\boldsymbol{x}|\boldsymbol{y}) = \nabla_{\boldsymbol{x}} \log p_t(\boldsymbol{x}) + \nabla_{\boldsymbol{x}} \log p_t(\boldsymbol{y}|\boldsymbol{x}), \tag{14}$$

based on Bayes' rule.

To simplify notation in the sequel, we perform a time reparameterization $t = T - t'$, so that the reverse process is written as a forward SDE over $t \in [0, T]$:

$$\mathrm{d}\boldsymbol{x}_t = -[f(\boldsymbol{x}_t, T - t) - \beta_{T-t}[\nabla_{\boldsymbol{x}} \log p_{T-t}(\boldsymbol{x}_t) + \nabla_{\boldsymbol{x}} \log p_t(y|\boldsymbol{x}_t)]]\,\mathrm{d}t + \sqrt{\beta_{T-t}}\,\mathrm{d}\boldsymbol{w}_t, \tag{15}$$

This form describes the generative process as evolving forward from $t = 0$ to $t = T$, matching the usual direction of analysis in gradient flow frameworks.

## G.3   Proof of Proposition G.1

In this work, we consider Wasserstein gradient flow under the setting where the functional $\mathcal{F}$ depends on time.

**Lemma G.1.** *Consider a time-dependent functional $\mathcal{F}(\rho, t) = \int \rho(\boldsymbol{x})V(\boldsymbol{x}, t)\mathrm{d}x + \int \alpha(t)\rho \log \rho \mathrm{d}x$. Then the particle description of Wasserstein-2 gradient flow associated with this functional derived by JKO scheme is*

$$\mathrm{d}\boldsymbol{x}_t = -\nabla V(\boldsymbol{x}_t, t)\mathrm{d}t + \sqrt{2\alpha(t)}\mathrm{d}\boldsymbol{w}_t. \tag{16}$$

*Proof.* Consider the following optimization

$$\min_{\rho'} \mathcal{F}(\rho', t + \Delta t) - \mathcal{F}(\rho, t) + \frac{1}{2\Delta t}W_2^2(\rho, \rho'), \tag{17}$$

where the change of density is restricted to the Liouville equation

$$\partial_t \rho = -\nabla \cdot (\rho v(\boldsymbol{x}, t)), \text{ and } \rho'(x) = \rho(x) - \Delta t \nabla \cdot (\rho(\boldsymbol{x})v(\boldsymbol{x})) + o(\Delta t). \tag{18}$$

Using the static formulation of $W_2$ distance, we have

$$W_2^2(\rho, \rho') = \int \rho(\boldsymbol{x})\|\boldsymbol{x} - T^*(\boldsymbol{x})\|^2\,\mathrm{d}\boldsymbol{x} = \Delta t^2 \int \rho(\boldsymbol{x})\|v^*(\boldsymbol{x})\|^2\,\mathrm{d}\boldsymbol{x}, \tag{19}$$

where $T^*(\boldsymbol{x})$ is the optimal transport map, and $v^*(\boldsymbol{x})$ is the associated optimal velocity field.

Thus, we can rewrite the eq. (17) as

$$\inf_v \mathcal{F}(\rho, t) - \Delta t \int \nabla \cdot (\rho(\boldsymbol{x})v(\boldsymbol{x})) \frac{\delta \mathcal{F}(\rho, t)}{\delta \rho}(\boldsymbol{x})\,\mathrm{d}\boldsymbol{x} + \Delta t \int \left[\rho(\boldsymbol{x})\partial_t V(\boldsymbol{x}, t) + \dot{\alpha}(t)\rho \log \rho\right]\mathrm{d}\boldsymbol{x} \tag{20}$$

$$- \mathcal{F}(\rho, t) + \frac{\Delta t}{2} \int \rho(\boldsymbol{x})\|v(\boldsymbol{x})\|^2\,\mathrm{d}\boldsymbol{x}, \tag{21}$$

which simplifies to

$$\min_{v} \int \rho(\boldsymbol{x}) \left\langle v(\boldsymbol{x}), \nabla \frac{\delta \mathcal{F}(\rho, t)}{\delta \rho}(\boldsymbol{x}) \right\rangle \mathrm{d}\boldsymbol{x} + \frac{1}{2} \int \rho(\boldsymbol{x}) \|v(\boldsymbol{x})\|^2 \, \mathrm{d}\boldsymbol{x}, \tag{22}$$

since the last term in the first line of (20) does not depend on $v$. and further to

$$\min_{v} \int \rho(\boldsymbol{x}) \left\| v(\boldsymbol{x}) + \nabla \frac{\delta \mathcal{F}(\rho, t)}{\delta \rho}(\boldsymbol{x}) \right\|^2 \, \mathrm{d}\boldsymbol{x}. \tag{23}$$

From the optimality condition of the above problem, we obtain

$$v(\boldsymbol{x}, t) = -\nabla \frac{\delta \mathcal{F}(\rho, t)}{\delta \rho}(\boldsymbol{x}) = -(\nabla V(\boldsymbol{x}, t) + \alpha(t) \nabla \log \rho(\boldsymbol{x}, t)). \tag{24}$$

We note that By Hörmander's theorem, a smooth density $\rho(\boldsymbol{x}, t)$ exists for $t > 0$, ensuring that the above $v$ is well-defined. The corresponding evolution of probability density is

$$\partial_t \rho(\boldsymbol{x}, t) = -\nabla \cdot (\rho(\boldsymbol{x}, t) v(\boldsymbol{x}, t)) \tag{25}$$

$$= \nabla \cdot (\rho(\boldsymbol{x}, t)(\nabla V(\boldsymbol{x}, t) + \alpha(t) \frac{\nabla \rho(\boldsymbol{x}, t)}{\rho})) \tag{26}$$

$$= -\nabla \cdot (\rho(\boldsymbol{x}, t)(-\nabla V(\boldsymbol{x}, t)) + \alpha(t) \Delta \rho(\boldsymbol{x}, t), \tag{27}$$

which is exactly the Fokker-Planck equation describing the evolution of the probability density describing the particles following

$$\mathrm{d}\boldsymbol{x}_t = -\nabla V(\boldsymbol{x}_t, t) \mathrm{d}t + \sqrt{2\alpha(t)} \mathrm{d}\boldsymbol{w}_t. \tag{28}$$

$\square$

Now we come back to Proposition G.1. From eq. (15) we know that choosing

$$V(\boldsymbol{x}, t) = h(\boldsymbol{x}, T - t) - \beta_{T-t}[\log p_{T-t}(\boldsymbol{x}) + \log p_{T-t}(\boldsymbol{y}|\boldsymbol{x})] \text{ and } \alpha(t) = \frac{\beta_{T-t}}{2} \tag{29}$$

in Lemma G.1 completes the proof, where $h$ is defined in Assumption G.1.

## G.4  DETAILED VERSION OF PROPOSITION G.2

In practice, one does not have access to $\log p_t(\boldsymbol{y}|\boldsymbol{x}_t)$ which appears in the reverse SDE. The most popular approach is do the following approximation,

$$p_t(\boldsymbol{y}|\boldsymbol{x}_t) = \int p(\boldsymbol{y}|\boldsymbol{x}_0) p(\boldsymbol{x}_0|\boldsymbol{x}_t) \mathrm{d}\boldsymbol{x}_0 = \mathbb{E}_{\boldsymbol{x}_0 \sim p(\boldsymbol{x}_0|\boldsymbol{x}_t)}[p(\boldsymbol{y}|\boldsymbol{x}_0)] \approx p(\boldsymbol{y}| \mathbb{E}[\boldsymbol{x}_0|\boldsymbol{x}_t]), \tag{30}$$

which can be interpreted as exchanging two operations, the conditional expectation and the measurement $p(\boldsymbol{y}|\cdot)$.

As discussed in the main text, since the conditional expectation is a linear combination over all possible values of $\boldsymbol{x}_0$, it may fall outside the data manifold, resulting in physically invalid samples. One of the central challenges in diffusion-based inverse sampling is guiding the sampling trajectory, generated by the reverse SDE dynamics, toward the data manifold. A common strategy is to incorporate regularization into the reverse SDE to encourage manifold adherence. In this work, building on the perspective of Wasserstein gradient flow as outlined above, we provide a novel interpretation of the role played by such regularization terms.

We show that the regularizer serves to reweight the contribution of different regions in the calculation of the underlying functional being minimized, $\Phi(\rho, t)$ defined in Proposition G.1. Specifically, it amplifies the influence of regions where the density estimate is reliable (typically near the data manifold), while down-weighting regions with poor approximation quality of based on eq. (30), often corresponding to off-manifold samples.

Following from what we have shown in the main text, $\Phi(\rho, t)$ has the form of $\beta_{T-t} \int [\rho \phi(\boldsymbol{x}, t) + \frac{1}{2} \rho \log \rho] d\boldsymbol{x}$ for a function $\phi(\boldsymbol{x}, t)$, which can be derived by (29). The $\log p_t(\boldsymbol{y}|\boldsymbol{x})$ term in (29) or

$\nabla \log p_t(\boldsymbol{y}|\boldsymbol{x})$ term in (28), equivalently, is computed based on approximation (30). We denote the corresponding approximation of $\phi(\boldsymbol{x}, t)$ as $\hat{\phi}(\boldsymbol{x}, t)$. As discussed in the main text, we can assume without loss of generality that $\phi(\boldsymbol{x}, t) < 0$ and $\hat{\phi}(\boldsymbol{x}, t) < 0$. We have

$$\hat{\Phi}(\rho, t) = \beta_{T-t}\Big[\int_{\boldsymbol{x} \in N(\mathcal{M})} \rho(\boldsymbol{x})\hat{\phi}(\boldsymbol{x}, t)\mathrm{d}\boldsymbol{x} + \int_{\boldsymbol{x} \notin N(\mathcal{M})} \rho(\boldsymbol{x})\hat{\phi}(\boldsymbol{x}, t)\mathrm{d}\boldsymbol{x} + \frac{1}{2}\int \rho \log \rho \mathrm{d}\boldsymbol{x}\Big], \quad (31)$$

where $N(\mathcal{M})$ denotes a neighborhood of the data manifold $\mathcal{M}$. Intuitively, we aim to focus on the contribution from regions near $\mathcal{M}$, which corresponds to the first term, while down-weighting the influence of points farther away, where the approximation tends to be unreliable. For instance, we can introduce two positive weights $A \gg B$ and adopt the modified functional

$$\tilde{\Phi}(\rho, t) = \beta_{T-t}\Big[A\int_{\boldsymbol{x} \in N(\mathcal{M})} \rho(\boldsymbol{x})\hat{\phi}(\boldsymbol{x}, t)\mathrm{d}\boldsymbol{x} + B\int_{\boldsymbol{x} \notin N(\mathcal{M})} \rho(\boldsymbol{x})\hat{\phi}(\boldsymbol{x}, t)\mathrm{d}\boldsymbol{x} + \frac{1}{2}\int \rho \log \rho \mathrm{d}\boldsymbol{x}\Big]. \quad (32)$$

In this work, we further generalize this idea and consider a continuous weight function,

$$\tilde{\Phi}(\rho, t) = \beta_{T-t}\Big[\int \rho(\boldsymbol{x})\hat{\phi}(\boldsymbol{x}, t)\lambda(\boldsymbol{x})\mathrm{d}\boldsymbol{x} + \frac{1}{2}\int \rho \log \rho \mathrm{d}\boldsymbol{x}\Big], \quad (33)$$

where the non-negative weight $\lambda(\boldsymbol{x})$ is large for $\boldsymbol{x} \in N(\mathcal{M})$ and small elsewhere.

In practice, a nonnegative regularization function $\mathcal{R}(\boldsymbol{x})$ is introduced, ideally being nearly zero for $\boldsymbol{x}$ near the data manifold and much larger elsewhere. We consider the following modified functional with weight function $\lambda(\boldsymbol{x}, t) := e^{\frac{\mathcal{R}(\boldsymbol{x})}{\hat{\phi}(\boldsymbol{x}, t)}}$,

$$\tilde{\Phi}(\rho, t) = \beta_{T-t}\Big[\int \rho(\boldsymbol{x})\hat{\phi}(\boldsymbol{x}, t)e^{\frac{\mathcal{R}(\boldsymbol{x})}{\hat{\phi}(\boldsymbol{x}, t)}}\mathrm{d}\boldsymbol{x} + \frac{1}{2}\int \rho(\boldsymbol{x}) \log \rho(\boldsymbol{x})d\boldsymbol{x}\Big]. \quad (34)$$

Note that $\hat{\phi} < 0$, we have that

$$\mathcal{R}(\boldsymbol{x}) \approx \begin{cases} 0, & \boldsymbol{x} \in N(\mathcal{M}) \\ \gg 1, & \boldsymbol{x} \text{ far away from } N(\mathcal{M}) \end{cases} \quad \Rightarrow \quad \lambda(\boldsymbol{x}, t) \approx \begin{cases} 1, & \boldsymbol{x} \in N(\mathcal{M}) \\ 0, & \boldsymbol{x} \text{ far away from } N(\mathcal{M}) \end{cases}.$$

Next, we consider practical algorithms based on this reweighted functional. In practice, we only have the score function instead of the function value of $\log p_{T-t}(\boldsymbol{x})$. Thus, the Wasserstein gradient flow associated with (34) is intractable since we cannot evaluate the weight function. We consider the following approximation based on $e^\delta \approx 1 + \delta$ when $\delta$ is sufficiently small,

$$\tilde{\Phi}(\rho, t) \approx \beta_{T-t}\Big[\int \rho(\boldsymbol{x})\hat{\phi}(\boldsymbol{x}, t)\big(1 + \frac{\mathcal{R}(\boldsymbol{x})}{\hat{\phi}(\boldsymbol{x}, t)}\big)\mathrm{d}\boldsymbol{x} + \frac{1}{2}\int \rho(\boldsymbol{x}) \log \rho(\boldsymbol{x})d\boldsymbol{x}\Big] \quad (35)$$

$$= \beta_{T-t}\Big[\int \rho(\boldsymbol{x})\big(\hat{\phi}(\boldsymbol{x}, t) + \mathcal{R}(\boldsymbol{x})\big)\mathrm{d}\boldsymbol{x} + \frac{1}{2}\int \rho(\boldsymbol{x}) \log \rho(\boldsymbol{x})d\boldsymbol{x}\Big]. \quad (36)$$

By Lemma G.1, the dynamics of $\boldsymbol{x}$ driven by the Wasserstein gradient flow associated with the approximated functional above is

$$\mathrm{d}\boldsymbol{x} = [-f(\boldsymbol{x}, T-t) - \beta_{T-t}\nabla_{\boldsymbol{x}}\big(\log p_{T-t}(\boldsymbol{x}) + \log \hat{p}_{T-t}(\boldsymbol{y}|\boldsymbol{x}) + \mathcal{R}(\boldsymbol{x})\big)]\mathrm{d}t + \sqrt{\beta_{T-t}}\mathrm{d}\bar{\boldsymbol{w}}. \quad (37)$$

This completes the proof.

**Remark 1.** *Since $\hat{\phi} < 0$, and $e^A \geq 1 + A$ for any $A \in \mathbb{R}$, the dynamics derived by the approximated functional in (36) is evolving to minimize an upper bound of the reweighted functional $\tilde{\Phi}$.*

## H  ADDITIONAL BACKGROUND INFORMATION

**Solving inverse problems with deep learning prior to diffusion models.**  Earlier works (Metzler et al., 2016; Romano et al., 2017; Zhang et al., 2017; Metzler et al., 2017) used deep neural networks as denoisers to solve inverse problems. Furthermore, deep generative models such as variational autoencoders (VAEs) (Kingma, 2013), and generative adversarial networks (GANs) (Goodfellow

et al., 2014) were employed. Notable applications include compressed sensing (Bora et al., 2017) and MRI (Jalal et al., 2021).

**Applications on diffusion models to solve inverse problems.** Most popular applications include image restoration (Chung et al., 2023; 2022b; Kawar et al., 2022; Lugmayr et al., 2022; Saharia et al., 2022; Song et al., 2023a; Rout et al., 2023; Zhu et al., 2023; **?**; Zirvi et al., 2025), medical imaging (Song et al., 2022; Chung and Ye, 2022; Chung et al., 2022a; Hung et al., 2023; Dorjsembe et al., 2024; Li et al., 2024; Kazerouni et al., 2023; Bian et al., 2024), and solving partial differential equations (PDEs) (Isakov, 2006; Huang et al., 2024; Shysheya et al., 2024; Liu et al., 2023; Li et al., 2025; Baldassari et al., 2023; Mammadov et al., 2024a; Yao et al., 2025). On the methodology side, there has been numerous advancements (Chung et al., 2023; 2022b; Kawar et al., 2022; Lugmayr et al., 2022; Saharia et al., 2022; Song et al., 2023a; Rout et al., 2023; Zhu et al., 2023; **?**; Zirvi et al., 2025; Song et al., 2022; Chung and Ye, 2022; Chung et al., 2022a; Hung et al., 2023; Dorjsembe et al., 2024; Li et al., 2024; Kazerouni et al., 2023; Bian et al., 2024; Huang et al., 2024; Shysheya et al., 2024; Mammadov et al., 2024b; Cardoso et al., 2024).

**Resources for Definition 3.2 on vanishing-error autoencoders.** Manifold constrained distribution-dependent equivariance error uses the notion of *vanishing-error autoencoders* (Shao et al., 2018; Anders et al., 2020; He et al., 2024) (Definition H.1), also known as an asymptotically-trained autoencoder (Anders et al., 2020) or a perfect autoencoder (He et al., 2024). Vanishing-error autoencoders have previously been employed by diffusion-based inverse solvers to preserve the diffusion process on the manifold (He et al., 2024).

**Definition H.1** (Vanishing-Error Autoencoder). *A vanishing-error autoencoder under the manifold $\mathcal{M}$ with encoder $\mathcal{E} : \mathcal{X} \to \mathcal{Z}$ and decoder $\mathcal{D} : \mathcal{Z} \to \mathcal{X}$ with $\mathcal{Z} = \mathbb{R}^k$ where $k < d$, has zero reconstruction error under the support of the data distribution $\mathcal{X}$, i.e., $\forall \boldsymbol{x} \in \mathcal{X} \subset \mathcal{M}, \boldsymbol{x} = \mathcal{D}(\mathcal{E}(\boldsymbol{x}))$. It follows that the decoder is surjective on the data manifold, $\mathcal{D} : \mathcal{Z} \to \mathcal{M}$ (He et al., 2024), and the encoder-decoder composition forms an identity map, i.e., $\forall \boldsymbol{z} \in \mathcal{M}, \boldsymbol{z} = \mathcal{E}(\mathcal{D}(\boldsymbol{z}))$.*

**Equivariance.** Let $\boldsymbol{z} \in \mathbb{R}^d$ and $\boldsymbol{x} = f(\boldsymbol{z}) \in \mathbb{R}^d$. For rotation and reflection equivariance, the transformations $T_g$ and $S_g$ can be defined by a rotation matrix $\boldsymbol{R} \in \mathbb{R}^{d \times d}$; then, a function $f$ with the rotation equivariant property would satisfy $\boldsymbol{R}\boldsymbol{x} = f(\boldsymbol{R}\boldsymbol{z})$. For translation equivariance, the transformations would be $T_g(\boldsymbol{z}) = \boldsymbol{z} + g$ and $S_g(\boldsymbol{x}) = \boldsymbol{x} + g$, where $g \in \mathbb{R}^d$. Hence, for a translation equivariance function $f$, we would have $\boldsymbol{x} + g = f(\boldsymbol{z} + g)$. For the case where the output dimension is larger than the input, $f : \mathbb{R}^k \to \mathbb{R}^d$ with $d > k$, translation equivariance can be defined up to a discrete scale, i.e., $T_g(\boldsymbol{z}) = \boldsymbol{z} + g$ and $S_g(\boldsymbol{x}) = T_{sg}(\boldsymbol{z})$ where $s = {}^d/k$. The equivariance properties of translation, rotation, and reflections, combined, are referred to as E(3) symmetries. Without reflections, the symmetries form a Euclidean group SE(3) (Thomas et al., 2018; Fuchs et al., 2020).

E(3), SE(3), and SO(3) are important symmetry groups in 3D Euclidean space, with well-established applications in physics and chemistry, computer vision, and reinforcement learning (Cohen and Welling, 2016; Thomas et al., 2018; Hoogeboom et al., 2022; Xu et al., 2024; Park et al., 2025). Finally, our contributions are complementary to, and can be combined with, the growing literature on meta-learning and automatic symmetry discovery to learn symmetry groups and their actions directly from data (Zhou et al.; Quessard et al., 2020; Dehmamy et al., 2021; Mohapatra et al., 2025).

**Data manifold hypothesis.** Let data $\boldsymbol{x} \in \mathcal{X} \subset \mathbb{R}^d$ be in an ambient space of dimension $d$ with support $\mathcal{X}$ distribution. We assume that data are sampled from a low-dimensional manifold $\mathcal{M}$ (Cayton et al., 2005; Ma and Fu, 2012) embedded in a high-dimensional space (Assumption H.1). This hypothesis is popular in machine learning (Bordt et al., 2023), and has been studied mathematically in the literature (Narayanan and Mitter, 2010; Bortoli, 2022). Moreover, empirical evidence in image processing supports the manifold hypothesis (Weinberger and Saul, 2006; Fefferman et al., 2016), and diffusion-based solvers assume this property (He et al., 2024; Chung et al., 2022b; 2023).

**Assumption H.1** (Manifold Hypothesis). *Let $\boldsymbol{x} \in \mathcal{X} \subset \mathbb{R}^d$ be a data sample. The support $\mathcal{X}$ of the data distribution lies on a $k$ dimensional manifold $\mathcal{M}$ within an ambient space $\mathbb{R}^d$ where $k \ll d$.*

# I  ADDITIONAL EXPERIMENTS ON MPE FUNCTIONS

We compare several networks and show that MPE consistently emerges across them: as Gaussian noise is added to natural images, the equivariance loss systematically increases. We examine both (i) the emergence of MPE properties in different functions (neural networks) and (ii) the effect of using

these functions within EquiReg on identical inverse problem settings. Specifically, for each dataset (FFHQ 256 and ImageNet), we consider four MPE function classes: (1) the pre-trained encoder of the latent diffusion model (LDM) used in our main experiments (Rombach et al., 2022), (2) a CNN autoencoder trained on the corresponding training distribution (FFHQ or ImageNet) with flip (FFHQ) or rotation (ImageNet) augmentations, (3) a pre-trained ResNet-50 (He et al., 2016), and (4) a pre-trained CLIP encoder (Radford et al., 2021). For each network, we evaluate equivariance loss under the relevant symmetry (flip for FFHQ, rotation for ImageNet) as Gaussian noise is added to 100 natural images at increasing noise levels.

Our results show that all four networks exhibit clear MPE behavior; their equivariance error increases as the noise level of the input grows. At the same time, the strength of the MPE property varies across architectures. Notably, the CNN autoencoder trained on the true data distribution shows the strongest MPE behavior, with equivariance error rising most sharply as images are corrupted, in line with our systematic guidelines for constructing MPE functions (Section 3). This is precisely the regime where the training distribution of the function matches the distribution of the inverse problem (e.g., training on ImageNet train and evaluating on ImageNet test). In contrast, the LDM encoder exhibits the weakest MPE signal among the four, while ResNet-50 and CLIP fall between these extremes. These trends are visualized in Figures 15 and 20.

We then apply each of these MPE functions within the EquiReg framework on the same inverse problem configurations: two datasets (FFHQ 256 and ImageNet), two diffusion-based solvers (DPS and SITCOM), and two tasks (super-resolution and motion deblurring). Across all settings and all MPE choices, EquiReg consistently improves reconstruction quality relative to the corresponding baseline without regularization ("None"). Tables 14 and 15 summarize these results. Taken together, these experiments demonstrate that (a) MPE properties naturally emerge in widely used pre-trained networks, making EquiReg easy to deploy in practice, and (b) EquiReg is robust across a range of MPE functions, including cases where the MPE property is relatively weak. Importantly, our main results use the LDM encoder which is the weakest MPE function in this ablation, suggesting that even stronger empirical gains are achievable using other MPE functions, such as the CNN autoencoder. We leave this as a future area of exploration.

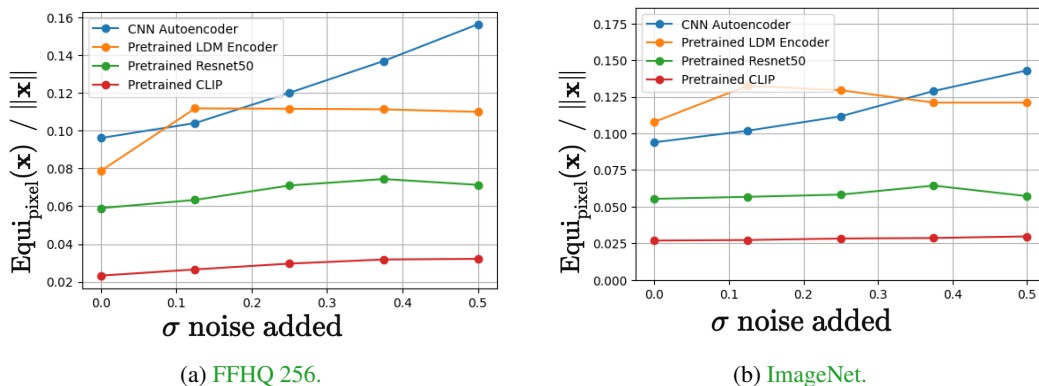

(a) FFHQ 256.                                    (b) ImageNet.

Figure 20: **Equivariance error vs. $\sigma$ noise added.** As more noise is added, equivariance error, computed with all MPE functions, increases.

## J  COMPUTING RESOURCES

We conduct experiments on two NVIDIA GeForce RTX 4090 GPUs with 24 GB of VRAM. We note that we use pre-trained models and perform inference, so not much compute is required.

## K  ASSETS

We use the publicly available code from PSLD (https://github.com/LituRout/PSLD), ReSample (https://github.com/soominkwon/resample), DPS (https://github.

Table 14: DPS superresolution with $\lambda = 0.01$ using different MPE functions.

(a) FFHQ 256.

| MPE function | PSNR | SSIM | LPIPS | FID |
|---|---|---|---|---|
| None | 23.160 (1.923) | 0.657 (0.072) | 0.193 (0.057) | 129.528 |
| LDM Encoder (FFHQ) | 26.581 (2.457) | 0.773 (0.044) | 0.120 (0.030) | 87.437 |
| CNN Autoencoder (FFHQ) | 26.866 (1.943) | 0.771 (0.044) | 0.116 (0.029) | 85.352 |
| Pretrained ResNet50 | 26.873 (1.941) | 0.771 (0.044) | 0.116 (0.029) | 85.138 |
| Pretrained CLIP | 26.860 (1.942) | 0.771 (0.044) | 0.116 (0.029) | 85.495 |

(b) ImageNet.

| MPE function | PSNR | SSIM | LPIPS | FID |
|---|---|---|---|---|
| None | 19.727 (4.292) | 0.407 (0.180) | 0.541 (0.182) | 446.829 |
| LDM Encoder (ImageNet) | 22.200 (4.295) | 0.568 (0.146) | 0.384 (0.130) | 311.636 |
| CNN Autoencoder (ImageNet) | 22.178 (4.294) | 0.568 (0.148) | 0.375 (0.125) | 312.530 |
| Pretrained ResNet50 | 22.176 (4.290) | 0.568 (0.148) | 0.375 (0.125) | 314.590 |
| Pretrained CLIP | 22.177 (4.293) | 0.568 (0.148) | 0.376 (0.125) | 313.468 |

Table 15: SITCOM motion deblurring on FFHQ 256 with $\lambda = 0.05$ using different MPE functions.

| MPE function | PSNR | SSIM | LPIPS |
|---|---|---|---|
| None | 27.670 (1.343) | 0.790 (0.031) | 0.221 (0.040) |
| LDM Encoder (FFHQ) | 28.357 (1.379) | 0.806 (0.031) | 0.200 (0.036) |
| CNN Autoencoder (FFHQ) | 28.852 (1.376) | 0.819 (0.044) | 0.193 (0.033) |
| Pretrained ResNet50 | 28.682 (1.388) | 0.811 (0.036) | 0.198 (0.036) |

com/DPS2022/diffusion-posterior-sampling), and SITCOM (https://github.
com/sjames40/SITCOM).

## L  BROADER IMPACTS

On the positive side, high-fidelity image restoration can improve downstream tasks in medical imaging, remote-sensing and environmental monitoring (e.g., denoising satellite observations to track pollution or deforestation). Likewise, accelerated PDE-solving via learned diffusion priors may enable faster, more accurate simulations for climate modeling, fluid-dynamics research, and engineering design. On the other hand, robust reconstruction methods could be misappropriated for privacy-invasive surveillance or to create deceptive imagery. We emphasize that our method does not amplify these existing risks.

## M  RESPONSIBLE RELEASE

Our approach uses only publicly available datasets and standard pre-trained diffusion models, introducing no novel dual-use or privacy risks. Consequently, no additional safeguards are required.

