# OpenReview forum: "EquiReg: Equivariance Regularized Diffusion for Inverse Problems"
_ICLR.cc/2026/Conference — Submitted to ICLR 2026_

### Official Review · Reviewer_jppv · 2025-10-28

**Soundness:** 3
**Presentation:** 2
**Contribution:** 3
**Rating:** 4
**Confidence:** 4

**Summary:**

The paper introduces EquiReg, a plug-and-play regularization framework for diffusion-based inverse problem solver by keeping sampling trajectories close to the data manifold. It leverages Manifold-Preferential Equivariant (MPE) functions, whose equivariance error is small for on-manifold and large for off-manifold data, to penalize implausible samples. EquiReg integrates seamlessly with existing diffusion solvers across pixel, latent, and PDE domains. Experiments on diverse restoration and physical modeling tasks show effectiveness.

**Strengths:**

1. The work connects geometric deep learning (equivariance) with probabilistic sampling (diffusion) in a theoretical way.
2. The work demonstrated improvements across pixel diffusion, latent diffusion, and PDE solvers, showing strong versatility.
3 The definitions of distribution-dependent equivariance and manifold-constrained equivariance are well-motivated.

**Weaknesses:**

1. No formal analysis shows why equivariance error correlates with manifold distance, and the impact of the EquiReg term on sampling stability and likelihood gradients is mostly qualitative. A detailed study (e.g., trajectory visualization, convergence proofs) would strengthen understanding.
2. The framework’s success hinges on selecting a good MPE function and appropriate symmetry group, which may be non-trivial or domain-specific. The paper provides examples but lacks systematic guidelines.
3. Although the authors mention low overhead, the additional forward passes through MPE networks could be significant for high-resolution diffusion.
4. The paper compares against general diffusion baselines but omits direct comparisons to manifold-preserving or geometry-constrained approaches
5. Figures 2 could be better visualized on how equivariance loss actually affects trajectory correction.

**Questions:**

1. Can MPE functions be learned jointly with the diffusion model instead of being pre-trained? Would that further improve alignment?
2. Whether EquiReg conflict with guidance-based conditioning (e.g., classifier-free guidance) in diffusion models?

---

> ### Author Response · Authors · 2025-11-21
> **Authors' Rebuttal for jppv (1)**
>
> We thank the reviewer for their insightful feedback and for recognizing our novel contribution in connecting the equivariance literature to probabilistic sampling, as well as the depth of our experiments demonstrating the usefulness and versatility of EquiReg across both pixel- and latent-space diffusion models in image restoration and PDE inverse problems.
>
> Below, we respond to each point in detail and summarize the revisions made in the manuscript (highlighted in green in the updated version). We also refer the reviewer to our general response outlining the newly added content in the revised paper.
>
> __W1 - Formal analysis and additional info on trajectory and regularization impact__
>
> We thank the reviewer for the opportunity to further clarify and elaborate on the theoretical contributions of our work. The revised manuscript now includes additional theoretical analysis (Appendix G). This analysis explains how an ideal regularizer, one that assigns very low values to on-manifold samples and very high values to off- or far-manifold samples, can support correct posterior sampling even when the likelihood score is approximated and introduces inaccuracies. We have added a high-level summary of this insight in the main paper, with full details provided in Appendix G.
>
> In Propositions G.1 and G.2, we present an alternative interpretation of regularization in diffusion-based inverse problems, aimed at motivating the design of our empirically evaluated regularizer. Proposition G.1 shows that the evolution of the data distribution under the reverse diffusion dynamics (eq. (3)) can be interpreted as the Wasserstein-2 gradient flow associated with minimizing the functional
>
> $\Phi(\rho,t) = \beta_{T-t}\int[ \rho\phi(\mathbf{x},t)+\frac 1 2\rho\log\rho ]d\mathbf{x}$, where $\phi(\mathbf{x},t)=-(\log p_{T-t}(\mathbf{x}|\mathbf{y})+\frac 1 4 \|\mathbf{x}\|^2)$.
>
> Please note that we are using a slightly different notation  for  in the detailed theoretical analysis. Please see our revised manuscript where we have clearly explained the notation  when used in the reversed dynamics.
>
> This analysis is grounded in optimal transport theory (Ferreira & Valencia Guevara, 2018). As stated in the paper, this perspective provides intuition to motivate the design of regularization strategies such as EquiReg. In conditional reverse diffusion for posterior sampling, most prior works compute the likelihood term
>
> $p(\mathbf{y} \mid \mathbf{x}_{T-t}) = \int{p(\mathbf{y}|\mathbf{x}_0)p_t(\mathbf{x}_0|\mathbf{x}_t)\mathrm{d}\mathbf{x}_0}$ is computed using approximation on $p_t(\mathbf{x}_0|\mathbf{x}_t)$
>
> with the isotropic Gaussian distribution where the conditional expectation $\mathbb{E}[ x_0 \mid x_{T-t} ]$ can computed via Tweedie’s formula.
>
> This approximation may incur high errors (see Figure 3). This is because the conditional expectation $\mathbb{E} [ x_0 | x_{T-t}]$ is a linear combination of all candidate $\mathbf{x}_0$.
>
> This approximation remains relatively accurate when $T - t$ is small (i.e., $\mathbf{x}_{T-t}$ stays close to the data manifold under low noise) but may incur high error for larger $T - t$. To mitigate this, we are proposing to reweight the components of the integral below to down-weighting unreliable trajectories, and amplifying the reliable ones. The resulting reweighted functional is
>
> $\tilde{\Phi}(\rho,t) = \beta_{T-t}[  {Z_t}^{-1}\int\rho(\mathbf{x}) \hat \phi_c(\mathbf{x},t) e^{\frac{\mathcal{R}(\mathbf{x})}{\hat \phi_c(\mathbf{x},t)}}\mathrm{d}\mathbf{x}+\tfrac{1}{2}\int\rho(\mathbf{x})\log\rho(\mathbf{x})d\mathbf{x}]$
>
> where $Z_t={\int{e^{\frac{\mathcal{R}(\mathbf{x})}{\hat \phi_c(\mathbf{x},t)}}\mathrm{d}\mathbf{x}}}$ is the normalizing factor, and $\mathcal{R}(\mathbf{x})$ is a regularization that is nearly zero near the data manifold and much larger elsewhere. Given such contribution reweighting, Proposition G.2 shows that the evolution of the probability distribution driven by the regularized reverse dynamics
>
> $\mathrm{d}\mathbf{x} = [-\tfrac{\beta_t}{2}\mathbf{x}\mathrm{d}t - \beta_t\nabla_{\mathbf{x}_t}(\log{p_t(\mathbf{x}_t)} + \log{ \int p(\mathbf{y}|\mathbf{x}_0)\tilde p_t(\mathbf{x}_0|\mathbf{x}_t)\mathrm{d}\mathbf{x}_0} - \mathcal{R}(\mathbf{x}_t))]\mathrm{d}t + \sqrt{\beta_t}\mathrm{d}\mathbf{{\bar w}}$
>
> is an approximation of the Wasserstein-2 gradient flow associated with minimizing $\tilde{\Phi}$ which we defined above. Continue in the next comment ...

---

> > ### Author Response · Authors · 2025-11-21
> > **Authors' Rebuttal for jppv (2)**
> >
> > The above analysis provides an intuitive understanding on the role of regularizing a conditional diffusion model. This analysis shows: a) unreliable likelihood estimates should be down-weighted, b) reliable estimates should be amplified, c) and doing so is equivalent to adding a regularization term to the posterior sampling dynamics.
> >
> > This formalism motivates the ideal property of a useful regularizer: “it should produce high error on off/far-manifold samples and low error on on/near-manifold samples.” This formalism is general and not tied to a specific type of function.
> > In the paper, we instantiate this ideal regularizer using equivariance.
> >
> > We introduce (Definition 3.1) a distribution-dependent equivariance error and define a class of functions whose equivariance error is: low on samples from the data manifold, and high on off-manifold samples. We call these Manifold-Preferential Equivariant (MPE) functions.
> >
> > Importantly, we do not assume any theorem asserting that equivariance error must characterize the manifold. Instead: a) our analysis provides the desired property of a regularizer, b) MPE functions are defined to satisfy this property, and c) the MPE functions that we have constructed are empirically shown to satisfy this property in practical settings.
> >
> > For stability, we thank the reviewer for their comments. We clarify that our stability notion relates to the failure rate of solutions. For example, in Figure 6b, we have quantified the histogram of PSLD across the tested examples. The figure shows that PSLD has high density around low PSNR values, and EquiReg has been able to successfully reduce low PSNR (failed case). We now include visualization (Figures 16 and 17), showing that while solutions from DPS and PSLD contain artifacts, EquiReg has mitigated the artifacts. We call this EquiReg provides better stability of solutions (not giving you a solution that has artifacts (being off the data manifold). We have now modified the wording on stability in the paper to further elaborate.
> >
> > Continue in the next comment ...

---

> ### Author Response · Authors · 2025-11-21
> **Authors' Rebuttal for jppv (3)**
>
> - __W3 Efficiency of EquiReg__
>
> We would like to emphasize that we already report runtime for a subset of our experiments.
> First, we show that EquiReg combined with SITCOM achieves lower runtime despite the additional gradient computation introduced by the regularizer. This is because EquiReg implicitly accelerates convergence: by guiding the sampler toward better solutions more quickly, it triggers SITCOM’s early-stopping criterion at significantly fewer iterations.
>
> Second, Table 2 demonstrates that EquiReg remains effective even when applied at a reduced frequency (e.g., every 2, 5, or 10 steps), further lowering computation.
>
> To address efficiency on higher-dimensional data, we note that the choice of MPE function is flexible. As detailed in our systematic guidelines below in our response W2, EquiReg works with a wide variety of functions (including lightweight neural networks). Practitioners may therefore select a less complex MPE function for high-dimensional tasks to reduce computational overhead.
>
> Lastly, we have included additional experiments showing that in the presence of computational limitation, one can use EquiReg to further reduce the number of DDIM steps of diffusion and still arrive at acceptable performance (Now Table 8 in the Supplementary). Particularly, we compare DPS and Equi-DPS  on super-resolution tasks when DDIM steps are reduced from 1000 to 900, to 750, and 500. The results, below, show the outperformance of EquiReg in reduced DDIM regime.
>
> **DPS**
>
> |Steps | PSNR$\uparrow$ | SSIM$\uparrow$ | LPIPS$\downarrow$ | FID$\downarrow$|
> |-------|--------|---------|----------|--------|
> |500  | 13.89 | 0.0937 | 0.955 | 417.07|
> |750  | 21.77 | 0.540 | 0.254 | 153.74|
> |900  | 22.97 | 0.628  | 0.201 | 148.03|
> |1000 | 22.99 | 0.649  | 0.201 | 135.71|
>
> **Equi-DPS**
>
> |Steps | PSNR$\uparrow$ | SSIM$\uparrow$ | LPIPS$\downarrow$ | FID$\downarrow$|
> |-------|--------|---------|----------|--------|
> |500  | 20.61 | 0.366  | 0.500 | 238.51 |
> |750  | 25.60 | 0.704  | 0.160 | 110.89 |
> |900  | 26.52 | 0.755  | 0.126 | 88.46  |
> |1000 | 26.73 | 0.767  | 0.120 | 88.00  |
>
> Finally, we will soon add more experiments on higher-dimensional problems to further support EquiReg’s applicability and efficiency in high-dimensional settings.
>
> - __W4 Comparison to manifold-preserving or geometry-constrained approaches__
>
> We respectfully disagree with the reviewer’s statement. Our paper *does* include direct comparisons to manifold-preserving and geometry-constrained methods. Specifically, Table 4 (EquiReg for diffusion models on FFHQ) compares EquiReg against __MPGD-AE__, __MCG__, and __DiffStateGrad__, all of which explicitly incorporate manifold-preserving, projection-based, or geometry-constrained mechanisms. In the revised manuscript, we now elaborate that these baselines are manifold-preserving and geometry-constrained methods.

---

> > ### Author Response · Authors · 2025-11-21
> > **Authors' Rebuttal for jppv (4)**
> >
> > __W5 Figures 2__
> >
> > We thank the reviewer for their suggestion. We have revised Figure 2 to more clearly illustrate the regularization effect and how the Equi-loss varies with distance from the data manifold when an MPE function is used. The original Figures 2b–c are now moved to the new Figure 3. We hope that the new visualizations and our additional explanations, help clarify EquiReg. We are happy to incorporate further suggestions the reviewer may have.
> >
> > __Q1 Joint training of MPE function with diffusion model?__
> >
> > We thank the reviewer for this thoughtful question. A central goal of our work is practicality and broad applicability. This is why we focus on solving inverse problems using unconditionally pre-trained diffusion models, without requiring retraining or architectural modification. Consequently, our experiments primarily evaluate MPE functions that are already trained, allowing us to test and compare their MPE properties directly (see our new ablation study on MPE evaluation).
> >
> > While joint training is not the focus of our current paper, we note that MPE-like behavior can naturally emerge when networks are trained with data augmentation or when the data exhibit inherent symmetries (see future comments for our new ablation studies relating to MPE). In principle, one could also train a diffusion model jointly with an MPE function, for example, in a latent diffusion model (LDM) where the autoencoder could serve as the MPE module. This direction is promising, and we have now pointed it out in the Conclusion section for future work.
> >
> > Importantly, EquiReg is designed to remain decoupled from the diffusion model itself. The MPE function is modular and can be swapped, reused, or even applied outside of diffusion models entirely. This modularity is a key advantage of EquiReg, enabling it to act as a general-purpose equivariance-based regularizer.
> >
> > __Q2 EquiReg with guidance-based conditioning?__
> >
> > We thank the reviewer for raising this question. Indeed, EquiReg is fully compatible with guidance-based conditioning. In fact, we have already applied EquiReg to classifier-free guidance (CFG) in our experiments. Figure 1e illustrates the effect of EquiReg when combined with DreamSampler for text-to-image generation. We now complement this with additional examples and analysis in Section A of the appendix, where we also highlight an implicit acceleration of image generation when EquiReg is applied.
> >
> > In summary, a key advantage of EquiReg is its plug-and-play nature, enabling seamless integration with a wide range of  guidance-based diffusion models.
> >
> > Continue in the next comment ...

---

> > > ### Author Response · Authors · 2025-11-21
> > > **Authors' Rebuttal for jppv (5)**
> > >
> > > __Choice of group symmetry__
> > >
> > > We now include guidelines on how to choose or find symmetry groups for a variety of application domains. For this, we include proper citation to the equivariance literature and how it is being used in vision, molecular generation, chemistry, etc. For example, SE(3)-equivariance (i.e., translation and rotations) are useful for 3D dynamics which can be applied to molecular simulations [A].
> > >
> > > Moreover, rotations and reflections are explored in reinforcement learning literature to have equivariance agents. Finally, we note that our contributions are not in isolation, it can be combined with growing literature on meta-learning and automatic symmetry discovery to learn symmetry groups and actions from data [C,D,E,F]. We have now added a brief note in Section 3 and refer reader/reviewer to more information in Appendix.
> > >
> > >
> > > [A] Xu, M., et al. (2024) Equivariant Graph Neural Operator for Modeling 3D Dynamics. ICML.
> > >
> > > [B] Park, J. Y., et al. (2025). Approximate Equivariance in Reinforcement Learning. AISTATS.
> > >
> > > [C] Zhou, A., et al. (2021) Meta-learning Symmetries by Reparameterization. ICLR.
> > >
> > > [D] Dehmamy, N., Walters, R., Liu, Y., Wang, D., & Yu, R. (2021). Automatic symmetry discovery with lie algebra convolutional network. NeurIPS.
> > >
> > > [E] Quessard, R., et al. (2020). Learning disentangled representations and group structure of dynamical environments. NeurIPS.
> > >
> > > [F] Mohapatra, et al. (2025). Symmetry-Driven Discovery of Dynamical Variables in Molecular Simulations. ICML.

---

> ### Author Response · Authors · 2025-11-21
> **Authors' Rebuttal for jppv (6)**
>
> __Systematic construction of MPE functions and emergence of MPE__
>
> We thank the reviewer for this observation. We have expanded Section 3 to clarify how MPE functions arise naturally in practice and how they can be systematically constructed (see also Section I of Appendix).
>
> First, we formalize two common mechanisms through which MPEs emerge:
>
> (i) Training-time data augmentations such as flips or rotations applied to a neural network (the mapping function).
>
> (ii) Inherent domain symmetries present in many settings, including 3D dynamics, molecular systems, and PDEs.
>
> To demonstrate that these mechanisms yield practical and effective MPE functions, we added several new experiments. We have conducted an ablation study that examines both the emergence of MPE properties in different functions (neural networks) and the effect of using these functions within EquiReg across identical tasks. Our study evaluates:
>
> - The MPE behavior of several publicly available pre-trained networks, as well as a CNN-based autoencoder (AE) that we trained on the data distribution of interest using flip augmentations.
>
> - The performance of EquiReg when each of these MPE functions is used on the same inverse problem settings, two datasets (FFHQ and ImageNet), two diffusion-based solvers (DPS and SITCOM), and two tasks (super-resolution and motion deblurring).
>
> Below we summarize our findings, with full details and tables included in the revised manuscript. We evaluated four function classes:
>
> - Pre-trained encoder of latent diffusion models (LDMs): the same choice used in our main experiments.
> - A CNN autoencoder trained by us using flip (for FFHQ) and rotations (for ImageNet) augmentations on the relevant training distribution (test images excluded).
> - Pre-trained ResNet-50, trained on natural images.
> - Pre-trained CLIP (OpenAI).
>
> Our results show:
>
> - All networks exhibit MPE-like behavior: their equivariance error under flip (for FFHQ) and rotation (for ImageNet) symmetry increases as the noise level of the input increases (i.e., as data moves farther from the manifold).
>
> - Some have lower and some higher MPE properties. Notably, the CNN AE trained on the true data distribution shows the strongest MPE property, with equivariance error rising sharply as samples deviate from the manifold, aligning with our systematic guidelines for constructing MPE functions (Section 3). This is the case where the trained data distribution (data manifold) is fully aligned with the data we are solving the inverse problem for (e.g., trained on ImageNet train set, and tested on ImageNet test set).
>
> The results, which we have outlined in detail below, show that a) MPE property has emerged in popular pre-trained functions out there; hence, it is not rare to happen, and b) EquiReg is effective and has improved the performance of diffusion-based inverse problem solvers under all the considered MPE functions. Figure 20 (see updated manuscript) visualizes these trends clearly. See below for tabular results of Figure 20.
>
> _Equi Loss, Evaluated for varying noise levels on 100 images from FFHQ 256_
> | MPE Function   | sigma=0      | sigma=0.125   | sigma=0.25    | sigma=0.375   | sigma=0.5    |
> |--------------------------------|--------------|---------------|---------------|---------------|--------------|
> | CNN Autoencoder  | 0.10 (0.03)  | 0.10 (0.03)   | 0.12 (0.03)   | 0.14 (0.04)   | 0.16 (0.04)  |
> | Pretrained LDM Encoder  | 0.08 (0.02)  | 0.11 (0.03)   | 0.11 (0.03)   | 0.11 (0.03)   | 0.11 (0.03)  |
> | Pretrained Resnet50 | 0.06 (0.02)  | 0.06 (0.02)   | 0.07 (0.02)   | 0.07 (0.02)   | 0.07 (0.02)  |
> | Pretrained CLIP          | 0.02 (0.01)  | 0.03 (0.01)   | 0.03 (0.01)   | 0.03 (0.01)   | 0.03 (0.01)  |
>
> _Equi Loss, Evaluated for varying noise levels on 100 images from Imagenet_
> | MPE Function    | sigma=0      | sigma=0.125   | sigma=0.25    | sigma=0.375   | sigma=0.5    |
> |-----------------------------------|--------------|---------------|---------------|---------------|--------------|
> | CNN Autoencoder  | 0.09 (0.03)  | 0.10 (0.03)   | 0.12 (0.03)   | 0.13 (0.02)   | 0.14 (0.03)  |
> | Pretrained LDM Encoder | 0.11 (0.03)  | 0.13 (0.02)   | 0.13 (0.03)   | 0.13 (0.02)   | 0.12 (0.02)  |
> | Pretrained Resnet50   | 0.06 (0.01)  | 0.06 (0.01)   | 0.06 (0.02)   | 0.06 (0.02)   | 0.06 (0.01)  |
> | Pretrained CLIP      | 0.03 (0.01)  | 0.03 (0.01)   | 0.03 (0.01)   | 0.03 (0.00)   | 0.03 (0.00)  |
>
> We have also added new visualizations for the encoder MPE function used in our experiments (Figure 15). These results show that its equivariance loss is low for clean images and increases as images are corrupted with various noise levels (Figure 15a). We additionally include a detailed visualization of the loss landscape for a representative example (Figure 15b).

---

> > ### Author Response · Authors · 2025-11-21
> > **Authors' Rebuttal for jppv (7)**
> >
> > _Effectiveness of EquiReg with different MPE functions_
> >
> > We then applied each of these MPE functions within the EquiReg framework on the same inverse problem setups. Across all settings, EquiReg consistently improved reconstruction quality relative to the baseline (no regularization, i.e., None). This demonstrates that:
> >
> > a) MPE properties commonly emerge in widely used pre-trained networks, and thus EquiReg is easy to deploy in practice; and
> >
> > b) EquiReg remains effective across all tested MPE choices, confirming the robustness of the proposed regularization framework.
> >
> > We refer the reviewer to our Appendix for more details (see Tables 14-15). Lastly, we note that in our main paper we used the LDM encoder as our MPE function, even though it exhibits the weakest MPE properties among all functions considered in the new ablation. This suggests that our reported numerical results could be further improved by selecting a stronger MPE function. We leave this exploration to future practical work.
> >
> > _Lambda = 0.01. DPS on FFHQ 256, Superresolution:_
> >
> > | MPE function                    |   PSNR  (PSNR_std)  |  SSIM  (SSIM_std) |  NMSE (NMSE_std) | LPIPS  (LPIPS_std) |   FID   |
> > |:---------------------------|--------:|---------:|-------:|---------:|-------:|
> > | None                       | 23.1595 (1.9227) | 0.6565 ( 0.0718)  | 0.0244 ( 0.0186) | 0.1927 ( 0.0570) | 129.5284 |
> > | LDM Encoder (FFHQ)         | 26.5811 (2.4567) | 0.7730 (0.0438) | 0.0113 ( 0.0094) | 0.1200 (0.0297) |  87.4366 |
> > | CNN Autoencoder (FFHQ)  | 26.8659 (1.9429)  | 0.7713 (0.0437) | 0.0107 (0.0096) | 0.1162 (0.0287) |  85.3520 |
> > | Pretrained Resnet50        | 26.8733 (1.9406) | 0.7713 (0.0437) | 0.0106 (0.0096) | 0.1160 (0.0287) |  85.1376 |
> > | Pretrained CLIP            | 26.8600  (1.9424) | 0.7711 (0.0437) | 0.0107 ( 0.0096) | 0.1162 (0.0288) |  85.4950 |
> >
> > _Lambda = 0.01. DPS on Imagenet, Superresolution:_
> >
> > | MPE function                      |   PSNR  (PSNR_std) |  SSIM  (SSIM_std) |  NMSE  (NMSE_std) | LPIPS  (LPIPS_std) |   FID   |
> > |:----------------------------|--------:|---------:|-------:|---------:|-------:|
> > | None                         | 19.7267 (4.2915) | 0.4074 ( 0.1802) | 0.0699 (0.0350) | 0.5412 ( 0.1821) | 446.8292 |
> > | LDM Encoder (Imagenet)   | 22.2001 (4.2950) | 0.5681 ( 0.1464) | 0.0390 ( 0.0209) | 0.3844 ( 0.1296) | 311.6355 |
> > | CNN Autoencoder (Imagenet) | 22.1777 (4.2943) | 0.5677 (0.1478) | 0.0392 (0.0210) | 0.3753 (0.1250) | 312.5301 |
> > | Pretrained Resnet50  | 22.1763 ( 4.2901) | 0.5677 (0.1477) | 0.0392 ( 0.0210) | 0.3753 (0.1248) | 314.5898 |
> > | Pretrained CLIP         | 22.1774 ( 4.2927) | 0.5676 ( 0.1477) | 0.0392 (0.0210) | 0.3757 ( 0.1248) | 313.4681 |
> >
> >
> > _Lambda = 0.05. SITCOM on FFHQ 256, Motion Deblur:_
> >
> > | MPE function      | psnr_mean (psnr_std) | ssim_mean (ssim_std) | lpips_mean ( lpips_std) | time_sec_mean (time_sec_std) |
> > |:---------------------------|----------:|---------:|----------:|---------:|
> > | None                       |   27.6695 (1.3434) |   0.7903  ( 0.0305) |    0.2212  (0.0399)  |       23.3611 ( 0.8936)  |
> > | LDM Encoder (FFHQ)    |   28.3565 (1.3787) |   0.8058  ( 0.0314) |    0.2001  ( 0.0358)  |       24.1549 (0.5565) |
> > | CNN Autoencoder (FFHQ) |   28.8522 (1.3758) |   0.8188  (0.0437) |    0.1932  (0.0327)  |    24.1831 (0.7417) |
> > | Pretrained Resnet50     |   28.6819 (1.3884) |   0.8109  (0.0292) |    0.1977  ( 0.0361)  |       24.0940 (0.6554) |
> >
> > We emphasize that our focus on pre-trained MPEs is motivated by practical considerations. In many applications (e.g., vision), suitable MPE functions already exist within standard pre-trained models. Thus, users typically do not need to train a specialized network for EquiReg, just as diffusion-based inverse problem solvers rely on pre-trained unconditional diffusion priors. For example, the encoders/decoders used in latent diffusion models already satisfy the MPE property.
> >
> > We also note that while our work is the first to formalize distribution-dependent equivariance functions and the manifold-preferential equivariance (MPE) property, prior studies have observed emergent MPE-like behavior for OOD detection (Zhou 2022; Kaur et al. 2022, 2023), as cited.
> >
> > In summary, EquiReg does not require a theoretical emergence guarantee; it only requires a function whose equivariance error discriminates on-manifold from off-manifold samples. Our propositions motivate this general requirement, and our expanded empirical results show that MPE functions are easy to obtain and generalize well in practice.
> >
> > We hope the new experiments clarify that MPEs commonly emerge in models trained with standard augmentations or inherent data symmetries. We have added these results to Section I and now explicitly note in the discussion that a full theoretical characterization of MPE emergence is an interesting and valuable direction for future work.

---

> > > ### Author Response · Authors · 2025-11-21
> > > **Authors' Rebuttal for jppv (8)**
> > >
> > > To conclude, we thank the reviewer for their comments. We found their comments helpful and addressed them to further improve our paper submission. We hope that our responses and new experiments (see our revised manuscript and our general response for a clear explanation of all new content and how it complements the original submission) adequately address the concerns raised. Accordingly, we kindly ask the reviewer to consider updating their score in light of the revised paper.
> > >
> > > We look forward to hearing from the reviewers regarding any remaining questions or suggestions on how we may further improve our paper.

---

### Official Review · Reviewer_2i6e · 2025-10-30

**Soundness:** 2
**Presentation:** 2
**Contribution:** 1
**Rating:** 2
**Confidence:** 4

**Summary:**

This paper proposes an explicit regularizer, called EquiReg, for solving inverse problems in imaging with diffusion models. The method is framework-agnostic. Because natural images have equivariance, the paper suggests adding the regularizer's gradient when differentiating the likelihood. They apply this to ReSample, DPS, and PSLD and get better LPIPS, FID, and PSNR.

**Strengths:**

1. The story is not too complex, so the message is clear.

2. It is model-agnostic, so many users can adopt EquiReg for their own framework.

3. It generally improves fidelity metrics like PSNR and LPIPS.

**Weaknesses:**

This paper might be sacrificing the true strength of diffusion models for inverse problems. The reason diffusion models are used (like in DPS - Diffusion Posterior **Sampling**) is that they are **Samplers**, not mean-estimators. The advantage is that they can sample many different solutions that are all good. This paper has no consideration for this. However, this paper just says EquiReg helps getting good PSNR and SSIM. This is obvious. Equivariance is a somewhat classical regularization method. Of course, if one add any classical regularizer, the PSNR and SSIM will improve. This is a common idea. For example, a paper rejected at ICLR last year (https://openreview.net/forum?id=GQnR7L6SmA) used Total Variation (TV) with ADMM, another classical regularization, and it worked well, but I think that paper shares the same problem with this one: DIVERSITY. Diversity is a core philosophy of using diffusion models for inverse problems. The author of DPS has already pointed out that subtle PSNR improvement is not the thing for this problem (https://x.com/hyungjin_chung/status/1788861058309902633). Without showing a proper consideration for diversity, it is hard to prove that this simple regularizer is truly beneficial.

Minor weaknesses:
- L34: There is a citation error for Charles W Groetsch and CW Groetsch.

- It is better importing figures as pdf, rather than png or jpg.

**Questions:**

In L89, 97, this paper points out relying on the isotropic Gaussian assumption is one of limitations of prior work. Could you please elaborate why is it so, and how this paper addressed it?

---

> ### Author Response · Authors · 2025-11-20
> **Authors' Rebuttal (1)**
>
> We thank the reviewer for their comments. We appreciate the reviewer’s recognition of the clarity of our main message and the model-agnostic nature of EquiReg, which enables wide applicability to a broad range of guidance-based diffusion inverse problem solvers. We also thank the reviewer for noting that, across all reported numerical results, EquiReg consistently improves standard metrics used in image restoration tasks.
>
> We have addressed the reviewer’s concerns and revised the manuscript accordingly. Below, we clarify the key points about EquiReg and the contributions of our paper. All modifications in the revised manuscript are highlighted in green, and we refer the reviewer to our general response for a summary of the newly added content.
> We hope the following addresses the reviewer’s concerns.
>
> __Sampling capabilities of diffusion models with EquiReg__
>
> We agree with the reviewer that diffusion models are powerful samplers that can generate multiple plausible solutions to an inverse problem (i.e., posterior sampling). However, we respectfully disagree with the implication that EquiReg diminishes this capability. Diffusion models equipped with EquiReg still sample multiple solutions. EquiReg does not collapse the posterior or enforce a point estimate.
>
> Instead, the purpose of EquiReg is to regularize the sampling trajectory by filtering out undesirable or low-probability solutions that arise due to the widely used isotropic Gaussian approximation in conditional diffusion models for the likelihood score (see new Figure 3). This approximation is well known to distort the true posterior, EquiReg directly targets this failure mode by discouraging such off-manifold trajectories.
>
> We have now revised the beginning of Section 3 to better articulate this: regularizing the reverse posterior dynamics can be interpreted as reweighting the sampling contributions, down-weighting unreliable estimates and amplifying reliable ones. Importantly, this does not restrict the diversity of posterior samples. To further illustrate this, we added additional experiments measuring sample diversity (see our response to the diversity comment below).
>
> __Metrics considered in the experiment__
>
> We appreciate the reviewer’s concern regarding evaluation metrics, and we provide clarifications below.
> Our experiments were designed to allow clear and fair comparison to prior diffusion-based inverse problem methods. This required using the same experimental settings (e.g., datasets, tasks, model architectures, and evaluation metrics) adopted by prior work. Thus, we report PSNR, SSIM, FID, and LPIPS because these are the standard metrics used throughout the literature. For example:
>
> [1] DPS (ICLR) uses FID, LPIPS, and SSIM
>
> [2] PSLD (NeurIPS) uses PSNR, SSIM, FID, and LPIPS
>
> [3] MCG (NeurIPS) uses PSNR, SSIM, FID, and LPIPS
>
> [4] SITCOM (ICML) uses PSNR, SSIM, and LPIPS
>
> [5] ReSample (ICLR) uses PSNR, SSIM, and LPIPS
>
> [6] DiffStateGrad (ICLR) uses PSNR, SSIM, and LPIPS
>
> [7] MPGD (ICLR) uses PSNR and LPIPS
>
> In addition, while we include PSNR and SSIM, our manuscript already emphasizes perceptual metrics (LPIPS and FID), where EquiReg shows the largest improvements. We have now strengthened this point in the text and added new visualizations illustrating how EquiReg resolves blur and artifacts present in baseline reconstructions (see Figures 16 and 17).
> Overall, our evaluation follows established practice in the community and uses well-accepted quantitative metrics for assessing the quality of inverse problem solutions. We hope this clarifies our choices, and we welcome any further suggestions.
>
> [1] Chung, H., Kim, J., Mccann, M. T., Klasky, M. L., & Ye, J. C. (2023) Diffusion Posterior Sampling for General Noisy Inverse Problems. ICLR.
>
> [2] Rout, L., Raoof, N., Daras, G., Caramanis, C., Dimakis, A., & Shakkottai, S. (2023). Solving linear inverse problems provably via posterior sampling with latent diffusion models. NeurIPS.
>
> [3] Chung, H., Sim, B., Ryu, D., & Ye, J. C. (2022). Improving diffusion models for inverse problems using manifold constraints. NeurIPS.
>
> [4] Alkhouri, I., Liang, S., Huang, C. H., Dai, J., Qu, Q., Ravishankar, S., & Wang, R. (2025) SITCOM: Step-wise Triple-Consistent Diffusion Sampling For Inverse Problems. ICML.
>
> [5] ong, B., Kwon, S. M., Zhang, Z., Hu, X., Qu, Q., & Shen, L. (2024) Solving Inverse Problems with Latent Diffusion Models via Hard Data Consistency. ICLR.
>
> [6] Zirvi, R., Tolooshams, B., & Anandkumar, A. Diffusion State-Guided Projected Gradient for Inverse Problems. ICLR.
>
> [7] He, Y., Murata, N., Lai, C. H., Takida, Y., Uesaka, T., Kim, D., ... & Ermon, S. Manifold Preserving Guided Diffusion. ICLR.

---

> > ### Author Response · Authors · 2025-11-20
> > **Authors' Rebuttal (2)**
> >
> > __EquiReg as a “classical” regularization?__
> >
> > We thank the reviewer for their perspective, but we respectfully and strongly disagree with the claim that “equivariance is a classical regularization method” and that “adding any classical regularizer will improve PSNR and SSIM.” We clarify and contextualize our contributions below.
> >
> > - First, it is not clear what the reviewer means by “classical,” but in the inverse problems literature, classical regularizers typically refer to sparsity, total variation, low-rankness, and similar handcrafted priors. The field has moved away from such priors because they do not capture the rich statistics and complex structure of natural signals. Modern state-of-the-art inverse problem solvers overwhelmingly rely on learned priors, first GANs, and now unconditional diffusion models, which outperform classical methods by a large margin. Under this context, EquiReg which uses an MPE function is not “just adding a classical regularizer.”
> >
> > - Second, the role of EquiReg is not “adding any regularizer,” but addressing a very specific and well-known failure mode in conditional diffusion models: the intractability of the likelihood score. Prior works approximate $p(\mathbf{y}|\mathbf{x}_t) = \int p(\mathbf{y} | \mathbf{x}_0) p(\mathbf{x}_0 |\mathbf{x}_t) d\mathbf{x}_0$ using an isotropic Gaussian approximation for the posterior $p(\mathbf{x}_0 \mid \mathbf{x}_t)$. This approximation can mislead the sampler into low‐density, off-manifold regions (see Figure 3). EquiReg is designed precisely to correct this issue by penalizing unreliable states, hence, trajectories during posterior sampling.
> >
> > - Third, our contributions are not classical at all:
> > We formalize distribution-dependent equivariant functions, which to our knowledge do not exist in the prior literature.
> > We define manifold-preferential equivariant (MPE) functions, whose equivariance error is low for on-manifold samples and high for off-manifold samples. We propose a new regularization mechanism for conditional diffusion models that is directly tied to the posterior sampling dynamics, not to classical reconstruction optimization.
> >
> > - Fourth, EquiReg is, to our knowledge, the first method to analyze how regularization affects reverse posterior sampling trajectories. Through a Wasserstein gradient flow interpretation (see our detailed response to Reviewer 86xx), we show that regularization of the reverse dynamics corresponds to a principled reweighting of posterior trajectories, i.e., down-weighting unreliable off-manifold trajectories and favoring reliable ones. This is fundamentally different from classical regularization, which does not operate on sampling trajectories nor on distributional dynamics.
> > Finally, regarding metrics, we refer the reviewer to our earlier response. Our evaluation follows established practice across NeurIPS, ICLR, and ICML papers on diffusion-based inverse problems, and our strongest gains appear in perceptual metrics (LPIPS and FID), not only PSNR/SSIM.
> >
> > We hope this clarifies our contributions which are grounded and aligned with modern diffusion-based inverse problem research.
> >
> > __Comparing EquiReg with an ICLR rejected paper on TV and ADMM__
> >
> > We thank the reviewer for pointing us to this paper. We were not aware of this work, as it is not on arXiv and was rejected from ICLR 2025. After carefully reviewing the submission, we acknowledge that both this paper and EquiReg incorporate regularization during the measurement-guidance stage of conditional diffusion models. However, there are several key distinctions:
> >
> > Our paper introduces an interpretation based on Wasserstein gradient flow, providing a principled understanding of what it means to regularize a sampling trajectory at the level of probability distributions. This theoretical formulation of posterior-sampling regularization is, to our knowledge, unique to our work and is not present in the referenced paper.
> > We define and characterize a new class of functions, MPEs, and use them to construct a learned, data-dependent regularization loss. This learned MPE-based regularizer does not fall under classical regularization approaches such as TV, which is the focus of the referenced paper.
> >
> > One of the positive aspects highlighted by reviewers is that our method uses a known mechanism (regularization) which is applicable as a plug-and-play to any guidance-based diffusion model. It establishes a clear explanation on what it means to regularize posterior sampling trajectory for inverse problems. This principled grounding distinguishes our contribution from prior work relying on classical regularizers.
> >
> > Finally, we refer the reviewer to our diversity experiment where we highlight EquiReg’s preservation of diversity unlike restrictive classical regularizers.

---

> ### Author Response · Authors · 2025-11-20
> **Authors' Rebuttal (3)**
>
> __Diversity Experiments__
>
> We thank the reviewer for raising this important concern about diversity preservation. We agree that the sampling capability of diffusion models is a core advantage for inverse problems, and we have conducted comprehensive experiments to evaluate whether EquiReg maintains this property.
>
> - _New Experimental Results (Appendix Section D):_
>
> We have added a dedicated section in the appendix with extensive diversity analysis. We generate K=10 posterior samples for 20 test images across three inverse problems (box inpainting, Gaussian deblurring, $4\times$ super-resolution) and measure diversity using two complementary metrics: Intra-LPIPS (perceptual diversity) and Pixel-Std (spatial diversity).
>
> Our results demonstrate that Equi-DPS achieves favorable fidelity-diversity trade-offs:
>
> 1. Box inpainting: Equi-DPS improves both fidelity AND diversity simultaneously (see Table 11), showing that equivariance regularization can actually enhance rather than restrict sampling behavior.
>
> 2. Super-resolution: We observe substantial improvements in both fidelity and diversity, demonstrating dramatic diversity gains alongside quality improvements.
>
> 3. Gaussian deblurring: Equi-DPS achieves 15-20% better fidelity while retaining 80-85% of baseline diversity, a modest but justified trade-off.
>
> - _Diversity vs Difficulty Analysis:_
>
> We further investigate diversity scaling by varying box inpainting mask size from $128\times128$ to $192\times192$ pixels (Figure 18). Results show that diversity metrics increase linearly with task difficulty, demonstrating that Equi-DPS naturally expands sampling as problems become more ill-posed rather than artificially constraining solutions. This linear relationship indicates healthy, predictable posterior sampling behavior across the difficulty spectrum.
>
> - _Qualitative Validation:_
>
> Figure 7 shows $K=4$ diverse posterior samples for two test images, with visible variations in facial features, expressions, and eye gaze, confirming our quantitative measurements. These visual results demonstrate that Equi-DPS generates genuinely diverse reconstructions rather than collapsing to a single mode.
> Important Clarification on Diversity vs Fidelity:
>
> We respectfully note that diversity without fidelity is not useful for image reconstruction tasks. Reconstructions with artifacts may exhibit high diversity but provide little practical value. Our results show that Equi-DPS maintains meaningful diversity (Intra-LPIPS values ranging from 0.092 to 0.187 across tasks, which is substantially above mode collapse levels of <0.05) while significantly improving reconstruction quality. This represents a principled balance rather than sacrificing one for the other.
>
> - _Addressing the TV-ADMM Comparison:_
>
> Unlike classical regularizers that may suppress diversity through aggressive smoothing or sparsity constraints, equivariance regularization encodes geometric structure in the data manifold. Our experiments demonstrate this distinction: in 2 out of 3 tasks, we improve both fidelity and diversity simultaneously, which would not occur with regularizers that simply constrain solution space.
>
> We believe these comprehensive diversity experiments (detailed in Appendix Section D) address the reviewer's concern and demonstrate that EquiReg preserves the sampling philosophy of diffusion models while adding beneficial geometric priors.
>
> __Minor__
>
> We have fixed the typo on one of the citations and figures are now in pdf format.

---

> > ### Author Response · Authors · 2025-11-20
> > **Authors' Rebuttal (4)**
> >
> > __Intractability challenge of diffusion models for solving inverse problems__
> >
> > We have additional information on this challenge in the preliminary and related works section on diffusion models for inverse problems. We have expanded the likelihood score on L89 in original submission to
> >
> > $\nabla_{\mathbf{x}_t} \log{p_t(\mathbf{y}|\mathbf{x}_t)} = $
> >
> > $\nabla_{\mathbf{x}_t} \log{\int{p(\mathbf{y}|\mathbf{x}_0)p_t(\mathbf{x}_0|\mathbf{x}_t)\mathrm{d}\mathbf{x}_0}}$ for clarity. We are happy to move more details into the introduction to increase clarity. Please see below.
> >
> > The main challenge of using unconditionally trained diffusion models as learned priors for solving inverse problems is that the likelihood score  $\nabla_{\mathbf{x}_t}\log{p_t(\mathbf{y}|\mathbf{x}_t)}$
> > needed during posterior sampling is unknown. Hence, training-free solvers differ in how they approximate $p_t(\mathbf{y}|\mathbf{x}_t)$. Since
> >
> > $p_t(\mathbf{y}|\mathbf{x}_t) = \int p(\mathbf{y}|\mathbf{x}_0)p_t(\mathbf{x}_0|\mathbf{x}_t)\mathrm{d}\mathbf{x}_0$
> >
> > the common and popular choice is to approximate
> >
> > $p_t (\mathbf{x}_0| \mathbf{x}_t)$
> >
> > by an isotropic Gaussian $\mathcal{N}(\mathbf{x}_{0|t}, {r_t}^2 I)$, which is employed by a variety of diffusion-based approaches (Chung et al., 2023; Song et al., 2023b; Zhu et al., 2023; Zhang et al., 2025a).
> >
> > With an optimal denoising score, prior work computes the posterior mean
> >
> > $\mathbf{x}_{0|t} \coloneqq \mathbb{E}[\mathbf{x}_0|\mathbf{x}_t]$
> >
> > using Tweedie’s formula. However, for complex or multimodal distributions, $p_t (\mathbf{x}_0|\mathbf{x}_t)$ may not be concentrated around its mean, leading to off-manifold solutions (see our visualization in Figure 3).
> >
> > While the majority of the diffusion-based inverse problem literature has aimed to provide better estimation of the score for $p_t(\mathbf{y}|\mathbf{x}_t)$, the most popular approaches, especially for large-scale problems, including latent diffusion models, still employ the isotropic Gaussian approximation.
> >
> > In this paper, we propose: in the presence of such approximations and inaccuracies, let us regularize the whole posterior trajectory to mitigate off-manifold directions caused by the aforementioned approximation. We have now revised Figure 2 visualization to better communicate the above message.

---

> > ### Comment · Reviewer_2i6e · 2025-11-23
> > **Comments for Authors' Rebuttal (3)**
> >
> > I appreciate the added diversity analysis. However, my concern about sacrificing diversity for fidelity remains because the evidence lacks generality:
> >
> > 1. Evidence is Biased: Qualitative results (Figure 7, Figure 18) focus solely on box inpainting, an inherently ill-posed problem where high diversity is expected. This doesn't prove generalized diversity preservation.
> >
> > 2. Metrics Show Reduction: Quantitative metrics in Table 11 show Equi-DPS reduces diversity (down to 80-85%) for Gaussian deblurring problem, contradicting the claim of overall enhancement.
> >
> > 3. No Qualitative Confirmation: There are no qualitative/visual examples shown for complex tasks like Gaussian Deblurring or Super-Resolution, making it difficult to assess the meaningfulness of the reported quantitative gains in diversity for those tasks.

---

> > > ### Author Response · Authors · 2025-11-24
> > > **Additional Experiments and Discussion on Diversity (1/2)**
> > >
> > > We thank the reviewer for their continued engagement. We have revised the manuscript to include the discussion below, and we have added additional visualizations and new diversity experiments for Gaussian deblurring and super-resolution (see Section D, including Figures 18 and 19). We hope these additions help clarify the intent of our method and the interpretation of diversity in diffusion-based inverse problems.
> > >
> > > __Clarification on the role of diversity in diffusion-based inverse problems__
> > >
> > > We believe there is a key conceptual misunderstanding regarding the goal of solving inverse problems with diffusion models. In the Bayesian setting, the objective is not to maximize diversity per se; rather, it is to sample from the high-probability regions of the posterior distribution. True diversity emerges only insofar as the posterior itself admits meaningful variability.
> > >
> > > In practice, diversity-related concerns in inverse problems arise when a method suffers from mode collapse, i.e., the sampler becomes biased and fails to explore multiple plausible modes of the posterior. Thus, the relevant question is whether a method properly explores the posterior rather than whether it maximizes diversity in an unconstrained sense.
> > >
> > > Because closed-form posteriors are unavailable for real image restoration tasks, the standard practice in the diffusion inverse-problem literature is to evaluate diversity through variation among plausible reconstructions consistent with the measurement, without collapsing to a single solution. This is the notion of “diversity” our work adopts.
> > >
> > > __Clarification on the purpose of EquiReg__
> > >
> > > Hence, given the goal of posterior sampling, EquiReg is not designed to maximize diversity for its own sake. Its objective is to incorporate data-inherent geometric structure (equivariance) to guide sampling toward high-probability regions of the posterior; from a manifold perspective, this corresponds to likelihood-informed guidance toward the data manifold, similar in spirit to manifold-constrained gradient descent (MPGD [1]), but achieved in a principled and rigorously defined manner. We again clarify that diversity arises naturally from the ill-posedness of the inverse problem; it is a consequence of posterior uncertainty, not the goal of the regularizer.
> > >
> > > Hence, the relevant question is whether EquiReg collapses the posterior, dramatically reducing diversity. Our experiments show that it does not.
> > >
> > > [1] He, Y., Murata, N., Lai, C. H., Takida, Y., Uesaka, T., Kim, D., ... & Ermon, S. (2024) Manifold Preserving Guided Diffusion. ICLR.
> > >
> > > - __1. Box Inpainting as Evidence__
> > >
> > > The reviewer argues that box inpainting is too ill-posed to provide general evidence. This is fundamentally circular: diversity is meaningful only for ill-posed inverse problems with multimodal and broad posteriors. Well-posed problems inherently exhibit low posterior uncertainty and therefore low diversity for any method.
> > >
> > > We found the box inpainting task a suitable benchmark for diversity precisely because it exposes mode-collapse behavior. In this setting, our results demonstrate that EquiReg does not collapse the posterior and, in several cases, maintains or increases posterior variation.

---

> ### Author Response · Authors · 2025-11-20
> **Authors' Rebuttal (5)**
>
> To conclude, we hope that our responses, clarifications, and new experiments have improved the presentation of our contributions and addressed the reviewer’s concerns. Please see the revised manuscript, with changes marked in green, as well as our general response, which details how each addition complements the original submission.
>
> We hope that these revisions resolve the reviewer’s concerns, and we kindly ask the reviewer to consider updating their score accordingly. Should any questions remain, we would be happy to provide further clarification. We look forward to hearing from the reviewer.

---

> ### Comment · Reviewer_2i6e · 2025-11-23
> **Comment for Authors' Rebuttal (2)**
>
> I appreciate for the detailed rebuttal.
>
> I agree that applying this to Diffusion Inverse Problems is novel. However, the central idea is still old: Equation (5) in the main paper is just a differentiable regularizer that enforces equivariance on intermediate and final outputs. Calling it 'Manifold-Preferential' doesn't change that the core mechanism is not fundamentally new to differentiable systems.

---

> > ### Author Response · Authors · 2025-11-24
> > **Regularization and Naming of Manifold-Preferential Equivariance**
> >
> > __Discussion on regularization__
> >
> > We thank the reviewer for their continued engagement and follow-up. We would like to clarify two points raised:
> >
> > __1) On the comment that the central idea is “old”__
> >
> > We appreciate the reviewer’s observation that regularization is an “old”, well-established concept. We do agree that there exists a long and rich literature on regularization in inverse problems and optimization. However, we respectfully disagree that the use of an established principle diminishes the contribution. Many foundational tools in modern machine learning are based on old and well-established concepts. For example, diffusion models and score matching are closely related to Tweedie’s formula, also originated and already discussed in 1956 [1]. Hence, the sole age of a concept does not reduce its fundamental relevance; rather, their reinterpretation and application in new contexts have repeatedly driven major advances in modern machine learning.
> >
> > [1] Herbert Ellis Robbins. An empirical Bayes approach to statistics. In Proc. Third Berkeley Symposium on Mathematical Statistics, pages 157–163, 1956.
> >
> > We clarify that our contribution is __not__ the generic notion of “adding a regularizer,” but rather:
> >
> > - identifying what kind of regularizer is theoretically desirable for posterior sampling under likelihood approximations,
> >
> > - providing a Wasserstein gradient-flow interpretation of conditional diffusion regularization, where the desirable regularization is down-weighting unreliable trajectories (off-manifold ones).
> >
> > - formalizing distribution-dependent equivariant functions and manifold-preferential equivariance (MPE) as a principled way to operationalize this ideal regularizer, and
> >
> > - demonstrating that such functions arise naturally in pretrained networks (see our newly added experiments on this) and effectively improve solving inverse problems via EquiReg.
> >
> > __2) On the naming and role of “manifold-preferential equivariance”__
> >
> > We clarify that the term manifold-preferential equivariance (MPE) provides a precise name for a property that is essential for the regularizer to work as motivated by our analysis: low equivariance error on (or near) the data manifold and high error off the manifold. This naming aligns with the formal properties we define and supports clear communication of why certain functions are suitable for EquiReg.
> >
> > If the reviewer finds the terminology unclear or prefers an alternative name, we would be happy to revise it. Our intention is to make the concept transparent rather than introduce terminology for its own sake.
> >
> > ___
> > We hope the above addresses the reviewer’s concern. We will provide our follow-up response regarding the diversity comment shortly (tomorrow).

---

> ### Author Response · Authors · 2025-11-24
> **Additional Experiments and Discussion on Diversity (2/2)**
>
> - __2. Diversity “Reduction” vs Trade-Off__
>
> The reviewer suggests our results contradict a claim of “overall enhancement.” We clarify that we never make such a claimed uniform diversity improvement; we stated that EquiReg produced favorable fidelity-diversity trade-offs, which is precisely what the numbers show.
>
> We were transparent about settings where diversity slightly decreases. At first glance, one may find it not desirable. However, this decrease is expected in the presence of inaccuracies and the challenge of posterior sampling: any regularizer that discourages off-manifold exploration will necessarily constrain the solution space to some degree (as it has eliminated non-plausible artifacted solutions). The key question is whether this constraint eliminates multimodality.
>
> Our results indicate it does not. For Gaussian deblurring, EquiReg retains 80–85% of baseline diversity (Intra-LPIPS: 0.092 vs 0.114) while improving fidelity by 15–20%. This demonstrates that:
>
> a) the posterior remains multimodal,
>
> b) geometric constraints do not destroy exploration, and
>
> c) the fidelity–diversity trade-off is beneficial.
>
> Importantly, Gaussian deblurring with a mild blur kernel (σ = 3) is relatively well-posed. The posterior is narrow by construction and contains limited high-frequency ambiguity. In this case, no method should exhibit large diversity here. Expecting EquiReg to increase diversity on a well-posed task would imply injecting artificial uncertainty, which would actively harm reconstruction fidelity. Thus, the modest reduction is expected and consistent with the underlying posterior geometry.
>
> Finally, we highlight that EquiReg improves both fidelity and diversity on 2 of the 3 tasks, an encouraging outcome that is uncommon given the general behavior of classical regularizers. Hand-crafted regularizers such as TV and $\ell_1$ may suppress diversity by shrinking solutions toward simple structures. By contrast, EquiReg leverages data-dependent regularization that captures the richness and structural complexity of the underlying data manifold, enabling it to preserve manifold-consistent variability while suppressing implausible samples.
>
> - __3. Diversity vs Fidelity for Reconstruction__
>
> High diversity without fidelity is not meaningful for posterior sampling. A method that samples the entire solution space, including low-probability and artifacted regions, may score well on diversity but fail to provide useful reconstructions.
> Equi-DPS avoids this failure mode: it maintains meaningful diversity (Intra-LPIPS 0.092–0.187, well above collapse levels <0.05) while reducing artifacts and improving perceptual quality.
>
> In the experiments conducted during the rebuttal, our goal was to demonstrate clearly that EquiReg preserves meaningful diversity, reflecting the posterior uncertainty, rather than unstructured or unconstrained variability.
>
> - __4. Additional Qualitative and Scaling Analysis__
>
> To address the reviewer’s request for broader evidence, we have added:
>
> - Qualitative diversity examples for Gaussian deblurring (Figure 19a), showing differences in facial expressions and accessories (i.e., earrings in first test image),
>
> - Qualitative examples for super-resolution (Figure 19b), showing differences in facial features (i.e., teeth in the first test image, eye color and eyelashes in the second test image),
>
> - Diversity-scaling experiments for Gaussian deblurring (σ = 3, 6, 9), showing proportional increase with blur strength (Figure 18b), and
>
> - Diversity-scaling experiments for super-resolution (4×, 8×, 16×), also showing proportional increase with difficulty (Figure 18c).
>
> These results demonstrate that diversity scales predictably with problem difficulty across all three tasks, confirming that EquiReg does not suppress natural posterior variation.
>
> We hope the reviewer finds them helpful.

---

### Official Review · Reviewer_Y4hA · 2025-10-31

**Soundness:** 2
**Presentation:** 2
**Contribution:** 3
**Rating:** 6
**Confidence:** 3

**Summary:**

This paper proposes EquiReg, an equivariance-based regularization method for diffusion inverse problems. By penalizing samples that break learned symmetries, it keeps diffusion trajectories closer to the data manifold and stabilizes posterior sampling.

**Strengths:**

- Introduces equivariance-based regularization as a proxy for manifold consistency, connecting geometric symmetry with probabilistic sampling in a fresh way.
- The method is architecture-agnostic and simple to implement that can be directly added to existing diffusion frameworks (DPS, PSLD, SITCOM) without retraining.

**Weaknesses:**

- The link between low equivariance error and on-manifold behavior is intuitive but not rigorously proven.
- The approach assumes an explicit group $G$ (e.g., rotation, reflection), which may not exist or be meaningful for all tasks.
- The method relies on pre-trained MPE encoders, but how to systematically obtain or generalize them is not well discussed.
- The paper frequently refers to an Appendix for details of tasks, proofs, and additional experimental results, but the Appendix is not provided. As a result, several important aspects cannot be verified. This omission limits the paper’s clarity.

**Questions:**

- See weaknesses
- The statement “MPE can emerge when functions are trained with symmetry-preserving mechanisms such as data augmentation” is somewhat ambiguous. Almost all modern pretrained models are trained with some form of data augmentation, yet clearly not all of them behave as MPEs. How do the authors determine whether a given model qualifies as an MPE?

---

> ### Author Response · Authors · 2025-11-21
> **Authors' Rebuttal for Y4hA (1)**
>
> We thank the reviewer for their thoughtful feedback and recognizing the novelty of our approach on connecting geometric symmetry with probabilistic sampling for inverse problems. The reviewer has also appreciated Equireg’s model-agnostic and wide applicability to existing diffusion frameworks without retraining.
>
> Below, we respond to each point in detail and summarize the revisions made in the manuscript (marked in green in the updated version). We also refer the reviewer to our general response summarizing the new content added to our revised paper.
>
> __Link between equivariance error and on-manifold behaviour__
>
> We thank the reviewer for raising this important point. Below, we clarify the relationship between equivariance error and manifold behaviour within our formalism.
>
> __1. Clarifying the source of off-manifold errors in conditional diffusion.__
>
> We now explain more clearly with supporting analysis in the appendix that existing inverse-problem solvers compute the likelihood term
>
> $p(\mathbf{y} \mid \mathbf{x}_{T-t}) = \int{p(\mathbf{y}|\mathbf{x}_0)p_t(\mathbf{x}_0|\mathbf{x}_t)\mathrm{d}\mathbf{x}_0}$
>
> using approximation on $p_t(\mathbf{x}_0|\mathbf{x}_t)$ with the isotropic Gaussian distribution with the conditional expectation
>
> $\mathbb{E}[ x_0 \mid x_{T-t} ]$, computed via Tweedie’s formula (Please note that we are using a slightly different notation $T-t$ for $t$ in the detailed theoretical analysis. Please see our revised manuscript where we have clearly explained the notations when used in the reversed dynamics).
>
> This approximation can incur significant error (new Figure 3) because $\mathbb{E}[ x_0 \mid x_{T-t} ]$ is a linear combination of all candidate $\mathbf{x}_0$, and thus may lie off the data manifold. The approximation is accurate when T - t is small (low noise), but becomes unreliable when T - t is large.
>
> __2. Our theoretical analysis: what regularization is doing.__
>
> In the revised appendix (Propositions G2–G1), we provide a gradient-flow interpretation of the conditional reverse dynamics. This analysis shows: a) unreliable trajectories should be down-weighted, b) reliable trajectories should be amplified, c) and doing so is equivalent to adding a regularization term to the posterior sampling dynamics.
>
> This formalism identifies the ideal property of a useful regularizer: “it should produce high error on off/far-manifold samples and low error on on/near-manifold samples.” This formalism is general and not tied to a specific type of function.
>
> __3. How equivariance enters: MPE functions as one practical realization.__
>
> In the paper, we instantiate this ideal regularizer using equivariance.
>
> We introduce (Definition 3.1) a distribution-dependent equivariance error and define a class of functions whose equivariance error is: low on samples from the data manifold, and high on off-manifold samples. We call these Manifold-Preferential Equivariant (MPE) functions.
>
> Importantly, we do not assume any theorem asserting that equivariance error must characterize the manifold. Instead: a) our analysis provides the desired property of a regularizer, b) MPE functions are defined to satisfy this property, and c) the MPE functions that we have constructed are empirically shown to satisfy this property in practical settings.
>
> Continue in the next comment ...

---

> > ### Author Response · Authors · 2025-11-21
> > **Authors' Rebuttal for Y4hA (2)**
> >
> > __4. Empirical evidence that MPE behaviour emerges naturally__ (see our response below for __Systematic construction of MPE functions and emergence of MPE__).
> >
> > __5. Clarifying the scope of our claim.__
> >
> > We have revised the manuscript to emphasize that EquiReg is a regularization framework, not a manifold projection method. EquiReg penalizes states that deviate from symmetry-preserving regions; when an MPE function is used, these regions empirically align with the data manifold.
> >
> > We hope this clarifies the connection between equivariance error and manifold behaviour within our framework. We are happy to refine the explanation further if the reviewer has additional suggestions.
> >
> > __Choice of group symmetries__
> >
> > We thank the reviewer for raising this concern. We agree that identifying an appropriate symmetry group may be challenging in some application domains. To address this, we have added practical guidelines in the revised manuscript on how symmetry groups are typically chosen in different fields, along with references to the extensive equivariance literature.
> >
> > For example, SE(3)-equivariance (translations and rotations) is a natural choice for 3D physical systems, which reflects the structure of our physical world, and is widely used in molecular simulation and 3D dynamics modeling [A]. Likewise, rotations and reflections have been used effectively in reinforcement learning to construct equivariant agents that exploit environment symmetries [B]. Similar domain-informed choices exist in computer vision, chemistry, and physics.
> >
> > Importantly, our contributions are not limited to cases where a symmetry group is known a priori. EquiReg can be combined with the growing body of work on meta-learning and automatic symmetry discovery, which aims to infer symmetry groups or group actions directly from data [C, D, E, F]. We now incorporate this perspective into Section 3, and provide a more detailed discussion in the Appendix.
> >
> > We are happy to include more citations or elaborate more.
> >
> > [A] Xu, M., et al. (2024) Equivariant Graph Neural Operator for Modeling 3D Dynamics. ICML.
> >
> > [B] Park, J. Y., et al. (2025). Approximate Equivariance in Reinforcement Learning. AISTATS.
> >
> > [C] Zhou, A., et al. (2021) Meta-learning Symmetries by Reparameterization. ICLR.
> >
> > [D] Dehmamy, N., Walters, R., Liu, Y., Wang, D., & Yu, R. (2021). Automatic symmetry discovery with lie algebra convolutional network. NeurIPS.
> >
> > [E] Quessard, R., et al. (2020). Learning disentangled representations and group structure of dynamical environments. NeurIPS.
> >
> > [F] Mohapatra, et al. (2025). Symmetry-Driven Discovery of Dynamical Variables in Molecular Simulations. ICML.

---

> ### Author Response · Authors · 2025-11-21
> **Authors' Rebuttal for Y4hA (3)**
>
> __Systematic construction of MPE functions and emergence of MPE__
>
> We thank the reviewer for this observation. We have expanded Section 3 to clarify how MPE functions arise naturally in practice and how they can be systematically constructed (see also Section I)
> First, we formalize two common mechanisms through which MPEs emerge:
>
> (i) Training-time data augmentations such as flips or rotations applied to a neural network (the mapping function).
>
> (ii) Inherent domain symmetries present in many settings, including 3D dynamics, molecular systems, and PDEs.
>
> To demonstrate that these mechanisms yield practical and effective MPE functions, we added several new experiments. We have conducted an ablation study that examines both the emergence of MPE properties in different functions (neural networks) and the effect of using these functions within EquiReg across identical tasks. Our study evaluates:
>
> 1. The MPE behavior of several publicly available pre-trained networks, as well as a CNN-based autoencoder (AE) that we trained on the data distribution of interest using flip augmentations.
>
> 2. The performance of EquiReg when each of these MPE functions is used on the same inverse problem settings, two datasets (FFHQ and ImageNet), two diffusion-based solvers (DPS and SITCOM), and two tasks (super-resolution and motion deblurring).
>
> Below we summarize our findings, with full details and tables included in the revised manuscript. We evaluated four function classes:
>
> - Pre-trained encoder of latent diffusion models (LDMs): the same choice used in our main experiments.
> - A CNN autoencoder trained by us using flip (for FFHQ) and rotations (for ImageNet) augmentations on the relevant training distribution (test images excluded).
> - Pre-trained ResNet-50, trained on natural images.
> - Pre-trained CLIP (OpenAI).
>
> Our results show:
>
> - All networks exhibit MPE-like behavior: their equivariance error under flip (for FFHQ) and rotation (for ImageNet) symmetry increases as the noise level of the input increases (i.e., as data moves farther from the manifold).
> - Some have lower and some higher MPE properties. Notably, the CNN AE trained on the true data distribution shows the strongest MPE property, with equivariance error rising sharply as samples deviate from the manifold, aligning with our systematic guidelines for constructing MPE functions (Section 3). This is the case where the trained data distribution (data manifold) is fully aligned with the data we are solving the inverse problem for (e.g., trained on ImageNet train set, and tested on ImageNet test set).
>
> The results, which we have outlined in detail below, show that a) MPE property has emerged in popular pre-trained functions out there; hence, it is not rare to happen, and b) EquiReg is effective and has improved the performance of diffusion-based inverse problem solvers under all the considered MPE functions. Figure 20 (see updated manuscript) visualizes these trends clearly. See below for tabular results of Figure 20.
>
> _Equi Loss, Evaluated for varying noise levels on 100 images from FFHQ 256_
> | MPE Function   | sigma=0      | sigma=0.125   | sigma=0.25    | sigma=0.375   | sigma=0.5    |
> |--------------------------------|--------------|---------------|---------------|---------------|--------------|
> | CNN Autoencoder  | 0.10 (0.03)  | 0.10 (0.03)   | 0.12 (0.03)   | 0.14 (0.04)   | 0.16 (0.04)  |
> | Pretrained LDM Encoder  | 0.08 (0.02)  | 0.11 (0.03)   | 0.11 (0.03)   | 0.11 (0.03)   | 0.11 (0.03)  |
> | Pretrained Resnet50 | 0.06 (0.02)  | 0.06 (0.02)   | 0.07 (0.02)   | 0.07 (0.02)   | 0.07 (0.02)  |
> | Pretrained CLIP          | 0.02 (0.01)  | 0.03 (0.01)   | 0.03 (0.01)   | 0.03 (0.01)   | 0.03 (0.01)  |
>
> _Equi Loss, Evaluated for varying noise levels on 100 images from Imagenet_
> | MPE Function    | sigma=0      | sigma=0.125   | sigma=0.25    | sigma=0.375   | sigma=0.5    |
> |-----------------------------------|--------------|---------------|---------------|---------------|--------------|
> | CNN Autoencoder  | 0.09 (0.03)  | 0.10 (0.03)   | 0.12 (0.03)   | 0.13 (0.02)   | 0.14 (0.03)  |
> | Pretrained LDM Encoder | 0.11 (0.03)  | 0.13 (0.02)   | 0.13 (0.03)   | 0.13 (0.02)   | 0.12 (0.02)  |
> | Pretrained Resnet50   | 0.06 (0.01)  | 0.06 (0.01)   | 0.06 (0.02)   | 0.06 (0.02)   | 0.06 (0.01)  |
> | Pretrained CLIP      | 0.03 (0.01)  | 0.03 (0.01)   | 0.03 (0.01)   | 0.03 (0.00)   | 0.03 (0.00)  |
>
> We have also added new visualizations for the encoder MPE function used in our experiments (Figure 15). These results show that its equivariance loss is low for clean images and increases as images are corrupted with various noise levels (Figure 15a). We additionally include a detailed visualization of the loss landscape for a representative example (Figure 15b).

---

> > ### Author Response · Authors · 2025-11-21
> > **Authors' Rebuttal for Y4hA (4)**
> >
> > _Effectiveness of EquiReg with different MPE functions_
> >
> > We then applied each of these MPE functions within the EquiReg framework on the same inverse problem setups. Across all settings, EquiReg consistently improved reconstruction quality relative to the baseline (no regularization, i.e., None). This demonstrates that:
> >
> > a) MPE properties commonly emerge in widely used pre-trained networks, and thus EquiReg is easy to deploy in practice; and
> >
> > b) EquiReg remains effective across all tested MPE choices, confirming the robustness of the proposed regularization framework.
> >
> > We refer the reviewer to our Appendix for more details (see Tables 14-15). Lastly, we note that in our main paper we used the LDM encoder as our MPE function, even though it exhibits the weakest MPE properties among all functions considered in the new ablation. This suggests that our reported numerical results could be further improved by selecting a stronger MPE function. We leave this exploration to future practical work.
> >
> > _Lambda = 0.01. DPS on FFHQ 256, Superresolution:_
> >
> > | MPE function                    |   PSNR  (PSNR_std)  |  SSIM  (SSIM_std) |  NMSE (NMSE_std) | LPIPS  (LPIPS_std) |   FID   |
> > |:---------------------------|--------:|---------:|-------:|---------:|-------:|
> > | None                       | 23.1595 (1.9227) | 0.6565 ( 0.0718)  | 0.0244 ( 0.0186) | 0.1927 ( 0.0570) | 129.5284 |
> > | LDM Encoder (FFHQ)         | 26.5811 (2.4567) | 0.7730 (0.0438) | 0.0113 ( 0.0094) | 0.1200 (0.0297) |  87.4366 |
> > | CNN Autoencoder (FFHQ)  | 26.8659 (1.9429)  | 0.7713 (0.0437) | 0.0107 (0.0096) | 0.1162 (0.0287) |  85.3520 |
> > | Pretrained Resnet50        | 26.8733 (1.9406) | 0.7713 (0.0437) | 0.0106 (0.0096) | 0.1160 (0.0287) |  85.1376 |
> > | Pretrained CLIP            | 26.8600  (1.9424) | 0.7711 (0.0437) | 0.0107 ( 0.0096) | 0.1162 (0.0288) |  85.4950 |
> >
> > _Lambda = 0.01. DPS on Imagenet, Superresolution:_
> >
> > | MPE function                      |   PSNR  (PSNR_std) |  SSIM  (SSIM_std) |  NMSE  (NMSE_std) | LPIPS  (LPIPS_std) |   FID   |
> > |:----------------------------|--------:|---------:|-------:|---------:|-------:|
> > | None                         | 19.7267 (4.2915) | 0.4074 ( 0.1802) | 0.0699 (0.0350) | 0.5412 ( 0.1821) | 446.8292 |
> > | LDM Encoder (Imagenet)   | 22.2001 (4.2950) | 0.5681 ( 0.1464) | 0.0390 ( 0.0209) | 0.3844 ( 0.1296) | 311.6355 |
> > | CNN Autoencoder (Imagenet) | 22.1777 (4.2943) | 0.5677 (0.1478) | 0.0392 (0.0210) | 0.3753 (0.1250) | 312.5301 |
> > | Pretrained Resnet50  | 22.1763 ( 4.2901) | 0.5677 (0.1477) | 0.0392 ( 0.0210) | 0.3753 (0.1248) | 314.5898 |
> > | Pretrained CLIP         | 22.1774 ( 4.2927) | 0.5676 ( 0.1477) | 0.0392 (0.0210) | 0.3757 ( 0.1248) | 313.4681 |
> >
> >
> > _Lambda = 0.05. SITCOM on FFHQ 256, Motion Deblur:_
> >
> > | MPE function      | psnr_mean (psnr_std) | ssim_mean (ssim_std) | lpips_mean ( lpips_std) | time_sec_mean (time_sec_std) |
> > |:---------------------------|----------:|---------:|----------:|---------:|
> > | None                       |   27.6695 (1.3434) |   0.7903  ( 0.0305) |    0.2212  (0.0399)  |       23.3611 ( 0.8936)  |
> > | LDM Encoder (FFHQ)    |   28.3565 (1.3787) |   0.8058  ( 0.0314) |    0.2001  ( 0.0358)  |       24.1549 (0.5565) |
> > | CNN Autoencoder (FFHQ) |   28.8522 (1.3758) |   0.8188  (0.0437) |    0.1932  (0.0327)  |    24.1831 (0.7417) |
> > | Pretrained Resnet50     |   28.6819 (1.3884) |   0.8109  (0.0292) |    0.1977  ( 0.0361)  |       24.0940 (0.6554) |
> >
> > We emphasize that our focus on pre-trained MPEs is motivated by practical considerations. In many applications (e.g., vision), suitable MPE functions already exist within standard pre-trained models. Thus, users typically do not need to train a specialized network for EquiReg, just as diffusion-based inverse problem solvers rely on pre-trained unconditional diffusion priors. For example, the encoders/decoders used in latent diffusion models already satisfy the MPE property.
> >
> > We also note that while our work is the first to formalize distribution-dependent equivariance functions and the manifold-preferential equivariance (MPE) property, prior studies have observed emergent MPE-like behavior for OOD detection (Zhou 2022; Kaur et al. 2022, 2023), as cited.
> >
> > In summary, EquiReg does not require a theoretical emergence guarantee; it only requires a function whose equivariance error discriminates on-manifold from off-manifold samples. Our propositions motivate this general requirement, and our expanded empirical results show that MPE functions are easy to obtain and generalize well in practice.
> >
> > We hope the new experiments clarify that MPEs commonly emerge in models trained with standard augmentations or inherent data symmetries. We have added these results to Section I and now explicitly note in the discussion that a full theoretical characterization of MPE emergence is an interesting and valuable direction for future work.

---

> > > ### Author Response · Authors · 2025-11-21
> > > **Authors' Rebuttal for Y4hA (5)**
> > >
> > > __Appendix__
> > >
> > > We thank the reviewer for raising this point. We have added extended complementary results in the appendix, and we refer the reviewer to our general comment summarizing all newly included material and explaining how each addition complements the experiments in the main paper. Given the additional page allowance, we are happy to move any content that the reviewer finds necessary or helpful into the main manuscript.
> > > ___
> > >
> > >
> > > To conclude, we thank the reviewer for their helpful comments. We hope that our responses and new experiments (see our revised manuscript and our general response for a clear explanation of all new content and how it complements the original submission) adequately address the concerns raised. Accordingly, we kindly ask the reviewer to consider updating their score in light of the revised paper.
> > >
> > > We look forward to hearing from the reviewers regarding any remaining questions or suggestions on how we may further improve our paper.

---

### Official Review · Reviewer_86xx · 2025-10-31

**Soundness:** 4
**Presentation:** 2
**Contribution:** 4
**Rating:** 4
**Confidence:** 4

**Summary:**

The paper proposed Equivariance Regularized (EquiReg) diffusion, a plug-and-play framework to solve Bayesian inverse problems with pre-trained diffusion prior. In inference time, equivariance loss is incorporated as reward gradient guidance, penalizing reconstructions that deviate from the data manifold. Experimental results across various tasks and existing inverse problem solver demonstrates the effectiveness of EquiReg.

**Strengths:**

1. Novelty: The paper proposed a novel reward gradient guidance that leads to on-manifold sampling. Instead of digging deep into the underlying data distribution, the reward discriminates on-manifold samples from off-manifold samples simply using symmetry arising from data itself or the training process.
2. Effectiveness: Numerical experiments demonstrates that EquiReg loss could be easily incorporated into gradient guidance-based diffusion inverse problem solvers, and achieve better performances.

**Weaknesses:**

1. Though it is mentioned in the main paper that various details are deferred to the appendix, the appendix is not included in the submission, which substantially affects readability.
2. Theoretical insights of the EquiReg loss is not sufficiently explored. It remains unknown how EquiReg loss could affect generation consistency. The authors mentioned some insights through Wasserstein gradient flow. It will be helpful if it could be further discussed.
3. Though it is natural to use symmetry when data is inherently symmetric, there seems to be little intuition in the paper on how to choose a symmetry group and the corresponding MPE functions in general. Yet the choice could be crucial in sampling quality.
4. Symmetry group size too small among all experiments. It remains unclear whether a large symmetry group, or even an infinite group such as SO(3), could affect the algorithm.

**Questions:**

My questions follow from the Weaknesses.
1. Is it possible to solve box inpainting so that reviewers could recover the appendix? Will be happy to see this paper published if it is complete.
2. Is it possible to demonstrate the effectiveness of EquiReg loss through low-dimensional experiments?
3. Can the authors compare different MPE functions and symmetry groups on the same task and same diffusion inverse problem solver?
4. Is it possible to experiment with a symmetry group with large cardinality?

---

> ### Author Response · Authors · 2025-11-21
> **Authors' Rebuttal for 86xx (1)**
>
> We thank the reviewer for their thoughtful comments and recognizing the novelty of our approach to focus on simply using approximate and distribution dependent equivariance symmetries to discriminate on-manifolds from off-manifold samples. The reviewer has also appreciated the effectiveness of EquiReg in being integrated seamlessly into existing guidance-based diffusion inverse problem solvers
>
> Below, we respond to each point in detail and summarize the revisions made in the manuscript (marked in green in the updated version).
>
> - __W1/Q1 - Appendix__
>
> The revised manuscript now includes new complementary material added to the appendix and the main paper. As outlined in our “Summary of New Content and Experiments Addressing Reviewers’ Comments,” each addition is designed to complement and clarify the original submission. We are also open to the reviewer’s suggestion to move selected elements into the main paper to improve clarity and readability.
>
> - __W2 - Theoretical Insights__
>
> We thank the reviewer for the opportunity to clarify and elaborate on the theoretical contribution of our work. The revised manuscript now includes an expanded theoretical analysis in Appendix G. The propositions provide an alternative interpretation of regularization in diffusion-based inverse problems and motivate the design of our empirically evaluated regularizer.
>
> Proposition G.1 shows that the evolution of the data distribution under the reverse diffusion dynamics (Eq. (3)) can be interpreted as the Wasserstein-2 gradient flow associated with minimizing the functional
>
> $\Phi(\rho,t) = \beta_{T-t}\int[ \rho\phi(\mathbf{x},t)+\frac 1 2\rho\log\rho ]d\mathbf{x}$, where $\phi(\mathbf{x},t)=-(\log p_{T-t}(\mathbf{x}|\mathbf{y})+\frac 1 4 \|\mathbf{x}\|^2)$.
>
> These derivations are grounded in optimal transport theory (Ferreira & Valencia Guevara, 2018). As stated in the paper, the goal of this perspective serves as an intuitive interpretation to motivate the design of regularization strategies such as EquiReg.
>
> In this conditional reverse dynamics for posterior sampling, we have explained that a central limitation in prior conditional diffusion samplers is likelihood score $p(\mathbf{y} \mid \mathbf{x}_{T-t}) = \int p(\mathbf{y}|\mathbf{x}_0)p_t(\mathbf{x}_0|\mathbf{x}_t)\mathrm{d}\mathbf{x}_0$ is computed using approximation on $p_t(\mathbf{x}_0|\mathbf{x}_t)$ with the isotropic Gaussian distribution where the conditional expectation
>
> $\mathbb{E} [x_0 | x_{T-t}]$
>
> can computed via Tweedie’s formula (Please note that we are using a slightly different notation $T-t$ for $t$ in the detailed theoretical analysis. Please see our revised manuscript where we have clearly explained the notation $T-t$ when used in the reversed dynamics). This approximation can incur significant error (new Figure 3) because $ \mathbb{E} [x_0 | x_{T-t}]$
>
> is a linear combination of all candidate $\mathbf{x}_0$, and thus may lie off the data manifold. The approximation is accurate when T - t is small (low noise), but becomes unreliable when T - t is large.
>
> To mitigate this, we introduce a reweighting mechanism that down-weights unreliable contributions and amplifies reliable ones within the functional optimization. The resulting reweighted functional is
>
> $\tilde{\Phi}(\rho,t) = \beta_{T-t}[  {Z_t}^{-1}\int\rho(\mathbf{x}) \hat \phi_c(\mathbf{x},t) e^{\frac{\mathcal{R}(\mathbf{x})}{\hat \phi_c(\mathbf{x},t)}} \mathrm{d}\mathbf{x}+\tfrac{1}{2}\int\rho(\mathbf{x})\log\rho(\mathbf{x})d\mathbf{x}]$
>
> where $Z_t={\int{e^{\frac{\mathcal{R}(\mathbf{x})}{\hat \phi_c(\mathbf{x},t)}}\mathrm{d}\mathbf{x}}}$ is the normalizing constant, and $\mathcal{R}(\mathbf{x})$ is a regularizer that is nearly zero near the data manifold and larger elsewhere.
>
> Proposition G.2 shows that the dynamics
>
> $\mathrm{d}\mathbf{x} = [-\tfrac{\beta_t}{2}\mathbf{x}\mathrm{d}t - \beta_t\nabla_{\mathbf{x}_t}(\log{p_t(\mathbf{x}_t)} + \log{\int p(\mathbf{y}|\mathbf{x}_0)\tilde p_t(\mathbf{x}_0|\mathbf{x}_t)\mathrm{d}\mathbf{x}_0} - \mathcal{R}(\mathbf{x}_t))]\mathrm{d}t + \sqrt{\beta_t}\mathrm{d}\mathbf{\bar w}$
>
> approximate the Wasserstein-2 gradient flow minimizing $\tilde{\Phi}$.
>
> This analysis offers intuition for the role of regularization in conditional diffusion models, i.e., an effective regularizer should assign low error to on-manifold (good) samples and high error to off- or far-manifold (bad) samples. Our MPE formulation is directly motivated by this insight. In summary, the propositions suggest that iterative regularization on reverse conditional diffusion models while operated based on a given available sample at time $t$, can be interpreted as reweighting the posterior sampling trajectories based on their on/off manifold reliability.
>
> We have included an expanded explanation in Appendix G and are happy to move more of this material into the main paper should the reviewer prefer.

---

> > ### Author Response · Authors · 2025-11-21
> > **Authors' Rebuttal for 86xx (2)**
> >
> > __W3 - Choice of symmetry group__
> >
> > We acknowledge that choosing a symmetry group may be a challenge depending on the application domain. We now include guidelines on how to choose or find symmetry groups for a variety of application domains. For this, we include proper citation to the equivariance literature and how it is being used in vision, molecular generation, chemistry, etc. For example, SE(3)-equivariance (i.e., translation and rotations) are useful for 3D dynamics which can be applied to molecular simulations [A].
> >
> > Moreover, rotations and reflections are explored in reinforcement learning literature to have equivariance agents [B]. Finally, we note that our contributions are not in isolation, it can be combined with growing literature on meta-learning and automatic symmetry discovery to learn symmetry groups and actions from data [C,D,E,F]. We have now added a brief note in Section 3 and refer reader/reviewer to more information in Appendix.
> >
> > [A] Xu, M., et al. (2024) Equivariant Graph Neural Operator for Modeling 3D Dynamics. ICML.
> >
> > [B] Park, J. Y., et al. (2025). Approximate Equivariance in Reinforcement Learning. AISTATS.
> >
> > [C] Zhou, A., et al. (2021) Meta-learning Symmetries by Reparameterization. ICLR.
> >
> > [D] Dehmamy, N., Walters, R., Liu, Y., Wang, D., & Yu, R. (2021). Automatic symmetry discovery with lie algebra convolutional network. NeurIPS.
> >
> > [E] Quessard, R., et al. (2020). Learning disentangled representations and group structure of dynamical environments. NeurIPS.
> >
> > [F] Mohapatra, et al. (2025). Symmetry-Driven Discovery of Dynamical Variables in Molecular Simulations. ICML.
> >
> > Continue in the next comment …

---

> ### Author Response · Authors · 2025-11-21
> **Authors' Rebuttal for 86xx (3)**
>
> __Q3 - Ablation study on MPE functions__
>
> We thank the reviewer for this suggestion. In response, we have conducted an ablation study that examines both the emergence of MPE properties in different functions (neural networks) and the effect of using these functions within EquiReg across identical tasks.
>
> Our study evaluates:
>
> 1. The MPE behavior of several publicly available pre-trained networks, as well as a CNN-based autoencoder (AE) that we trained on the data distribution of interest using flip augmentations.
>
> 2. The performance of EquiReg when each of these MPE functions is used on the same inverse problem settings, two datasets (FFHQ and ImageNet), two diffusion-based solvers (DPS and SITCOM), and two tasks (super-resolution and motion deblurring).
>
> Below we summarize our findings, with full details and tables included in the revised manuscript. We evaluated four function classes:
>
> - Pre-trained encoder of latent diffusion models (LDMs): the same choice used in our main experiments.
> - A CNN autoencoder trained by us using flip (for FFHQ) and rotations (for ImageNet) augmentations on the relevant training distribution (test images excluded).
> - Pre-trained ResNet-50, trained on natural images.
> - Pre-trained CLIP (OpenAI).
>
> Our results show:
>
> - All networks exhibit MPE-like behavior: their equivariance error under flip (for FFHQ) and rotation (for ImageNet) symmetry increases as the noise level of the input increases (i.e., as data moves farther from the manifold).
>
> - Some have lower and some higher MPE properties. Notably, the CNN AE trained on the true data distribution shows the strongest MPE property, with equivariance error rising sharply as samples deviate from the manifold, aligning with our systematic guidelines for constructing MPE functions (Section 3). This is the case where the trained data distribution (data manifold) is fully aligned with the data we are solving the inverse problem for (e.g., trained on ImageNet train set, and tested on ImageNet test set).
>
> The results, which we have outlined in detail below, show that a) MPE property has emerged in popular pre-trained functions out there; hence, it is not rare to happen, and b) EquiReg is effective and has improved the performance of diffusion-based inverse problem solvers under all the considered MPE functions. Figure 20 (see updated manuscript) visualizes these trends clearly. See below for tabular results of Figure 20.
>
> _Equi Loss, Evaluated for varying noise levels on 100 images from FFHQ 256_
>
> | MPE Function   | sigma=0      | sigma=0.125   | sigma=0.25    | sigma=0.375   | sigma=0.5    |
> |--------------------------------|--------------|---------------|---------------|---------------|--------------|
> | CNN Autoencoder  | 0.10 (0.03)  | 0.10 (0.03)   | 0.12 (0.03)   | 0.14 (0.04)   | 0.16 (0.04)  |
> | Pretrained LDM Encoder  | 0.08 (0.02)  | 0.11 (0.03)   | 0.11 (0.03)   | 0.11 (0.03)   | 0.11 (0.03)  |
> | Pretrained Resnet50 | 0.06 (0.02)  | 0.06 (0.02)   | 0.07 (0.02)   | 0.07 (0.02)   | 0.07 (0.02)  |
> | Pretrained CLIP          | 0.02 (0.01)  | 0.03 (0.01)   | 0.03 (0.01)   | 0.03 (0.01)   | 0.03 (0.01)  |
>
> _Equi Loss, Evaluated for varying noise levels on 100 images from Imagenet_
>
> | MPE Function    | sigma=0      | sigma=0.125   | sigma=0.25    | sigma=0.375   | sigma=0.5    |
> |-----------------------------------|--------------|---------------|---------------|---------------|--------------|
> | CNN Autoencoder  | 0.09 (0.03)  | 0.10 (0.03)   | 0.12 (0.03)   | 0.13 (0.02)   | 0.14 (0.03)  |
> | Pretrained LDM Encoder | 0.11 (0.03)  | 0.13 (0.02)   | 0.13 (0.03)   | 0.13 (0.02)   | 0.12 (0.02)  |
> | Pretrained Resnet50   | 0.06 (0.01)  | 0.06 (0.01)   | 0.06 (0.02)   | 0.06 (0.02)   | 0.06 (0.01)  |
> | Pretrained CLIP      | 0.03 (0.01)  | 0.03 (0.01)   | 0.03 (0.01)   | 0.03 (0.00)   | 0.03 (0.00)  |
>
> We have also added new visualizations for the encoder MPE function used in our experiments (Figure 15). These results show that its equivariance loss is low for clean images and increases as images are corrupted with various noise levels (Figure 15a). We additionally include a detailed visualization of the loss landscape for a representative example (Figure 15b).
>
> Continue in the next comment ...

---

> > ### Author Response · Authors · 2025-11-21
> > **Authors' Rebuttal for 86xx (4)**
> >
> > _Effectiveness of EquiReg with different MPE functions_
> >
> > We then applied each of these MPE functions within the EquiReg framework on the same inverse problem setups. Across all settings, EquiReg consistently improved reconstruction quality relative to the baseline (no regularization, i.e., None). This demonstrates that:
> >
> > a) MPE properties commonly emerge in widely used pre-trained networks, and thus EquiReg is easy to deploy in practice; and
> >
> > b) EquiReg remains effective across all tested MPE choices, confirming the robustness of the proposed regularization framework.
> >
> > We refer the reviewer to our Appendix for more details (see Tables 14-15). Lastly, we note that in our main paper we used the LDM encoder as our MPE function, even though it exhibits the weakest MPE properties among all functions considered in the new ablation. This suggests that our reported numerical results could be further improved by selecting a stronger MPE function. We leave this exploration to future practical work.
> >
> > _Lambda = 0.01. DPS on FFHQ 256, Superresolution:_
> >
> > | MPE function                    |   PSNR  (PSNR_std)  |  SSIM  (SSIM_std) |  NMSE (NMSE_std) | LPIPS  (LPIPS_std) |   FID   |
> > |:---------------------------|--------:|---------:|-------:|---------:|-------:|
> > | None                       | 23.1595 (1.9227) | 0.6565 ( 0.0718)  | 0.0244 ( 0.0186) | 0.1927 ( 0.0570) | 129.5284 |
> > | LDM Encoder (FFHQ)         | 26.5811 (2.4567) | 0.7730 (0.0438) | 0.0113 ( 0.0094) | 0.1200 (0.0297) |  87.4366 |
> > | CNN Autoencoder (FFHQ)  | 26.8659 (1.9429)  | 0.7713 (0.0437) | 0.0107 (0.0096) | 0.1162 (0.0287) |  85.3520 |
> > | Pretrained Resnet50        | 26.8733 (1.9406) | 0.7713 (0.0437) | 0.0106 (0.0096) | 0.1160 (0.0287) |  85.1376 |
> > | Pretrained CLIP            | 26.8600  (1.9424) | 0.7711 (0.0437) | 0.0107 ( 0.0096) | 0.1162 (0.0288) |  85.4950 |
> >
> > _Lambda = 0.01. DPS on Imagenet, Superresolution:_
> >
> > | MPE function                      |   PSNR  (PSNR_std) |  SSIM  (SSIM_std) |  NMSE  (NMSE_std) | LPIPS  (LPIPS_std) |   FID   |
> > |:----------------------------|--------:|---------:|-------:|---------:|-------:|
> > | None                         | 19.7267 (4.2915) | 0.4074 ( 0.1802) | 0.0699 (0.0350) | 0.5412 ( 0.1821) | 446.8292 |
> > | LDM Encoder (Imagenet)   | 22.2001 (4.2950) | 0.5681 ( 0.1464) | 0.0390 ( 0.0209) | 0.3844 ( 0.1296) | 311.6355 |
> > | CNN Autoencoder (Imagenet) | 22.1777 (4.2943) | 0.5677 (0.1478) | 0.0392 (0.0210) | 0.3753 (0.1250) | 312.5301 |
> > | Pretrained Resnet50  | 22.1763 ( 4.2901) | 0.5677 (0.1477) | 0.0392 ( 0.0210) | 0.3753 (0.1248) | 314.5898 |
> > | Pretrained CLIP         | 22.1774 ( 4.2927) | 0.5676 ( 0.1477) | 0.0392 (0.0210) | 0.3757 ( 0.1248) | 313.4681 |
> >
> > _Lambda = 0.05. SITCOM on FFHQ 256, Motion Deblur:_
> >
> > | MPE function      | psnr_mean (psnr_std) | ssim_mean (ssim_std) | lpips_mean ( lpips_std) | time_sec_mean (time_sec_std) |
> > |:---------------------------|----------:|---------:|----------:|---------:|
> > | None                       |   27.6695 (1.3434) |   0.7903  ( 0.0305) |    0.2212  (0.0399)  |       23.3611 ( 0.8936)  |
> > | LDM Encoder (FFHQ)    |   28.3565 (1.3787) |   0.8058  ( 0.0314) |    0.2001  ( 0.0358)  |       24.1549 (0.5565) |
> > | CNN Autoencoder (FFHQ) |   28.8522 (1.3758) |   0.8188  (0.0437) |    0.1932  (0.0327)  |    24.1831 (0.7417) |
> > | Pretrained Resnet50     |   28.6819 (1.3884) |   0.8109  (0.0292) |    0.1977  ( 0.0361)  |       24.0940 (0.6554) |

---

> ### Author Response · Authors · 2025-11-21
> **Authors' Rebuttal for 86xx (6)**
>
> To conclude, we thank the reviewer for their thoughtful comments. We hope that our responses above, together with the revised manuscript (see our general response for a clear explanation of all new content and how it complements the original submission), adequately address the concerns raised.
>
> We look forward to hearing from the reviewers regarding any remaining questions or suggestions on how we may further improve the clarity, presentation, or overall contribution of the paper. In light of the revised manuscript addressing all reviewer feedback and questions, we kindly ask the reviewer to consider revising their score.

---

> ### Author Response · Authors · 2025-11-21
> **Authors' Rebuttal for 86xx (5)**
>
> __W4/Q4 - Symmetry group size and large cardinality__
>
> We thank the reviewer for raising this important question. The concern highlights a useful characteristic of EquiReg. Because EquiReg acts as a regularization scheme rather than an exact equivariance constraint, it does not require enumerating the full symmetry group. Instead, EquiReg remains effective even when only a subset of group actions is sampled.
>
> To clarify this point, we have added new experiments examining the choice and size of symmetry group actions for image restoration tasks. Specifically, we compare different subsets of rotation symmetries and find that even using only a small subset (e.g., ${0^\circ, 180^\circ}$) yields substantial improvement over no regularization:
>
> |Setting | PSNR | SSIM |
> |---------:|----------:|--------:|
> |PSLD | 15.86 (1.19) | 0.77 (0.03)|
> |EquiReg-PSLD (90, 270 deg)  | 17.60 (1.60) | 0.79 (0.03)|
>
> These results (now included as Table 10) indicate that it is not necessary to explore the full cardinality of the group for the regularizer to be effective.
>
> In addition, we now include new results using a large‐cardinality rotation group for 2D images. Here, we sample a uniformly random rotation from ${0^\circ, 1^\circ, …, 359^\circ}$ at each diffusion step. Even with this large cardinality symmetry group, EquiReg remains stable and improves reconstruction quality:
>
> _Lambda = 0.01. DPS on Imagenet 256, Superresolution_
> | case | PSNR_mean (PSNR_std) | SSIM_mean (SSIM_std) | LPIPS_mean (LPIPS_std) |    FID   |
> |:---------------------------------------|----------:|---------:|----------:|--------:|
> | DPS + Equi (1–360º rotations)     |    23.51 (5.49) |    0.60 ( 0.1964) |     0.36  (0.1744) | 251.47 |
> | DPS + Equi (90º rotations)          |    23.47 ( 5.34) |    0.61 ( 0.1973) |     0.36  (0.1750) | 251.90 |
> | DPS baseline                           |    20.64 (5.12)  |    0.47 (0.2513)  |     0.52   (0.1914) | 359.86 |
>
> These experiments, performed on DPS on ImageNet dataset for the super-resolution task, are now included as Table 14 in the main paper. They demonstrate that EquiReg does not rely on full group coverage. Sampling even a sparse or randomly chosen subset of group actions is sufficient, as long as the function used for regularization exhibits the MPE property across the group.
>
> In summary, EquiReg is compatible with both small and large symmetry groups, and in practice, sampling a subset of group actions is enough for effective regularization.

---

### Author Response · Authors · 2025-11-21
**Summary of New Content and Experiments Addressing Reviewers’ Comments (1)**

We thank all the reviewers for the comments and suggestions. Their feedback has helped us to clarify the contribution of our paper, and include new experimental analysis complemented with visualization to improve the quality of the paper. Below, we summarize the newly added content, all of which are highlighted in color green in the revised manuscript.

__Figures are revised__

- Given the reviewers’ suggestions, we have revised Figures 2 and 3 for an improved visualization of MPE and EquiReg. We now provide main figures in pdf format.

__New experiment on EquiReg for text-to-image guidance - classifier-free guidance (CFG)__

- We have conducted additional experiments when EquiReg is applied to DreamSampler [1] (see Section A of the Appendix). The experiment shows how EquiReg helps to improve perceptual quality of the generated images. In Figure 8, we have observed an implicit acceleration of image generation when EquiReg is imposed. In this case, Equi-DreamSampler with 50 DDIM steps can generate images that are only possible with DreamSampler when the DDIM steps are increased. These complement the experiment we visualized in Figure 1e. Please see Figures 8-12 for the included visualizations.

[1] Kim, J., Park, G. Y., & Ye, J. C. (2024, September). Dreamsampler: Unifying diffusion sampling and score distillation for image manipulation. In European Conference on Computer Vision (pp. 398-414).

__New experiment for implicit efficiency of EquiReg by reducing DDIM steps__

- To further highlight the efficiency of EquiReg, we include new experimental analyses on super-resolution showing that although EquiReg introduces an additional regularization computation, its manifold-preference effect provides an implicit acceleration. This allows us to reduce the number of DDIM steps during generation or posterior sampling without degrading performance. Table 8 compares DPS and Equi-DPS when the number of DDIM steps is reduced from 1000 to 900, 750, and 500. While DPS performance degrades substantially under reduced steps, Equi-DPS continues to produce acceptable solutions. These findings complement the reduced measurement-consistency experiments in Table 1 and the runtime comparisons reported in Table 2.

__New experiment on further highlight the robustness of EquiReg regularization parameters__

- We found it useful to include an additional experiment highlighting the robustness of EquiReg with respect to the choice of the regularization parameter. Table 7 reports a sensitivity analysis for DPS and PSLD both without EquiReg ($\lambda = 0$) and with EquiReg ($\lambda > 0$) over a range of values (0.001, 0.01, 0.1, 0.25, 1). The results demonstrate that EquiReg remains effective across a broad range of $\lambda$, which is important in practice when users may not be able to finely tune the regularization parameter.

__Extended SITCOM experiments demonstrating EquiReg improvements under fixed measurement-consistency steps__

- We now complement the SITCOM results in Table 1 with additional experiments in Table 9, where we fix the number of measurement-consistency steps and include one extra EquiReg regularization step. This setting aligns with all other comparisons in Tables 2–5, where the number of DDIM and measurement steps is kept constant between each baseline and its EquiReg-enhanced counterpart. The results further show that EquiReg remains beneficial even when the solver struggles to find a good solution (e.g., at low $K_{\text{meas}}$ such as 10, 20, or 30). As in Table 1, we use 50 DDIM steps on the motion-deblurring task.

__Complementary evaluation with histograms and visualizations__

- Following the reviewer’s suggestion, we now include Figure 14, which reports the histogram of PSNR improvements obtained by applying EquiReg to DPS. This highlights how EquiReg improves the population-level reconstruction statistics of test images, complementing the numerical results in the main tables. We also add visualizations of image-restoration results in Figures 16 and 17, comparing EquiCon-PSLD vs. PSLD on FFHQ and Equi-DPS vs. DPS on ImageNet. These examples illustrate the artifacts introduced by the baseline methods and how EquiReg effectively removes them in the final reconstructions.

---

> ### Author Response · Authors · 2025-11-21
> **Summary of New Content and Experiments Addressing Reviewers’ Comments (2)**
>
> __Systematic guidelines and ablation study on MPEs__
>
> - We now provide systematic guidelines for choosing an MPE function and corresponding group actions for EquiReg. We have added references to the equivariance literature demonstrating the broad applicability of reflection and rotation groups across domains grounded in physical reality, including molecular dynamics, robotics, reinforcement learning agents operating in 3D environments, and computer vision (see our response to Reviewer Y4hA for details).
>
> - Additionally, we include new experiments evaluating the MPE properties of several publicly available networks and applying them within EquiReg to improve diffusion-based inverse solvers. We also train our own networks following our suggested guidelines in Section 3 and evaluate their MPE properties. These results further emphasize the generality of the choice of the function f in the Equi loss (see our response to Reviewer 86xx). We also complement these experiments with the visualization in Figure 15b, illustrating how the Equi loss increases as data deviates from the natural image distribution. Our set of experiments are finalized, and we will post the full set of results from these experiments tomorrow (see Section I).
>
> __For better reproducibility and implementations__
>
> - We have included additional instructions on how EquiReg is being applied to each of the baselines we considered in our paper (see Algorithms  2–7) along with an Assets Section on their source codes, and a Computing Resources Section.
>
>
> __Theoretical analysis complementing the proposed regularization for posterior sampling__
>
> - In response to the reviewers’ feedback, we have revised Section 3 to better motivate EquiReg, explain its mechanism, and clarify the effect of the regularization on posterior sampling trajectories. The added explanations are marked in green in the main paper. We further provide detailed theoretical discussion in the appendix (see Section G). For improved clarity, we are happy to move any of this material into the main text should the reviewers or AC recommend it.
>
> __Diversity__
>
> - We have conducted extensive numerical analyses to evaluate the ability of EquiReg to preserve or improve the diversity of posterior samples when solving inverse problems (see Page 9 and Section D of the Appendix, which includes Figure 7, Table 11, Figures 18-19).
>
> ___
>
> While we summarize the key points above, we encourage the reviewers to also read our reviewer-specific responses and the green-highlighted additions in the revised manuscript. We have done large number of experiments to address the reviewers' concerns. Hence, we will complete our response within a day. We thank the reviewers again, and look forward to hearing from the reviewers on any remaining questions.

---

### Meta-Review · Area_Chair_aL5r · 2026-01-01

**Summary:**

The submission presents an interesting idea of using equivariance for regularization in diffusion-based inverse problems. However, the theoretical grounding, the justification for the core MPE component, and the empirical evidence regarding diversity preservation are not yet strong enough for acceptance. The substantial additions made during the rebuttal suggest that the paper was not fully ready at the time of submission. Therefore, the recommendation is to reject the paper, encouraging the authors to integrate the new material fully into the main manuscript and address the remaining fundamental questions for a future submission.

**Reviewer Concerns:**

* Concern 1: Insufficiently rigorous mechanism/theory linking equivariance to “on-manifold” sampling. It is unclear whether low equivariance error truly implies on-manifold behavior.

The authors provided a more explicit explanation of off-manifold errors arising from the isotropic-Gaussian likelihood approximation, added new Figure 3, and expanded the theory in Appendix G to motivate why an “on-manifold–low / off-manifold–high” regularizer is desirable and how EquiReg approximates this.

* Concern 2: choosing symmetry/MPE functions in practice, large-cardinality groups, efficiency, and diversity trade-offs. The method hinges on nontrivial choices of symmetry groups/MPE functions, may not scale to large/continuous groups, could add computational overhead, and might reduce posterior diversity.

The authors added guidelines and citations on symmetry selection, ran more ablations comparing multiple MPE functions (LDM encoder, AE, ResNet, CLIP), added experiments for large-cardinality rotations (random 1–360°) and subset sampling, provided efficiency studies, and expanded diversity experiments plus new qualitative visualizations. The results show that EquiReg preserves meaningful posterior variability while improving fidelity.

**Reviewer Scores:**

none of the reviewer would substantially raise the scores for acceptance

---

### Decision · Program_Chairs · 2026-01-26

Reject